# Fungal sensing by dectin-1 directs the non-pathogenic polarization of T$_H$17 cells through balanced type I IFN responses in human DCs

Sonja I. Gringhuis [1,2] ✉, Tanja M. Kaptein[1,2], Ester B. M. Remmerswaal[1,2,3], Agata Drewniak[1,2], Brigitte A. Wevers[1,2], Bart Theelen [4], Geert R. A. M. D'Haens[5,6], Teun Boekhout[4,7] & Teunis B. H. Geijtenbeek [1,2] ✉

The non-pathogenic T$_H$17 subset of helper T cells clears fungal infections, whereas pathogenic T$_H$17 cells cause inflammation and tissue damage; however, the mechanisms controlling these distinct responses remain unclear. Here we found that fungi sensing by the C-type lectin dectin-1 in human dendritic cells (DCs) directed the polarization of non-pathogenic T$_H$17 cells. Dectin-1 signaling triggered transient and intermediate expression of interferon (IFN)-β in DCs, which was mediated by the opposed activities of transcription factors IRF1 and IRF5. IFN-β-induced signaling led to integrin αvβ8 expression directly and to the release of the active form of the cytokine transforming growth factor (TGF)-β indirectly. Uncontrolled IFN-β responses as a result of IRF1 deficiency induced high expression of the IFN-stimulated gene *BST2* in DCs and restrained TGF-β activation. Active TGF-β was required for polarization of non-pathogenic T$_H$17 cells, whereas pathogenic T$_H$17 cells developed in the absence of active TGF-β. Thus, dectin-1-mediated modulation of type I IFN responses allowed TGF-β activation and non-pathogenic T$_H$17 cell development during fungal infections in humans.

DCs orchestrate differentiation of helper T (T$_H$) cells for host immunity and tolerance[1]. Pathogen-recognition receptors (PRRs) expressed by DCs sense invading pathogens and induce cytokine profiles that direct the development of appropriate T$_H$ cell subsets[2]. The C-type lectin receptor (CLR) dectin-1 recognizes the fungal cell wall polysaccharide β-glucan[3,4] and triggers signaling through both Syk-CARD9 and LSP1-Raf-1 to induce interleukin (IL)-12 to instruct T$_H$1 differentiation and IL-1β, IL-23 and IL-6 to induce T$_H$17 differentiation[5–7]. IL-17A and

IL-17F production by T$_H$17 cells is crucial for clearing fungal infections through the recruitment of neutrophils and the generation of antimicrobial peptides by epithelial cells[8]. Patients with dectin-1 or CARD9 deficiencies have recurrent and often severe fungal infections due to impaired T$_H$17 responses[9,10], underscoring the importance of dectin-1 signaling for antifungal immunity.

In addition to their key role in host defense and mucosal tissue homeostasis, T$_H$17 cells are also the driving force behind several inflammatory

[1]Department of Experimental Immunology, Amsterdam UMC - location AMC, University of Amsterdam, Amsterdam, The Netherlands. [2]Amsterdam Institute for Infection and Immunity, Amsterdam, The Netherlands. [3]Renal Transplant Unit, Amsterdam UMC - location AMC, University of Amsterdam, Amsterdam, The Netherlands. [4]Westerdijk Fungal Biodiversity Institute, Utrecht, The Netherlands. [5]Gastroenterology and Hepatology, Amsterdam UMC - location AMC, University of Amsterdam, Amsterdam, The Netherlands. [6]Amsterdam Gastroenterology Endocrinology Metabolism, Amsterdam, The Netherlands. [7]Institute for Biodiversity and Ecosystem Dynamics (IBED), University of Amsterdam, Amsterdam, The Netherlands. ✉e-mail: s.i.gringhuis@amsterdamumc.nl; t.b.geijtenbeek@amsterdamumc.nl

**Fig. 1 | Dectin-1 triggering on DCs induces transient intermediate type I IFN responses. a–d**, Real-time PCR analyses of *IFNB* relative mRNA levels in monocyte-derived DCs with wild-type dectin-1 expression (**a–d**) or DCs that lack functional dectin-1 expression due to the presence of the Y238X mutation (ΔDectin-1 DCs) (**c**), left unstimulated or after stimulation with curdlan (**a–d**), LPS (**a,c**), poly(I:C)-LyoVec (poly(I:C)-LV) (**a**), and *C. albicans* CBS2712, *C. albicans* CBS8758, *C. dubliniensis* CBS8500, *C. lusitaniae* CBS4414 and *A. fumigatus* CBS488.90 (**b,c**), in the presence of blocking dectin-1 antibodies or isotype IgG2b control antibodies (**c**) or after transduction with non-targeting (control) or specific siRNAs to silence Syk, CARD9, Bcl10, Malt1, LSP1 or Raf-1 expression (**d**), at indicated times (**a,b**, curdlan n = 8; other stimuli n = 4) or 2 h (**c**, curdlan, *C. albicans* CBS2712 n = 12; *C. albicans* CBS8758 n = 8; *C. dubliniensis* CBS8500, *C. lusitaniae* CBS4414, *A. fumigatus* CBS488.90 n = 4; LPS n = 3, except for all stimuli with ΔDectin-1 DCs n = 3; **d**, curdlan, n = 4, except with LSP1 and Raf-1 siRNA n = 2). **e,f**, Real-time PCR analyses of IFN-stimulated genes *MX1*, *ISG15*, *TRIM22*, *IRF7* and *IL27A* relative mRNA levels in unstimulated DCs or after stimulation of DCs with curdlan or *C. albicans* CBS2712 (hereafter referred to as *C. albicans*), in the presence of blocking dectin-1 or IFN-α/βR antibodies or isotype (IgG2b and IgG2a) control antibodies (**f**), at indicated times (**e**, n = 4) or 6 h (**f**, curdlan n = 7, except with IgG2b and IgG2a n = 3; *C. albicans* n = 4). ND, not determined. Results from real-time PCR were normalized to the expression of reference household gene *GAPDH* and shown relative to 2 h (**a–d**) or 6 h (**e,f**) curdlan stimulation, except for ΔDectin-1 DCs where results are shown relative to 2 h LPS stimulation (**c**). Data in **a–f** represent mean ± s.d. of independent donors. **P < 0.01 (paired, two-tailed Student's *t*-test), calculated between untreated and treated (**c,f**) or control and specific siRNA-transduced (**d**) samples that were likewise stimulated.

and autoimmune disorders. Cellular heterogeneity underlies these differences as T_H17 cells adopt different functional phenotypes depending on environmental cues and genetic factors[11,12]. T_H17 cells coexpressing anti-inflammatory IL-10 are considered non-pathogenic, restrain pro-inflammatory responses and limit host tissue damage[11,13], whereas T_H17 cells expressing IFN-γ and its transcriptional regulator T-bet are commonly linked with inflammatory immunopathogenesis in for example

Crohn's disease, diabetes and multiple sclerosis[14–16]. Adoptive transfer of in vitro polarized T cells in mice revealed that IL-6 and TGF-β1 induces non-pathogenic T_H17 cells, whereas IL-6/IL-1β/IL-23 or IL-6/TGF-β3 induces pathogenic T_H17 responses[17–19]. The polarization requirements for human pathogenic T_H17 cells remain poorly defined. Furthermore, it is unclear whether or how PRRs expressed by DCs instruct T_H17 immune responses toward a specific phenotype.

DCs produce type I IFNs to different classes of pathogens. While type I IFN responses are essential in antiviral immunity through the induction of antiviral effector molecules encoded by IFN-stimulated genes (ISGs)[20], their role in fungal infections is still unknown. *Candida albicans* stimulation of human peripheral blood mononuclear cells leads to type I IFN responses[21]. In mice, dectin-1, and to a lesser extent dectin-2, contribute to type I IFN expression by DCs during *C. albicans* infection, which results in recruitment of inflammatory cells but also causes hyper-inflammation and lethal kidney pathology[22,23].

Here, we demonstrated that recognition of curdlan and *C. albicans* by human dectin-1 initiated mostly non-pathogenic $T_H17$ responses. Dectin-1 triggering led to transient and intermediate expression of IFN-β by balancing the activities of the transcription factors IRF1 and IRF5. Type I IFN signaling induced the expression of integrin αvβ8, which allowed the MMP14-mediated cleavage of the latent TGF-β complex to release active TGF-β, which in turn resulted in the development of primarily non-pathogenic $T_H17$ cells. Our data underscore the importance of a tightly controlled type I IFN response, as high amounts of IFN-β inhibited MMP14 through the ISG *BST2* and increased pathogenic $T_H17$ polarization. Thus, dectin-1 regulated type I IFN expression to allow the TGF-β-mediated development of non-pathogenic $T_H17$ responses during fungal infections in humans.

## Results

### Dectin-1 induces transient intermediate IFN-β expression

In vitro stimulation of human monocyte-derived DCs (hereafter referred to as DCs) with the dectin-1-specific β-glucan ligand curdlan induced the transient expression of *IFNB* messenger RNA (mRNA) that peaked at 2 h post-stimulation (Fig. 1a). For comparison, the Toll-like receptor (TLR) 4 ligand lipopolysaccharide (LPS) induced a higher peak of *IFNB* mRNA at 2 h, followed by a secondary peak at 6 h post-stimulation, whereas RIG-I-like receptor ligand poly(I:C)-LyoVec induced higher expression of *IFNB* mRNA that plateaued 4 h post-stimulation (Fig. 1a). Various commensal and pathogenic yeast and filamentous fungal species, such as *C. albicans*, *C. dubliniensis*, *C. lusitaniae* and *Aspergillus fumigatus* induced transient *IFNB* transcription (Fig. 1b). Both curdlan- and fungi-induced, but not LPS-induced, *IFNB* responses were inhibited by blocking antibodies to dectin-1, but not isotype control antibodies, and were absent in DCs derived from donors without functional dectin-1 expression due to the Y238X mutation (Fig. 1c). Dectin-1 triggers signaling through Syk-CARD9-Bcl10-Malt1 and LSP1-Raf-1 (ref. [2]). Silencing of either Syk, CARD9, Bcl10 or Malt1 expression in DCs by transduction of specific short interfering RNAs (siRNAs) (Extended Data Fig. 1) blocked curdlan-induced *IFNB* expression, whereas silencing of LSP1 or Raf-1 had no effect (Fig. 1d). Curdlan and *C. albicans* strain CBS2712 (hereafter referred to as *C. albicans*) also induced mRNA expression for the ISGs *MX1*, *ISG15*, *TRIM22*, *IRF7* and *IL27A* (encoding IL-27 subunit p28)[20,24] (Fig. 1e), which was completely abrogated by blocking antibodies to dectin-1 or IFN-α/βR, but not isotype control antibodies (Fig. 1f).

Thus, fungal ligands trigger transient and intermediate type I IFN responses through dectin-1 in human DCs.

### Dectin-1 induces non-pathogenic $T_H17$ cells through IFN-β

To investigate the role of type I IFN responses in dectin-1-induced $T_H17$ differentiation, we co-cultured DCs that were primed for 48 h with curdlan or *C. albicans* with naive CD45RA⁺CD4⁺ T cells and determined the proliferation and intracellular expression of IL-17 and IFN-γ of the cultured T cells. Curdlan- and *C. albicans*-primed, but not immature DCs, induced strong T cell proliferation as measured by cell count and tracking cell division by carboxyfluorescein succinimidyl ester staining (Extended Data Fig. 2). At day 5, curdlan- and *C. albicans*-primed DCs mainly induced IFN-γ⁻IL-17⁺ T cells (81 ± 6.1% and 80 ± 5.5% of IL-17⁺ cells, respectively) (Fig. 2a–d and Extended Data Fig. 3). Development of IL-17⁺ cells directed by curdlan- or *C. albicans*-primed DCs (6.3 ± 1.1% and 6.7 ± 1.0% of total cells, respectively) was partly blocked by neutralizing antibodies against IL-1β (3.4 ± 0.9% and 3.5 ± 0.7%) or IL-23 (2.9 ± 1.0% and 3.0 ± 1.1%), but not IL-12 (6.9 ± 0.8% and 6.7 ± 0.9%) (Extended Data Fig. 4). T cell proliferation (Extended Data Fig. 2) or the overall differentiation of IL-17⁺ cells induced by curdlan- and *C. albicans*-primed DCs (11 ± 4.0% and 9.6 ± 3.5% of total cells, respectively) was not affected by blocking IFN-α/βR antibodies (11 ± 3.4% and 11 ± 2.9%) (Fig. 2a,c). In contrast, the addition of recombinant human (rh)IFN-β decreased the overall differentiation of $T_H17$ cells induced by curdlan- and *C. albicans*-primed DCs (2.5 ± 0.9% and 2.8 ± 1.1% of total cells, respectively) (Fig. 2b,c), while not affecting their proliferation (Extended Data Fig. 2), suggesting that type I IFN was not required for the production of IL-17 in CD4⁺ T cells, whereas high amounts of IFN-β blocked it, with dectin-1-induced IFN-β expression too low to affect $T_H17$ cell development. Notably, the frequency of IFN-γ⁺IL-17⁺ T cells induced by curdlan- and *C. albicans*-primed DCs (19 ± 6.1% and 21 ± 5.5% of IL-17⁺ cells, respectively) was increased by blocking IFN-α/βR antibodies (65 ± 5.5% and 67 ± 4.0%) (Fig. 2a,d). Supplementary rhIFN-β, while suppressing overall $T_H17$ differentiation, also increased the frequency of IFN-γ⁺IL-17⁺ T cells induced by curdlan- and *C. albicans*-primed DCs (61 ± 14% and 56 ± 6.4% of IL-17⁺ cells, respectively) (Fig. 2b,d).

We next co-cultured curdlan- or *C. albicans*-primed DCs with memory CD45RO⁺CD4⁺ T cells together with *Staphylococcus aureus* enterotoxin B (SEB), which induces antigen-independent T cell activation, to examine how dectin-1-induced cytokines affect the phenotype of differentiated $T_H$ cells. Immature as well as curdlan- and *C. albicans*-primed DCs induced strong T cell proliferation (Extended Data Fig. 2). At day 5, we detected IL-17⁺ cell differentiation in response to curdlan- and *C. albicans*-primed DCs (3.6 ± 0.9% and 3.1 ± 0.6% of total cells, respectively), but not immature DCs, whereas at day 11–14, when T cells were quiescent, IL-17⁺ cells represented 6.2 ± 2.9% and 5.9 ± 3.2% of total cells (Extended Data Figs. 3 and 4). Induction of IL-17⁺ cells at day 11–14 by curdlan- and *C. albicans*-primed DCs was blocked by IL-1β (1.4 ± 0.5% and 1.6 ± 0.8% of total cells, respectively) and IL-23 (1.5 ± 0.7% and

**Fig. 2 | Dectin-1 instructs non-pathogenic $T_H17$ polarization via IFN-β, while absence or excess IFN-β results in pathogenic $T_H17$ responses. a–h**, Flow cytometry analyses of $T_H$ polarization by staining for intracellular IL-17 and IFN-γ expression (FI) in restimulated T cells, outgrown in vitro by co-culture of either naive (Tn, **a–d**) or memory (Tm, **e,f**) CD4⁺ T cells with immature DCs (iDC) or DCs primed for 48 h with curdlan or *C. albicans*, in the presence of blocking IFN-α/βR antibodies (**a,c–e,g,h**) or rhIFN-β (**b–d,f–h**), at day 5 (**a–d**, IFN-α/βR antibody *n* = 4, rhIFN-β *n* = 3) or day 11–14 (**e–h**, IFN-α/βR antibody *n* = 8, rhIFN-β *n* = 7). FI, fluorescence intensity. In **a,b,e,f**, representative dot plots for independent donors are shown, with percentage positive cells indicated in each quadrant. In **c,g**, the percentage IL-17⁺ cells per total amount of T cells are shown and in **d,h**, the percentage IFN-γ⁻ and percentage IFN-γ⁺ cells per IL-17⁺ T cells are shown. Data in **c,d,g,h** represent mean ± s.d. of independent donors. ***P* < 0.01, **P* < 0.05 (paired, two-tailed Student's *t*-test), calculated between untreated and treated samples that were likewise stimulated. **i–k**, Real-time PCR analyses of *IFNG*, *TBX21*, *IL1R1*, *CXCL3*, *CCL3*, *CSF2*, *IL22*, *CCL5*,

*IL17A*, *IL17F*, *RORC*, *IL10*, *MAF* and *PTGDS* normalized mRNA levels in IL-17⁺ T cells, isolated by flow cytometry-based sorting from either blood of healthy donors (**i**, *n* = 4) or patients with active Crohn's disease (**i**, *n* = 4) or after $T_H$ polarization by co-culture of either naive or memory CD4⁺ T cells with curdlan- or *C. albicans*-primed DCs, as described for **a–h**, at day 5 (**j**, Tn outgrowth *n* = 4) or day 11–14 (**j**, Tm outgrowth *n* = 12; **k**, *n* = 4). In **j**, IL-17⁺ cells were further divided during sorting based on intracellular IFN-γ expression. Results from real-time PCR were quantified using standard curves for all genes and normalized to the expression of reference household gene *ACTB*. Data represent mean (number within square) of independent donors; more detailed data are presented in Extended Data Figs. 5i,j and 6k. *IFNG*, *TBX21*, *IL1R1* and *CXCL3* expression was more than five times higher (red); *CCL3*, *CSF2* and *IL22* expression was between two and five times higher (purple); *CCL5*, *IL17A*, *IL17F* and *RORC* expression varied between equal and two times higher (gray); and *IL10*, *MAF* and *PTGDS* expression was lower (green) in IFN-γ⁺IL-17⁺ T cells compared to IFN-γ⁻IL-17⁺ T cells.

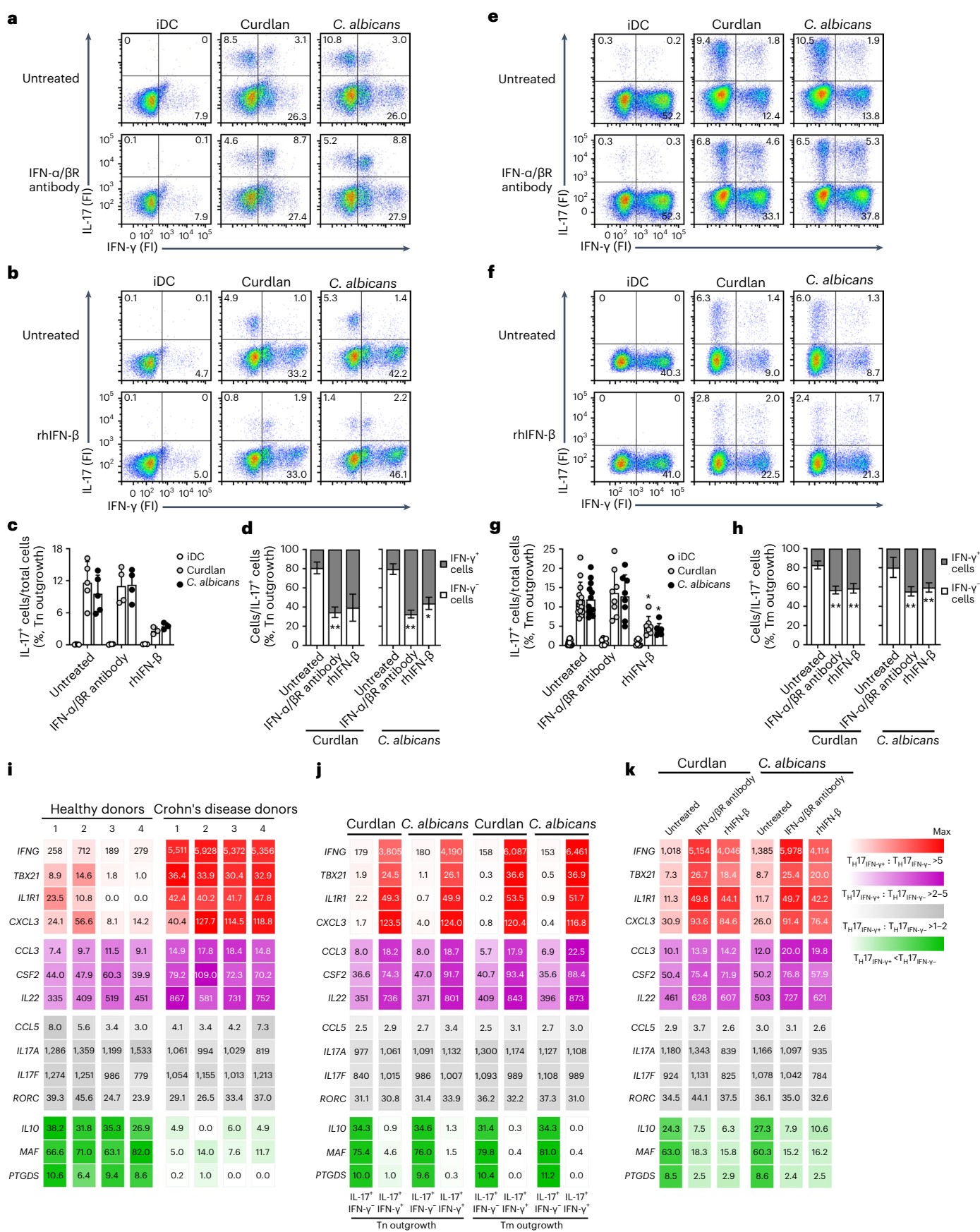

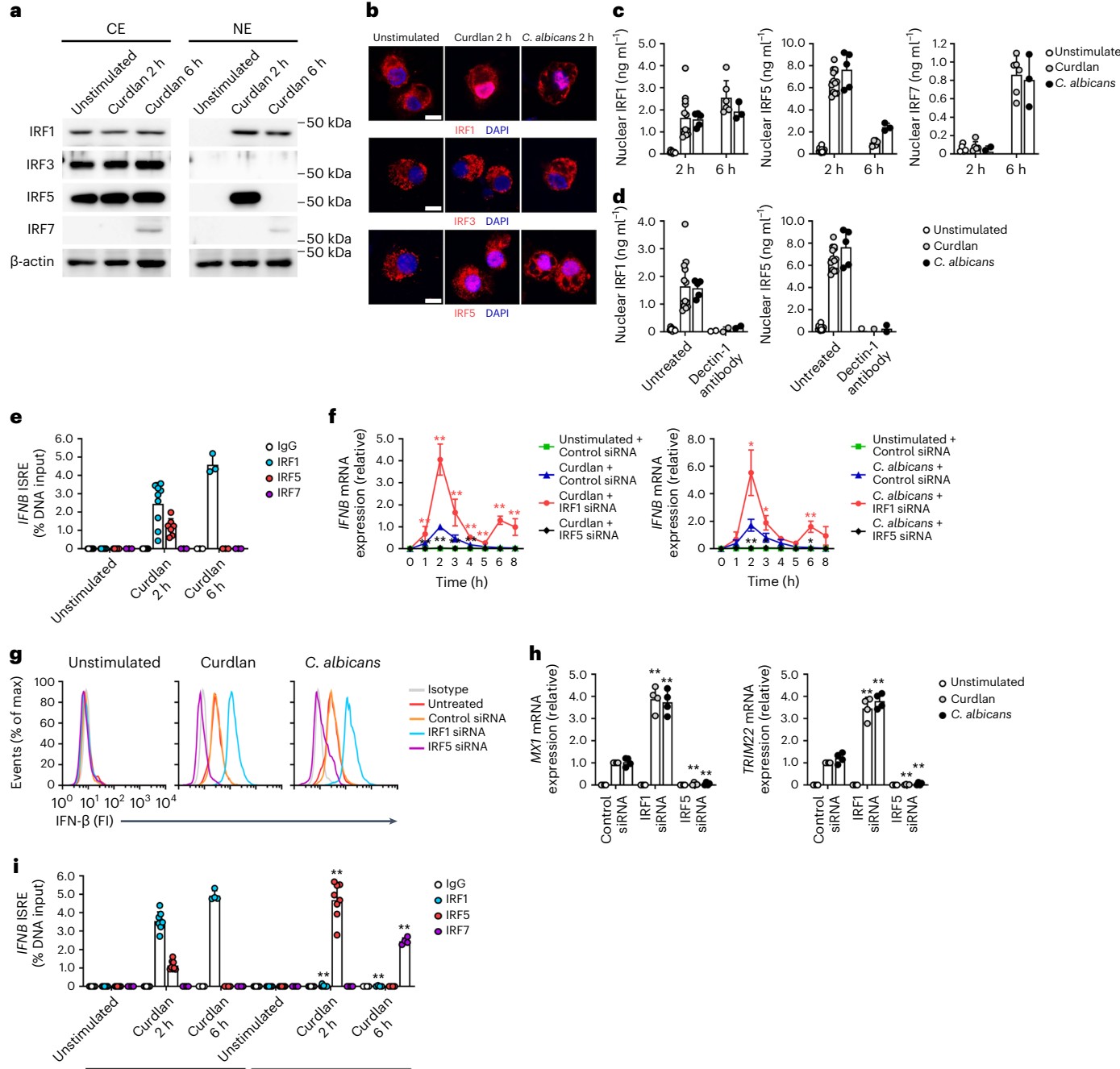

**Fig. 3 | Dectin-1 signaling tightly regulates IFN-β expression through opposite actions of IRF1 and IRF5. a**, Immunoblot analyses of cellular localization of IRF1, IRF3, IRF5 and IRF7 in cytoplasmic (CE) and nuclear (NE) extracts of unstimulated DCs or after stimulation of DCs with curdlan at 2 or 6 h (*n* = 3). β-actin, run on a separate gel, served as loading control. Molecular weights are indicated on the right. **b**, Confocal immunofluorescence analyses of cellular localization of IRF1, IRF3 and IRF5 (red) and DNA (DAPI staining, blue) in unstimulated DCs or after stimulation of DCs with curdlan or *C. albicans* at 2 h (*n* = 3). DAPI, 4,6-diamidino-2-phenylindole. Scale bars, 8 μm. Representative images for independent donors are shown (**a**,**b**). **c**,**d**, ELISA for quantification of IRF1, IRF5 and IRF7 expression in nuclear extracts of unstimulated DCs or after stimulation of DCs with curdlan or *C. albicans* (**c**), in the presence of blocking dectin-1 antibodies (**d**), at 2 h (**c**,**d**, IRF1, IRF5 curdlan *n* = 13; IRF7 curdlan *n* = 6; IRF1, IRF5 *C. albicans* *n* = 5; IRF7 *C. albicans* *n* = 3; **d**, dectin-1 antibody *n* = 2) or 6 h (**c**, curdlan *n* = 6, *C. albicans* *n* = 3). **e**,**i**, Chromatin immunoprecipitation (ChIP) assay of IRF1, IRF5 and IRF7 recruitment to the ISRE binding motif of the *IFNB* promoter in unstimulated DCs or after stimulation of DCs with curdlan (**e**), after transduction with non-targeting (control) or specific siRNAs to silence IRF1 expression (**i**), at

2 h (**e**, IgG1, IRF1 immunoprecipitation (IP) *n* = 9, IRF5 IP *n* = 7, IRF7 IP *n* = 3; **i**, IgG1, IRF1, IRF5 IP *n* = 8, IRF7 IP *n* = 4) or 6 h (**e**, *n* = 3; **e**, *n* = 4). IgG indicates negative controls. Results are expressed as percentage input DNA. Data in **c**–**e**,**i** represent mean ± s.d. of independent donors. **\*\****P* < 0.01 (paired, two-tailed Student's *t*-test), calculated between untreated and treated samples that were likewise stimulated. **f**,**h**, Real-time PCR analyses of *IFNB*, *MX1* and *TRIM22* relative mRNA levels in unstimulated DCs or after stimulation of DCs with curdlan or *C. albicans*, after silencing of IRF1 or IRF5 expression, at indicated times (**f**, curdlan with IRF1 siRNA *n* = 9, curdlan with IRF5 siRNA *n* = 5, *C. albicans* *n* = 4) or 6 h (**h**, *n* = 4). Results from real-time PCR were normalized to the expression of reference household gene *GAPDH* and shown relative to 2 h (**f**) or 6 h (**h**) curdlan stimulation. Data in **f**,**h** represent mean ± s.d. of independent donors. **\*\****P* < 0.01, **\****P* < 0.05 (paired, two-tailed Student's *t*-test), calculated between control and specific siRNA-transduced samples that were likewise stimulated. **g**, Flow cytometry analyses by staining for intracellular IFN-β expression (FI) in unstimulated DCs or after stimulation of DCs with curdlan or *C. albicans*, after silencing of IRF1 or IRF5 expression at 6 h (*n* = 2). Isotype indicates negative control staining. Representative histograms for independent donors are shown.

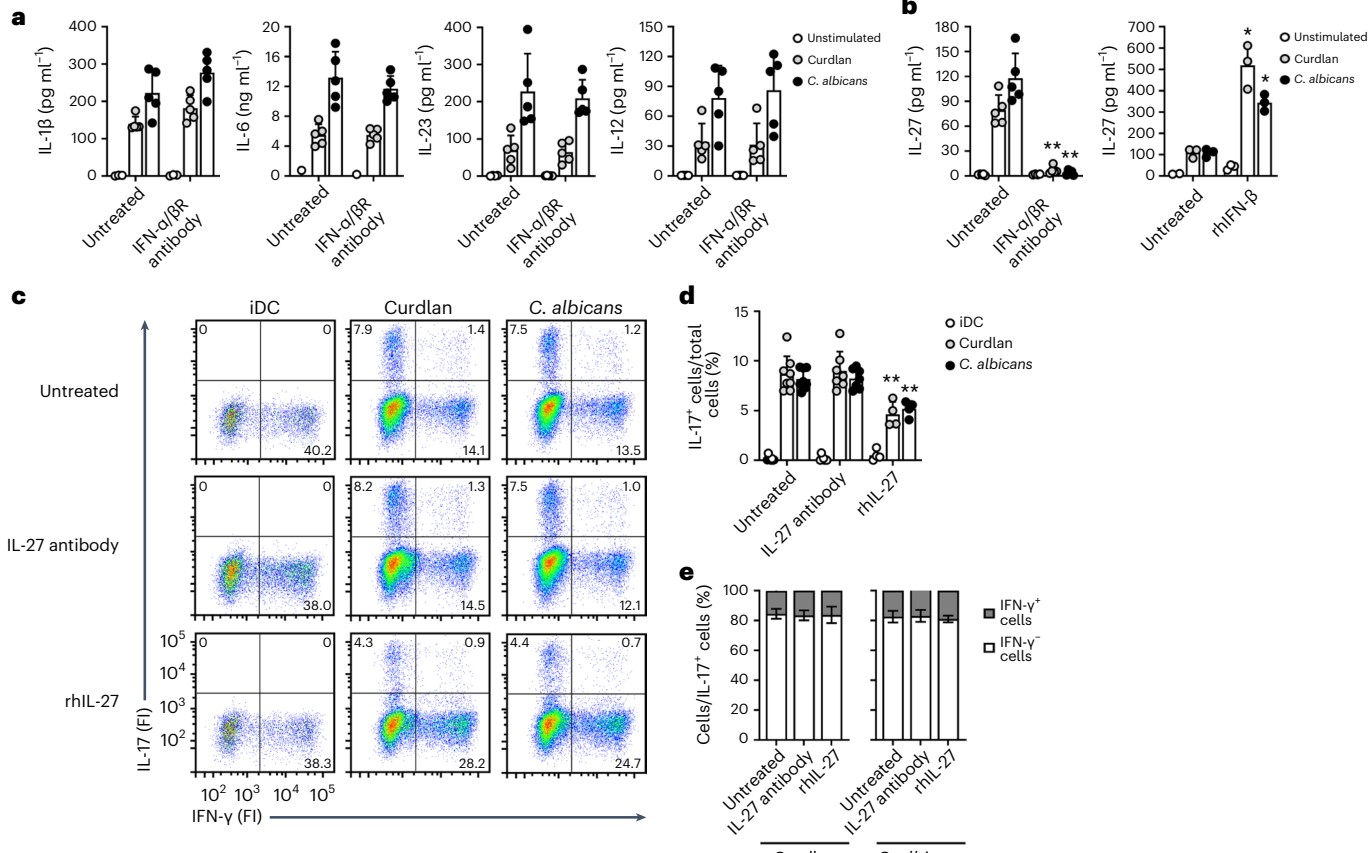

**Fig. 4 | Dectin-1 does not instruct IFN-β-mediated non-pathogenic T$_H$17 polarization via IL-1β, IL-6, IL-23, IL-12 or IL-27. a,b**, ELISA for quantification of IL-1β, IL-6, IL-23, IL-12 and IL-27 in the supernatant of unstimulated DCs or after stimulation of DCs with curdlan or *C. albicans*, in the presence of blocking IFN-α/βR antibodies (**a,b**) or rhIFN-β (**b**) at 28 h (**a**, *n* = 5; **b**, IFN-α/βR antibody *n* = 5, rhIFN-β *n* = 3). Data in **a,b** represent mean ± s.d. of independent donors. **P < 0.01, *P < 0.05 (paired, two-tailed Student's *t*-test), calculated between untreated and treated samples that were likewise stimulated. **c–e**, Flow cytometry analyses of T$_H$ polarization by staining for intracellular IL-17 and IFN-γ expression (FI) in restimulated T cells, outgrown in vitro by co-culture of memory CD4$^+$ T cells with iDC or DCs primed for 48 h with curdlan or *C. albicans*, in the presence of neutralizing IL-27 antibodies or rhIL-27 (both during DC-T co-culture), at day 11–14 (IL-27 antibody *n* = 7, rhIL-27 *n* = 4). In **c**, representative dot plots for independent donors are shown, with percentage positive cells indicated in each quadrant. In **d**, the percentage IL-17$^+$ cells per total amount of T cells are shown and in **e**, the percentage IFN-γ$^-$ and percentage IFN-γ$^+$ cells per IL-17$^+$ T cells are shown. Data in **d,e** represent mean ± s.d. of independent donors.

1.5 ± 0.8%), but not IL-12 (6.9 ± 1.8% and 6.9 ± 2.2%) neutralizing antibodies (Extended Data Fig. 4), explaining the lack of T$_H$17 induction by immature DCs, despite SEB-mediated T cell activation. Overall differentiation of IL-17$^+$ cells induced by curdlan- and *C. albicans*-primed DCs (12 ± 4.5% and 11.9 ± 4.2% of total cells, respectively) was reduced by supplementary rhIFN-β (5.1 ± 2.4% and 4.2 ± 1.6%), whereas blocking IFN-α/βR antibodies had no effect (14 ± 6.2% and 13 ± 5.6%) (Fig. 2e–h); however, the frequency of IFN-γ$^+$IL-17$^+$ T cells induced by curdlan- and *C. albicans*-primed DCs (17 ± 4.2% and 20 ± 10.4% of IL-17$^+$ cells, respectively) significantly increased by the addition of blocking IFN-α/βR antibodies (43 ± 4.1% and 45 ± 4.8%) or supplementary rhIFN-β (41 ± 5.0% and 41 ± 4.8%) (Fig. 2e–h), suggesting that type I IFN strongly influenced the phenotype of differentiated T$_H$17 cells.

Next, we compiled a panel of molecular markers to discriminate between T$_H$17 cell subtypes through real-time qPCR analyses, including transcription factors c-Maf (encoded by *MAF*), which regulates the expression of anti-inflammatory cytokine IL-10 and effector molecule prostaglandin D synthase (encoded by *PTGDS*)[13,25], and T-bet (encoded by *TBX21*); IL-1R; inflammatory cytokines IFN-γ, IL-22 and GM-CSF (encoded by *CSF2*); and the effector molecules CXCL3, CCL3 and CCL5, which are expressed by mouse pathogenic T$_H$17 cells[13,17,26–28]. The expression of these molecular markers was analyzed in peripheral

blood IL-17$^+$CD4$^+$ T cells isolated from healthy individuals, to provide a reference point for non-pathogenic T$_H$17 cells, and patients with acute untreated Crohn's disease (CD), in whom pathogenic T$_H$17 cells are the main mediators of gastrointestinal tract inflammation[29]. Expression of *TBX21*, *IFNG*, *IL1R1* and *CXCL3* expression was high in IL-17$^+$CD4$^+$ T cells from patients with acute untreated CD and low to moderate in IL-17$^+$CD4$^+$ T cells from healthy individuals (Fig. 2i and Extended Data Fig. 5). Expression of *CSF2*, *IL22* and *CCL3* was consistently higher for CD patients, whereas expression of *CCL5* was similar in IL-17$^+$CD4$^+$ T cells compared to healthy controls (Fig. 2i and Extended Data Fig. 5). Expression of *IL10*, *MAF* and *PTGDS* was high in IL-17$^+$CD4$^+$ T cells from healthy donors and minimal or absent in patients with CD (Fig. 2i,j and Extended Data Fig. 5). Of note, expression of *TBX21*, *IFNG*, *IL1R1* and *CXCL3* was much less pronounced, whereas expression of *IL10*, *MAF* and *PTGDS* was clearly detected in IL-17$^+$CD4$^+$ T cells from treated patients with CD (Extended Data Fig. 5). Expression of *IL17A*, *IL17F* and *RORC* (which encodes the transcription factor RORγt) was similar in IL-17$^+$CD4$^+$ T cells from healthy and all CD donors (Fig. 2i,j and Extended Data Fig. 5). These data suggested that expression of *IFNG*, *TBX21*, *IL1R1* and *CXCL3* defined a pathogenic signature, while expression of *IL10*, *MAF* and *PTGDS* defined non-pathogenic human T$_H$17 cells. IFN-γ$^+$IL-17$^+$ T cells purified from in vitro co-cultures of curdlan- or

*C. albicans*-primed DCs with either naive CD45RA⁺CD4⁺ or memory CD45RO⁺CD4⁺ T cells (Extended Data Fig. 3) showed high expression of the pathogenic signature genes *IFNG*, *TBX21*, *IL1R1* and *CXCL3*, which were almost completely absent in IFN-γ⁻IL-17⁺ T cells purified from the same co-cultures (Fig. 2j and Extended Data Fig. 5). In contrast, *IL10*, *MAF* and *PTGDS* were highly expressed in IFN-γ⁻IL-17⁺ T cells and almost absent in IFN-γ⁺IL-17⁺ T cells (Fig. 2j and Extended Data Fig. 5). *CSF2*, *IL22* and *CCL3* were detected in both subtypes, but significantly higher in IFN-γ⁺IL-17⁺ T cells (Fig. 2j and Extended Data Fig. 5). The signatures were similar across both naive and memory T cells co-cultures (Fig. 2j and Extended Data Fig. 5).

Curdlan- and *C. albicans*-primed DCs induced IL-17⁺ T cells with a predominantly non-pathogenic signature characterized by high *IL10*, *MAF* and *PTGDS* expression and low, but not absent, expression of *IFNG*, *TBX21*, *IL1R1* and *CXCL3* mRNA (Fig. 2k and Extended Data Fig. 6) and protein (Extended Data Fig. 7). Addition of blocking IFN-α/βR antibodies or rhIFN-β to curdlan- or *C. albicans*-primed DCs induced IL-17⁺ T cells with mixed molecular signatures (Fig. 2k and Extended Data Fig. 6). Thus, both loss of or high IFN-β promoted the polarization of naive and memory T cells into IFN-γ⁺IL-17⁺ T cells by dectin-1-stimulated DCs.

## Dectin-1 controls *IFNB* transcription through IRF1 and IRF5

*IFNB* transcription is mainly regulated through different transcription factors from the IRF family that bind the IFN-stimulated response element (ISRE) within its promoter[30]. IRF5 is involved in dectin-1-mediated IFN-β expression[22], while dectin-1 is known to activate IRF1[31]. Curdlan and *C. albicans* stimulation of DCs led to the nuclear translocation of IRF1 and IRF5 but not IRF3 at 2 h post-stimulation (Fig. 3a–c). IRF1, but not IRF5, was detected in the nucleus at 6 h post-stimulation (Fig. 3a–c). IRF7 expression was induced by curdlan or *C. albicans* stimulation and also detected in the nucleus at 6 h post-stimulation (Fig. 3a–c). Blocking antibodies to dectin-1 abrogated IRF1 and IRF5 nuclear translocation (Fig. 3d). Chromatin immunoprecipitation (ChIP) indicated that IRF1 and IRF5 were bound to the *IFNB* ISRE within 2 h after curdlan stimulation, whereas only IRF1 was detected 6 h post-stimulation (Fig. 3e). IRF7 did not bind *IFNB* ISRE, despite being present in the nucleus at 6 h post-stimulation (Fig. 3e). IRF5 silencing in DCs completely abrogated curdlan- or *C. albicans*-induced expression of *IFNB* mRNA and protein (Fig. 3f,g) as well as the transcription of ISGs *MXA* and *TRIM22* (Fig. 3h), indicating that IRF5 was critically required for dectin-1-mediated induction of *IFNB*. In contrast, IRF1 silencing enhanced both *IFNB* mRNA and protein and ISG mRNA after curdlan or *C. albicans* stimulation (Fig. 3f–h), suggesting that IRF1 has a negative regulatory role in dectin-1-mediated *IFNB* transcription. IRF1 silencing furthermore resulted in a secondary *IFNB* mRNA peak (Fig. 3f). ChIP analyses in IRF1-silenced DCs indicated that the *IFNB* ISRE was exclusively bound by IRF5 at 2 h post-stimulation, whereas IRF7 bound the *IFNB* ISRE at 6 h post-stimulation (Fig. 3i), at the time of the secondary *IFNB*

transcription peak (Fig. 3f). These data indicated that IRF1 acted as a negative regulator of dectin-1-induced *IFNB* transcription by blocking optimal binding of IRF5 and IRF7 to the *IFNB* promoter.

## IFN-β directs the polarization of non-pathogenic T_H17 cells through TGF-β

We next focused on how dectin-1-induced IFN-β modulated the polarization of T_H17 cells. Blocking IFN-α/βR antibodies did not significantly affect the secretion of IL-1β, IL-23, IL-6 and IL-12 by curdlan- or *C. albicans*-stimulated DCs (Fig. 4a), but blocked IL-27 secretion, while supplementary rhIFN-β increased IL-27 expression (Fig. 4b). IL-27 is a known negative regulator of T_H17 differentiation[32]; however, neutralizing IL-27 antibodies did not affect the numbers or phenotype of T_H17 cells induced by curdlan- or *C. albicans*-primed DCs (Fig. 4c–e), suggesting that dectin-1-induced IL-27 secretion by DCs was too low to affect IL-17 production. The development of IL-17⁺ cells induced by curdlan- and *C. albicans*-primed DCs (8.7 ± 1.8% and 8.2 ± 1.0% of total cells, respectively) was reduced by supplementary rhIL-27 (4.6 ± 1.3% and 5.2 ± 0.8%), but did not affect the phenotype of the induced T_H17 cells (Fig. 4c–e and Extended Data Fig. 6), suggesting that dectin-1-induced IL-27 did not affect T_H17 cell development.

The role of TGF-β in T_H17 development continues to be a topic of debate[33]. TGF-β activation requires disengagement of its active form from the large latent complex (LLC) that is sequestered within the extracellular matrix after secretion. Curdlan or *C. albicans* stimulation of DCs did not affect the total amount of TGF-β, but resulted in release of active TGF-β, which was inhibited by blocking antibodies against dectin-1 (Fig. 5a,b). Blocking IFN-α/βR antibodies completely abrogated TGF-β activation by curdlan- or *C. albicans*-stimulated DCs (Fig. 5b). Supplementary rhIFN-β blocked TGF-β activation by curdlan- or *C. albicans*-stimulated DCs in a concentration-dependent manner (Fig. 5c). Furthermore, enhancing or blocking the expression of IFN-β by silencing of IRF1 or IRF5 in DCs, respectively, completely blocked TGF-β activation by curdlan- or *C. albicans*-stimulated DCs (Fig. 5d). Thus, tightly regulated and intermediate IFN-β responses were required for the release of active TGF-β by DCs.

We next investigated the effect of TGF-β on T_H17 cell polarization. Overall differentiation of IL-17⁺ cells induced by curdlan- and *C. albicans*-primed DCs (13 ± 4.3% and 14 ± 3.5% of total cells, respectively) was not affected by the addition of blocking TGF-βR antibodies during co-culture with memory CD45RO⁺CD4⁺ T cells (18 ± 4.2% and 18 ± 2.6%); however, the frequency of IFN-γ⁺IL-17⁺ T cells (18 ± 3.7% and 22 ± 12% of IL-17⁺ cells, respectively) strongly increased by blocking TGF-βR antibodies (43 ± 3.0% and 44 ± 3.1%) (Fig. 5e–h and Extended Data Fig. 6). Supplementary rhTGF-β1 did not affect the number (13 ± 1.8% and 14 ± 3.8% of total cells, respectively) or phenotype (18 ± 4.2% and 18 ± 2.6% IFN-γ⁺ cells of IL-17⁺ cells) of T_H17 cells induced by curdlan- or *C. albicans*-primed DCs, but reduced the increased development of IFN-γ⁺IL-17⁺ T cells after

**Fig. 5 | Dectin-1 induces release of active TGF-β for non-pathogenic T_H17 polarization via tightly regulated IFN-β expression. a**, ELISA for quantification of total TGF-β in the supernatant of unstimulated DCs or after stimulation of DCs with curdlan or *C. albicans* at 24 h (*n* = 4). **b–d**, Secreted alkaline phosphatase (SEAP) assay on supernatant of HEK-Blue TGF-β reporter cells for quantification of active TGF-β in the supernatant of DCs after stimulation with curdlan or *C. albicans*, in the presence of blocking dectin-1 or IFN-α/βR antibodies (**b**), a concentration range of rhIFN-β (**c**) or after transduction with non-targeting (control) or specific siRNAs to silence IRF1 or IRF5 expression (**d**), at 4 h (**b**, *n* = 3), 8 h (**b**, *n* = 3) or 24 h (**b**, untreated *n* = 6, dectin-1 antibody, IFN-α/βR antibody *n* = 3; **c,d**, *n* = 4). Data in **a–d** represent mean ± s.d. of independent donors. **\****P* < 0.01, **\****P* < 0.05 (paired, two-tailed Student's *t*-test), calculated between untreated and treated (**b,c**) or control and specific siRNA-transduced (**d**) samples that were likewise stimulated. **e–g**, Flow cytometry analyses of T_H polarization by staining for intracellular IL-17 and IFN-γ expression (FI) in restimulated T cells, outgrown in vitro by co-culture of memory CD4⁺ T cells with iDCs or DCs primed for 48 h with

curdlan or *C. albicans*, in the presence of blocking IFN-α/βR antibodies (during DC stimulation) and/or blocking TGF-βR antibodies or rhTGF-β1 (both during DC-T co-culture), at day 11–14 (TGF-βR antibody, IFN-α/βR antibody + rhTGF-β1, *n* = 4; rhTGF-β1, *n* = 6, IFN-α/βR antibody, *n* = 8). In **e**, representative dot plots for independent donors are shown, with percentage positive cells indicated in each quadrant. In **e**, the percentage IL-17⁺ cells per total amount of T cells are shown and in **g**, the percentage IFN-γ⁻ and percentage IFN-γ⁺ cells per IL-17⁺ T cells are shown. Data in **f,g** represent mean ± s.d. of independent donors. **\****P* < 0.01 (paired, two-tailed Student's *t*-test), calculated between untreated and treated samples that were likewise stimulated. **h**, Real-time PCR analyses of normalized mRNA levels for indicated genes in IL-17⁺ T cells, isolated by flow cytometry-based sorting after T_H polarization by co-culture of memory CD4⁺ T cells with primed DCs, as described for **e–g**, at day 11–14 (*n* = 4). Results from real-time PCR were quantified using standard curves for all genes and normalized to the expression of reference household gene *ACTB*. Data represent mean (number within square) of independent donors: more detailed data are presented in Extended Data Fig. 6. Color legends are as described in Fig. 2.

blocking IFN-α/βR signaling (43 ± 4.1% and 45 ± 4.8% of IL-17⁺ cells) to the frequencies observed with untreated curdlan- or *C. albicans*-primed DCs (20 ± 3.0% and 16 ± 1.4% IFN-γ⁺ cells of IL-17⁺ cells) (Fig. 5e–h and Extended Data Fig. 6). Similarly, the frequency of IFN-γ⁺IL-17⁺ T cells induced by curdlan- and *C. albicans*-primed DCs during co-culture with naive CD45RA⁺CD4⁺ T cells (18 ± 6.5% and 18 ± 2.2% of IL-17⁺ cells, respectively) increased in the presence of blocking TGF-βR antibodies (63 ± 1.4% and 68 ± 5.1%), whereas supplementary rhTGF-β1 reduced the increase in frequency of IFN-γ⁺IL-17⁺ T cells after blocking IFN-α/βR signaling

(65 ± 5.5% and 67 ± 4.0%) to the frequencies observed with untreated curdlan- or *C. albicans*-primed DCs (14 ± 7.4% and 15 ± 7.7% IFN-γ⁺ cells of IL-17⁺ cells) (Extended Data Fig. 8). Notably, predominantly IFN-γ⁺IL-17⁺ T cells (77 ± 4.4% and 73 ± 11% of IL-17⁺ cells, respectively) developed when naive CD45RA⁺CD4⁺ T cells were co-cultured with DCs that were primed with curdlan or *C. albicans* for only 3 h, a time point where we could not yet detect release of active TGF-β (Fig. 5b), whereas mainly IFN-γ⁻IL-17⁺ T cells (81 ± 3.3% and 78 ± 3.5% of IL-17⁺ cells, respectively) developed when rhTGF-β1 was added to these co-cultures (Extended Data Fig. 8).

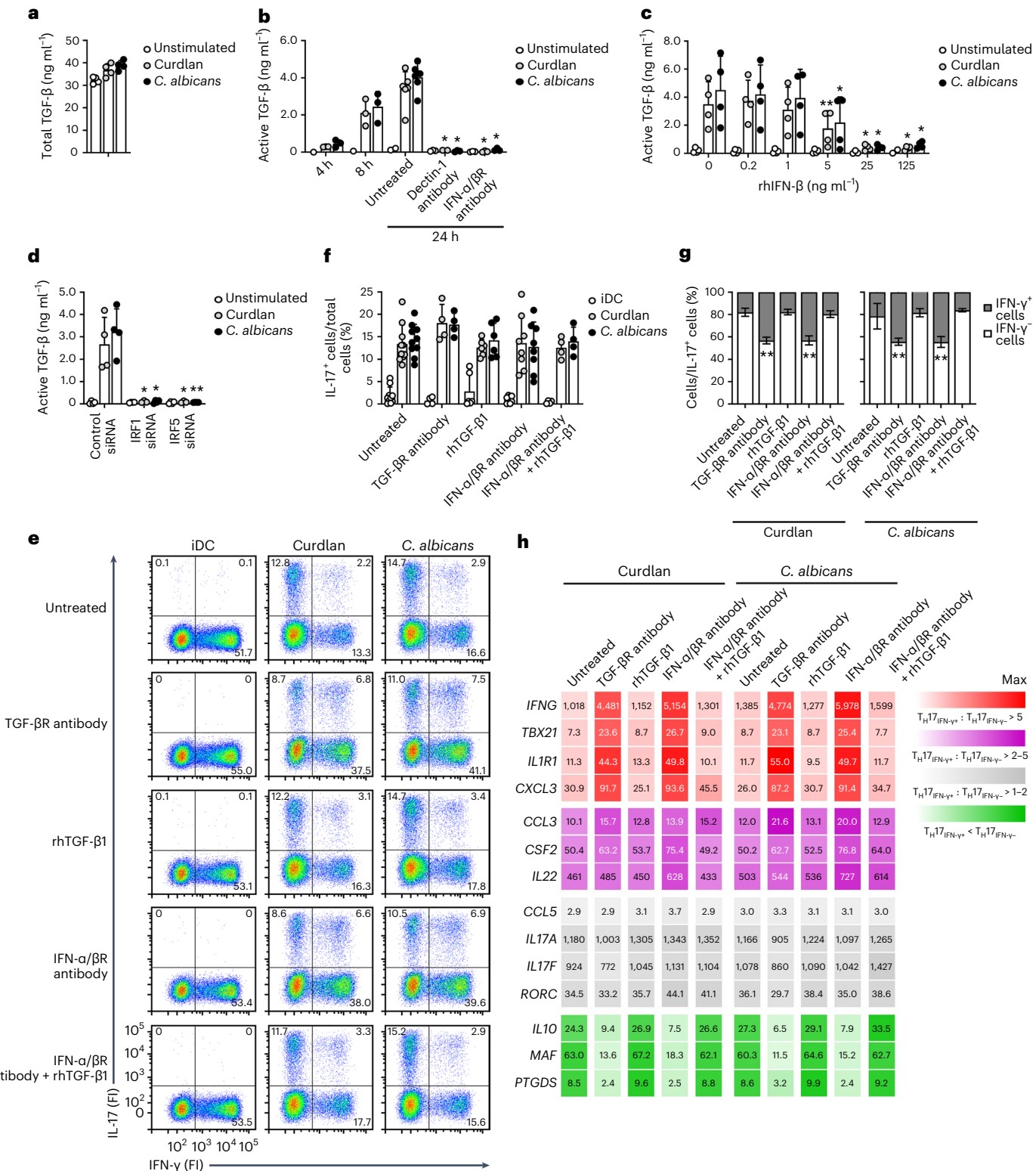

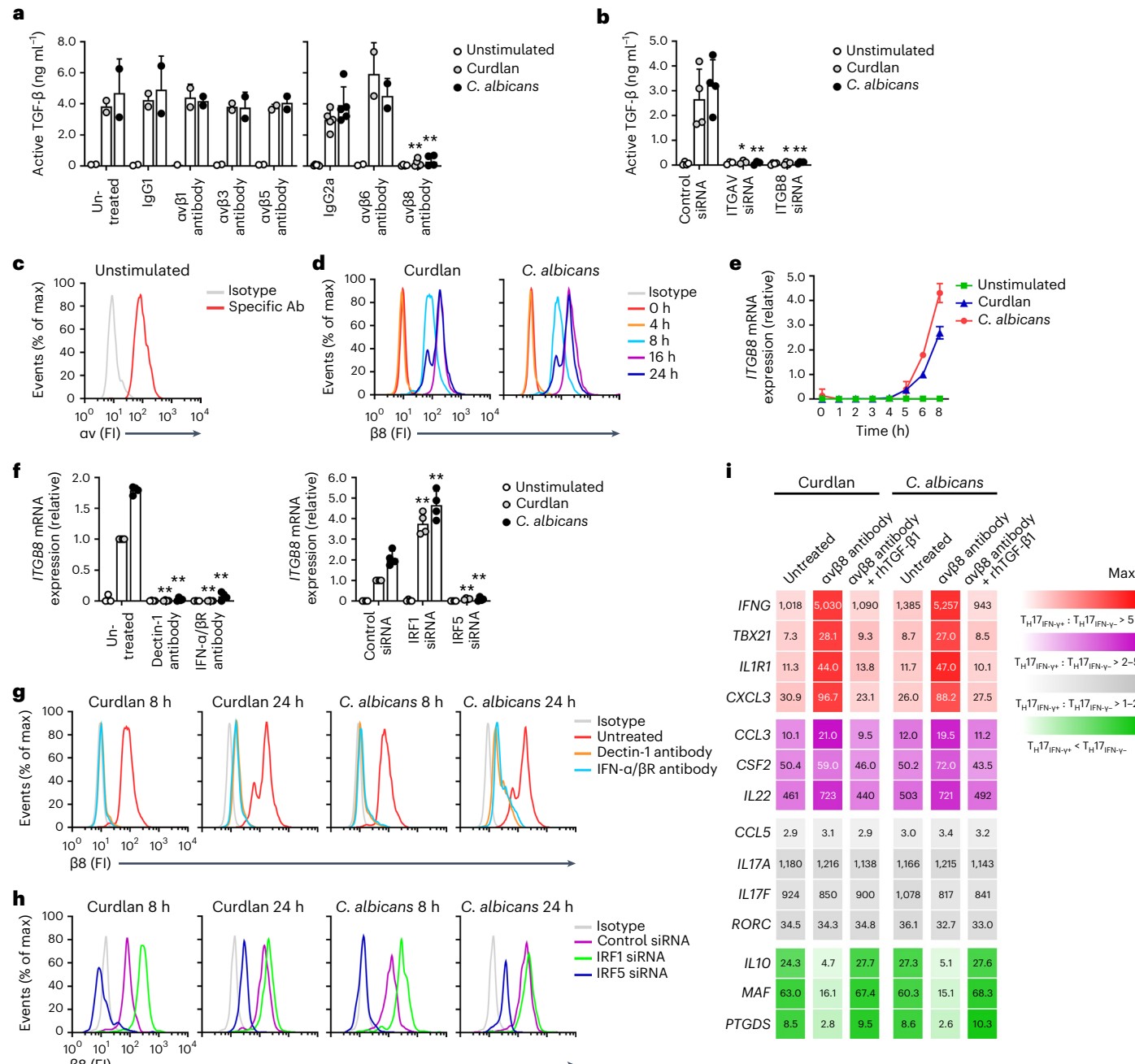

**Fig. 6 | Dectin-1 induces αvβ8 expression via IFN-β, which is critical for TGF-β activation and non-pathogenic T_H17 polarization. a,b**, SEAP assay on supernatant of HEK-Blue TGF-β reporter cells for quantification of active TGF-β in the supernatant of unstimulated DCs or after stimulation of DCs with curdlan or *C. albicans*, in the presence of blocking αvβ1, αvβ3, αvβ5, αvβ6 and αvβ8 antibodies or isotype (IgG1 and IgG2a) control antibodies (**a**) or after transduction with non-targeting (control) or specific siRNAs to silence αv (ITGAV) or β8 (ITGB8) expression (**b**) at 24 h (**a**, IgG1, αvβ1, αvβ3, αvβ5, αvβ6 Ab *n* = 2, IgG2a, αvβ8 Ab *n* = 5; **b**, *n* = 4). Data in **a,b** represent mean ± s.d. of independent donors. \*\**P* < 0.01, \**P* < 0.05 (paired, two-tailed Student's *t*-test), calculated between untreated and treated (**a**) or control and specific siRNA-transduced (**b**) samples that were likewise stimulated. **c,d,g,h**, Flow cytometry analyses by staining for αv (**c**) or β8 (**d,g,h**) expression (FI) in unstimulated DCs or after stimulation of DCs with curdlan or *C. albicans* (**d**), in the presence of blocking dectin-1 or IFN-α/ βR antibodies (**g**) or after silencing of IRF1 or IRF5 expression (**h**), at indicated times (**c**, *n* = 2; **d,g**, *n* = 3; **h**, *n* = 4). Isotype indicates negative control staining. Representative histograms for independent donors are shown. **e,f**, Real-time PCR analyses of *ITGB8* relative mRNA levels in unstimulated DCs or after stimulation of

DCs with curdlan or *C. albicans* (**e**), in the presence of blocking dectin-1 or IFN-α/ βR antibodies (**f**) or after silencing of IRF1 or IRF5 expression (**f**), at indicated times (**e**, curdlan *n* = 8, *C. albicans* *n* = 4) or 6 h (**f**, *n* = 4). Results from real-time PCR were normalized to the expression of reference household gene *GAPDH* and shown relative to 6 h curdlan stimulation. Data in **e,f** represent mean ± s.d. of independent donors. \*\**P* < 0.01 (paired, two-tailed Student's *t*-test), calculated between untreated and treated or control and specific siRNA-transduced (**f**) samples that were likewise stimulated. **i**, Real-time PCR analyses of normalized mRNA levels for indicated genes in IL-17⁺ T cells, isolated by flow cytometry-based sorting after T_H polarization by co-culture of memory CD4⁺ T cells with DCs primed for 48 h with curdlan or *C. albicans*, in the presence of blocking αvβ8 antibodies (during DC stimulation) and/or rhTGF-β1 (during DC-T co-culture), at day 11–14 (*n* = 3). Results from real-time PCR were quantified using standard curves for all genes and normalized to the expression of reference household gene *ACTB*. Data represent mean (number within square) of independent donors: more detailed data are presented in Extended Data Fig. 6. Color legends are as described in Fig. 2.

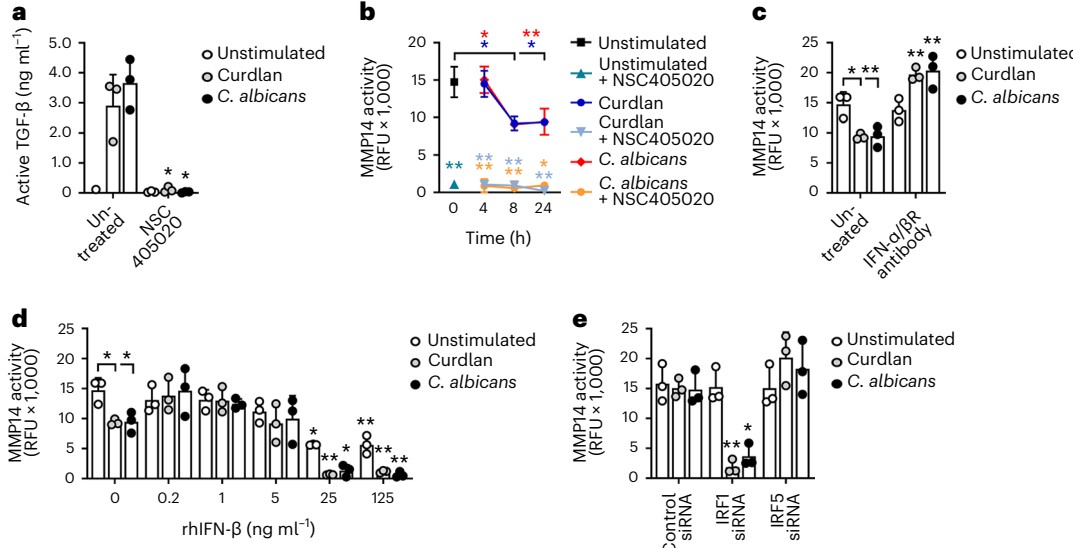

**Fig. 7 | Dectin-1-induced release of active TGF-β requires MMP14 activity.**
**a**, SEAP assay on supernatant of HEK-Blue TGF-β reporter cells for quantification of active TGF-β in the supernatant of unstimulated DCs or after stimulation of DCs with curdlan or *C. albicans*, in the presence of MMP14 inhibitor NSC405020 at 24 h (*n* = 3). **b–e**, FRET assay of extracellular MMP14 activity (RFU) in unstimulated DCs or after stimulation of DCs with curdlan or *C. albicans*, in the presence of NSC405020 (**b**), blocking IFN-α/βR antibodies (**c**), a concentration range of rhIFN-β (**d**) or after transduction with non-targeting (control) or specific siRNAs to silence IRF1 or IRF5 expression (**e**) at indicated times (**b**, *n* = 3) or 24 h (**c–e**, *n* = 3). RFU, relative fluorescence units. Data represent mean ± s.d. of independent donors. \*\**P* < 0.01, \**P* < 0.05 (paired, two-tailed Student's *t*-test), calculated between untreated and treated (**a–d**) or control and specific siRNA-transduced (**e**) samples that were likewise stimulated, unless otherwise indicated by brackets (**b–d**).

These data suggested that TGF-β activation limited the development of pathogenic T$_H$17 cells downstream of dectin-1-induced IFN-α/βR signaling in DCs.

**IFN-β-induced αvβ8 expression is critical for TGF-β activation**
Release of active TGF-β from the LLC requires conformational changes in LAP, a component of the LLC, through either traction force or proteolysis, which is applied when LAP is bound by αv-containing integrins[34]. Blocking antibodies to the DC-expressed heterodimeric integrins αvβ1, αvβ3, αvβ5, αvβ6 and αvβ8 showed that only blocking of αvβ8 interfered with release of active TGF-β by curdlan- or *C. albicans*-stimulated DCs (Fig. 6a), whereas silencing of αv or β8 expression in DCs completely blocked curdlan- or *C. albicans*-stimulated TGF-β activation (Fig. 6b). Immature DCs constitutively expressed αv but not β8 integrin (Fig. 6c,d). *ITGB8* transcription was induced in DCs within 5 h post-stimulation with curdlan or *C. albicans* (Fig. 6e), whereas β8 protein was detected after 8 h, reached maximum expression after 16 h and was still expressed at 24 h post-stimulation (Fig. 6d). Blocking antibodies to dectin-1 and IFN-α/βR completely inhibited *ITGB8* mRNA (Fig. 6f) and protein expression (Fig. 6g). Similarly, inhibition of IFN-β expression through silencing of IRF5 abrogated the induction of β8 (Fig. 6f,h), whereas increasing IFN-β expression through IRF1 silencing significantly boosted *ITGB8* transcription (Fig. 6f) and quickened and increased β8 expression (Fig. 6h). The addition of blocking antibodies against αvβ8 during co-culture with memory CD45RO⁺CD4⁺ T cells resulted in the development of both IL-10⁺IL-17⁺ and IFN-γ⁺IL-17⁺ T cells by curdlan- and *C. albicans*-primed DCs, while supplementary rhTGF-β1 restored the differentiation into predominantly IL-10⁺IL-17⁺ T cells (Fig. 6i and Extended Data Fig. 6). Thus, *ITGB8* was an ISG induced by dectin-1 signaling and the expression of αvβ8 was required for non-pathogenic T$_H$17 differentiation through the release of active TGF-β by DCs.

**IFN-β-induced BST2 obstructs TGF-β activation through MMP14**
Cleavage of LAP after its capture by αvβ8 is dependent on the proteolytic activity of the membrane-anchored metalloproteinase MMP14, which colocalizes with LAP-bound αvβ8 (refs. [35,36]). Inhibition of MMP14 activity with the specific inhibitor NSC405020 abrogated the release of active TGF-β by curdlan- or *C. albicans*-stimulated DCs (Fig. 7a). Immature DCs showed constitutive MMP14 activity as measured by the cleavage of a fluorescent substrate, which decreased after stimulation with curdlan or *C. albicans* and was abrogated by MMP14 inhibition (Fig. 7b). Blocking antibodies against IFN-α/βR increased MMP14 activity in curdlan- or *C. albicans*-stimulated, but not immature DCs (Fig. 7c), whereas supplementary rhIFN-β blocked MMP14 activity in a concentration-dependent manner (Fig. 7d). Blocking or increasing expression of IFN-β through silencing of IRF5 or IRF1 in DCs, respectively, slightly enhanced or significantly blocked MMP14 activity in curdlan- or *C. albicans*-stimulated DCs (Fig. 7e). These results indicated that excess type I IFN responses negatively impacted MMP14 activity and thereby TGF-β activation.

The ISG BST2 (ref. [20]) is a transmembrane protein that negatively modulates MMP14 activity through interactions between their cytoplasmic domains[37]. *BST2* mRNA and protein expression were induced 4 h and 8 h post-stimulation, respectively, in curdlan- or *C. albicans*-stimulated DCs and these were completely abrogated by blocking antibodies to dectin-1 or IFN-α/βR or through IRF5 silencing (Fig. 8a–e). In contrast, IRF1 silencing in DCs increased *BST2* transcription and expression of BST2 protein up to 24 h post-stimulation (Fig. 8b,e). Curdlan- or *C. albicans*-induced MMP14 activity increased slightly after silencing of BST2 in DCs (Fig. 8f), without significantly affecting TGF-β activation (Fig. 8g). Notably, silencing the expression of BST2 in IRF1-silenced DCs restored baseline MMP14 activity (Fig. 8f), whereas active TGF-β release was not only restored, as compared to IRF1-silenced DCs, but increased compared to control-silenced DCs after curdlan or *C. albicans* stimulation (Fig. 8g). BST2 silencing did not affect *IFNB* expression after stimulation (Extended Data Fig. 9). These results indicated that only high amounts of BST2 inhibited MMP14 activity and TGF-β activation.

To examine how high amounts of BST2 expression affect the differentiation of T$_H$17 cells by primed DCs, we transiently transfected DCs with a plasmid that encodes both eGFP and BST2 and separated

them based on their eGFP expression; eGFP⁻ DCs only expressed BST2 after curdlan or *C. albicans* stimulation, whereas eGFP⁺ DCs contained high amounts of BST2 without stimulation (Fig. 8h). *IFNB* expression after curdlan or *C. albicans* stimulation in either eGFP⁻ or eGFP⁺ DCs was unaffected (Extended Data Fig. 9). MMP14 activity was almost completely inhibited in unstimulated eGFP⁺ but not eGFP⁻ DCs (Fig. 8i), whereas TGF-β activation was almost absent after curdlan or *C. albicans* stimulation by eGFP⁺ but not eGFP⁻ DCs (Fig. 8j). Curdlan-primed eGFP⁺ DCs induced both IL-10⁺IL-17⁺ T cells and IFN-γ⁺IL-17⁺ T cells, whereas curdlan-primed eGFP⁻ cells instructed predominantly IL-10⁺IL-17⁺ T cells (Fig. 8k and Extended Data Fig. 6). The addition of rhTGF-β1 to curdlan-primed eGFP⁺ cells in co-culture with memory T cells resulted in the induction of predominantly IL-10⁺IL-17⁺ T cells (Fig. 8k). These results indicated that high amounts of BST2 blocked MMP14-mediated release of active TGF-β and the polarization of non-pathogenic T$_H$17 cells after dectin-1 stimulation.

Thus, dectin-1 regulated type I IFN expression to allow the TGF-β-mediated development of non-pathogenic T$_H$17 responses during fungal infections in humans (Extended Data Fig. 10).

## Discussion

Here, we demonstrated that dectin-1 signaling in human DCs specifically instructed the differentiation of non-pathogenic IFN-γ⁻ T$_H$17 cells. Strict regulation of type I IFN responses downstream of dectin-1 during fungal infections was required for the development of non-pathogenic IFN-γ⁻ T$_H$17 cells by controlling the release of active TGF-β by DCs.

Dectin-1 signaling in human DCs induced the release of cytokines such as IL-12, IL-1β, IL-23 and IL-6, but also active TGF-β. This combination of cytokines predominantly instructed the development of T$_H$17 cells with high expression of *IL10*, *MAF* and *PTGDS* and low expression of *IFNG*, *TBX21*, *IL1R1* and *CXCL3*, which define a non-pathogenic molecular signature for human T$_H$17 cells[13,17–19,25–28]. Active TGF-β was critical for non-pathogenic T$_H$17 cell differentiation, likely by blocking the expression of T-bet expression and transcription of *IFNG*, while also stimulating the expression of c-Maf and *IL10* transcription[38,39]. While signaling through IL-27R in T cells induces IL-10 production by T cells[40], excess IL-27 along with high amounts of type I IFN, did not induce the differentiation of non-pathogenic T$_H$17 cells, probably because IFN-γ expression in T cells counteracted the effects of IL-27R signaling[41]. Our study corroborates previous reports that circulating *C. albicans*-specific T$_H$17 cells in the blood of healthy people do not co-produce IFN-γ[42].

In contrast, in vitro differentiated *C. albicans*-specific T cells were reported to co-produce IL-17 and IFN-γ[43]. This apparent discrepancy was in fact due to the absence of active TGF-β at 3 h post-stimulation when antigen-presenting cells were co-cultured with T cells.

Type I IFN responses induced by dectin-1 stimulation were indispensable for TGF-β activation, but also required strict regulation, as higher amounts of IFN-β obstructed the release of active TGF-β. Dectin-1-induced, IFN-α/βR-dependent expression of αvβ8 allowed processing of the latent TGF-β complex by MMP14, thereby creating an environment that favored non-pathogenic T$_H$17 cell differentiation. In line with these observations, mice with specific deletion of αvβ8 in DCs but not T cells suffer from inflammatory bowel disease and autoimmunity[44] diseases that are now largely linked to pathogenic T$_H$17 cells[11,12]. Dectin-1-induced expression of IFN-β also led to expression of BST2. While BST2 expression induced by dectin-1 stimulation was too low to affect TGF-β activation, increased expression of BST2 on activated DCs impaired the activity of MMP14 and hence TGF-β activation, leading to pathogenic T$_H$17 cell polarization. MMP14 requires dynamic homodimerization for its proteolytic activity[45] and as such MMP14 repression might only occur after BST2 associates with both cytoplasmic domains of MMP14; such a titration effect might explain why high but not normal amounts of BST2 inhibit MMP14-mediated release of active TGF-β. Our data suggest that targeting the intracellular domain of BST2 in IFN-mediated inflammatory and autoimmune disorders might attenuate disease by redirecting T$_H$17 cell responses toward a non-pathogenic phenotype.

The balanced expression of IFN-β that was key to non-pathogenic T$_H$17 cell polarization after dectin-1 stimulation requires highly precise regulation of the *IFNB* promoter[30]. Opposite actions of IRF5 and IRF1 on the *IFNB* ISRE ensured a pattern of transient and intermediate IFN-β expression, in which only IRF5 drove transcription. IRF1 is mostly known as a transcriptional activator of *IFNB* and other promoters[46]. It remains to be determined how dectin-1 signaling renders IRF1 transcriptionally inept and this could be used for therapeutic intervention in autoimmune diseases that are characterized by enhanced type I IFN responses. Loss of negative regulation of IRF5 activity causes hyperproduction of type I IFN and development of systemic lupus erythematosus in mouse models[47]. The fine-tuning of type I IFN responses after dectin-1 stimulation might serve several purposes in antifungal immunity such as ensuring robust T$_H$17 cell differentiation as expression of IL-1β and IL-23 remains unaffected by IFN-α/βR signaling, while preventing the

---

**Fig. 8 | Dectin-1 tightly controls IFN-β-induced BST2 expression that negatively affects MMP14-mediated release of active TGF-β to direct non-pathogenic T$_H$17 polarization. a,b**, Real-time PCR analyses of *BST2* relative mRNA levels in unstimulated DCs or after stimulation of DCs with curdlan or *C. albicans*, in the presence of blocking dectin-1 or IFN-α/βR antibodies (**b**) or after transduction with non-targeting (control) or specific siRNAs to silence IRF1 or IRF5 expression (**a**), at indicated times (**a**, curdlan *n* = 8, *C. albicans* *n* = 4) or 6 h (**b**, IRF1 siRNA *n* = 8, IRF5 siRNA *n* = 4). Results from real-time PCR were normalized to the expression of reference household gene *GAPDH* and shown relative to 6 h curdlan stimulation. Data in **a,b** represent mean ± s.d. of independent donors. **P < 0.01, *P < 0.05 (paired, two-tailed Student's *t*-test), calculated between untreated and treated (**b**) or control and specific siRNA-transduced (**b**) samples that were likewise stimulated. **c–e**, Flow cytometry analyses by staining for BST2 expression (FI) in unstimulated DCs or after stimulation of DCs with curdlan or *C. albicans* (**c**), in the presence of blocking dectin-1 or IFN-α/βR antibodies (**d**) or after silencing of IRF1 or IRF5 expression (**e**), at indicated times (**c,d**, *n* = 3; **e**, *n* = 4). Isotype indicates negative control staining. Representative histograms for independent donors are shown. **f**, FRET assay of extracellular MMP14 activity (RFU) in unstimulated DCs or after stimulation of DCs with curdlan or *C. albicans*, after silencing of BST2 and/or IRF1 expression at 24 h (*n* = 6). **g**, SEAP assay on supernatant of HEK-Blue TGF-β reporter cells for quantification of active TGF-β in the supernatant of unstimulated DCs or after stimulation of DCs with curdlan or *C. albicans* after silencing of BST2 and/or IRF1 expression for 24 h (*n* = 4). Data in **f,g** represent

mean ± s.d. of independent donors. **P < 0.01 (paired, two-tailed Student's *t*-test), calculated between control and specific siRNA-transduced samples that were likewise stimulated. **h**, Flow cytometry analyses of eGFP and BST2 expression (FI) in DCs that were transfected with mammalian expression plasmid pCG–BST2–IRES–eGFP, subsequently sorted based on eGFP expression (eGFP⁻ or eGFP⁺) 24 h later and left unstimulated or stimulated with curdlan or *C. albicans* at 24 h (*n* = 3). Representative histograms for independent donors are shown. **i**, FRET assay of extracellular MMP14 activity (RFU) in DCs with normal (eGFP⁻) or increased (eGFP⁺) BST2 expression, either left unstimulated or after stimulation with curdlan or *C. albicans* at 24 h (*n* = 3). **j**, SEAP assay on supernatant of HEK-Blue TGF-β reporter cells for quantification of active TGF-β in the supernatant of DCs with normal (eGFP⁻) or increased (eGFP⁺) BST2 expression, either left unstimulated or after stimulation with curdlan or *C. albicans* at 24 h (curdlan *n* = 6, *C. albicans* *n* = 5). Data in **i,j** represent mean ± s.d. of independent donors. **P < 0.01, *P < 0.05 (paired, two-tailed Student's *t*-test), calculated between eGFP⁻ and eGFP⁺ samples that were likewise stimulated. **k**, Real-time PCR analyses of normalized mRNA levels for indicated genes in IL-17⁺ T cells, isolated by flow cytometry-based sorting after T$_H$ polarization by co-culture of memory CD4⁺ T cells with eGFP⁻ and eGFP⁺ DCs primed for 48 h with curdlan, in the absence or presence of rhTGF-β1 (during DC–T cell co-culture), at day 11–14 (*n* = 3). Results from real-time PCR were quantified using standard curves for all genes and normalized to the expression of reference household gene *ACTB*. Data represent mean (number within square) of independent donors: more detailed data are presented in Extended Data Fig. 6. Color legends are as described in Fig. 2.

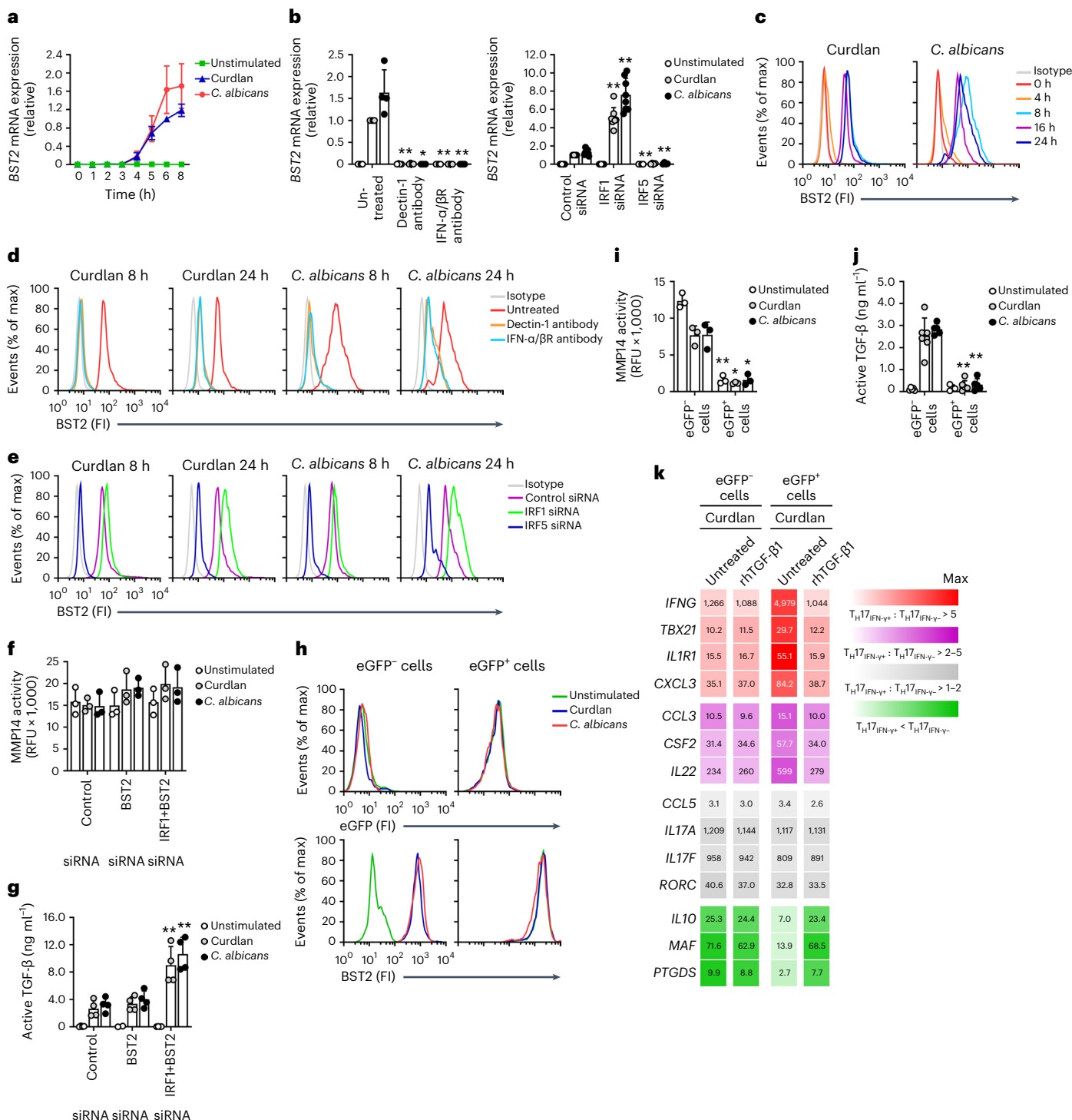

impairment of overall $T_H17$ cell development by limiting the production of IL-27 and promoting the development of non-pathogenic $T_H17$ cells by TGF-β activation.

Although dectin-1 is the main receptor for fungal detection on DCs, it is not the only PRR that senses fungal infections. Other CLRs, such as dectin-2, Mincle and DC-SIGN, as well as various TLRs, bind different fungal ligands[48]. How collaborations between these PRRs shape the environmental conditions in which $T_H17$ cell differentiation is primed during distinct fungal infections needs to be investigated as it might profoundly affect immunopathogenicity.

Deciphering how dectin-1 signaling finds a balance in type I IFN responses to orchestrate immunoprotective responses while restraining pathological responses during fungal infections will help

to understand inflammatory and autoimmune disorders and identify targets for therapeutical intervention.

## Online content

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

## Methods

### Ethics statement

This study was performed in accordance with the ethical guidelines of the Amsterdam UMC, location Academic Medical Center (AMC) and human material was obtained in accordance with the AMC Medical Ethics Review Committee (Institutional Review Committee) according to the Medical Research Involving Human Subjects Act. Buffy coats obtained after blood donation (Sanquin) are not subjected to informed consent according to the Medical Research Involving Human Subjects Act and the AMC Medical Ethics Review Committee. Blood obtained from healthy volunteers was covered by the BACON protocol. Patients with active CD were recruited at the IBD clinic of Amsterdam UMC, location AMC. After providing written informed consent, an additional blood sample was drawn in addition to routine blood draws. The project was covered by the Future-IBD Biobank protocol. All samples were handled anonymously.

### DC isolation and stimulation

Peripheral blood mononuclear cells were isolated from buffy coats of healthy volunteer blood donors (Sanquin) by a Lymphoprep (Axis-Shield) gradient step and monocytes were subsequently isolated by a Percoll (Amersham) gradient step. Monocytes were differentiated into immature DCs in the presence of 500 U ml$^{-1}$ IL-4 and 800 U ml$^{-1}$ GM-CSF (both Invitrogen) and used on day 6 or 7. Donors were routinely screened for dectin-1 single nucleotide polymorphism rs16910526 using TaqMan SNP Genotyping Assays (Assay ID C_33748481_10; Applied Biosystems); only dectin-1 wild-type DCs were used for experiments, unless otherwise indicated. DCs were stimulated with curdlan (10 μg ml$^{-1}$; Sigma), *Salmonella typhosa* LPS (10 ng ml$^{-1}$; Sigma), poly(I:C)-LyoVec (1 μg ml$^{-1}$; Invivogen), heat-killed *Candida* or *Aspergillus* species at multiplicity of infection (MOI) 10, rhIFN-β (0.2–125 ng ml$^{-1}$; Peprotech). DCs were preincubated for 2 h with MMP14 inhibitor NSC405020 (100 μM; Tocris) or blocking antibodies, anti-dectin-1 (20 μg ml$^{-1}$; clone 259931, MAB1859, R&D Systems), anti-IFN-α/βR2 (20 μg ml$^{-1}$; clone MMHAR-2, PBL Assay Science), anti-αvβ1 (10 μg ml$^{-1}$; clone P5D2, MAB17781, R&D Systems), anti-αvβ3 (10 μg ml$^{-1}$; clone 23C6, MAB3050, R&D Systems), anti-αvβ5 (10 μg ml$^{-1}$; clone P5H9, MAB2528, R&D Systems), anti-αvβ6 (10 μg ml$^{-1}$; clone 10D5, ab77906, Abcam), anti-αvβ8 (10 μg ml$^{-1}$; kind gift from S.L. Nishimura[49]) or isotype control antibodies, mouse IgG1 (20 μg ml$^{-1}$; clone MOPC-21, 555746, BD Pharmingen), mouse IgG2a (20 μg ml$^{-1}$; clone G155-178, 555571, BD Bioscience) and mouse IgG2b (20 μg ml$^{-1}$; clone 20116, MAB004, R&D Systems). DCs were transfected with 25 nM siRNA using transfection reagents DF4 (Dharmacon) according to the manufacturer's instructions and used for experiments 72 h after transfection. SMARTpool siRNAs used were Syk (M-003176-03), CARD9 (M-004400-01), Bcl10 (M-004381-02), Malt1 (M-005936-02), LSP1 (M-012640-00), Raf-1 (M-003601-02), IRF1 (M-011704-01), IRF5 (M-011706-00), IRF7 (M-011810-02), ITGAV (M-004565-03), ITGB8 (M-008014-02), BST2 (M-011817-00) and non-targeting siRNA (D-001206-13) as a control (all Dharmacon). Silencing of expression was verified by real-time PCR and flow cytometry (Extended Data Fig. 1 and refs. [7,50,51]); antibodies used for verification were anti-IRF1 (1:50 dilution; ab26109, Abcam), anti-IRF5 (1:50 dilution; ab124792, Abcam), anti-αv (1:50 dilution; AF1219, R&D Systems), anti-β8 (1:50 dilution; ab80673, Abcam), anti-BST2 (1:100 dilution; NIH AIDS Reagent Program 11721), followed by incubation with either PE-conjugated anti-rabbit (1:200 dilution; 711-116-152, Jackson Immunoresearch) or Alexa Fluor 488-conjugated anti-goat (1:400 dilution; A11055, Invitrogen). DCs were also transfected with a pCG–BST2–IRES–eGFP expression plasmid (a kind gift from F. Kirchhoff[52]) coupled to LyoVec (Invivogen), as described by the manufacturer. eGFP$^-$ and eGFP$^+$ cells were selected 24–40 h after transfection by sorting on a FACSAria IIu Cell Sorter (BD Biosciences) and used for experiments; BST2 (over)expression was determined in both cell subsets after staining with anti-BST2 and PE-conjugated anti-rabbit

(described above) by flow cytometry on a FACS Calibur or Canto (BD Biosciences). All flow cytometry data analysis was performed using FlowJo v.10.8.1 (BD Biosciences).

### Fungal strains

*Candida* strains were grown in Sabouraud dextrose broth and incubated at 25 °C for 3 d, while shaking. *A. fumigatus* was grown on potato dextrose agar at 37 °C for 3 d. Conidia were dislodged from slants by gentle tapping and then resuspended in PBS/0.1% Tween-80. Hyphal contamination was removed by straining the cell solution through a glass filter. Fungi were heat inactivated at 56 °C for 1 h.

### Cytokine and ISG expression

Transcript levels in DCs were quantified with the SYBR green method in an ABI 7500 Fast PCR detection system (Applied Biosystems), after mRNA isolation using the mRNA capture kit (Roche) and cDNA synthesis using the Reverse Transcription System (Promega). Specific primers were designed using Primer Express 2.0 (Applied Biosystems; Supplementary Table 1). The $C_t$ value is defined as the number of PCR cycles where the fluorescence signal exceeds the detection threshold value. The normalized amount of target mRNA was calculated from the obtained $C_t$ values for both target and *GAPDH* mRNA with $N_t = 2^{Ct(GAPDH) - Ct(target)}$. The relative mRNA expression in DCs was obtained by setting $N_t$ in either 2 h (*IFNB*) or 6 h (ISGs) curdlan-stimulated samples at 1 within one experiment and for each donor, except when donors were used that were homozygous for the minor alleles of rs16910526 where $N_t$ in 2 h LPS-stimulated samples was set at 1.

Secreted IL-1β, IL-12p70, IL-6, IL-23 and IL-27 levels in DC culture supernatants that were collected 28 h post-stimulation were determined by ELISA (Invitrogen). Intracellular IFN-β, αv, β8 and BST2 levels were determined by flow cytometry. Cells were first fixated in 3% paraformaldehyde for 10 min and permeabilized in 90% methanol at 4 °C for 30 min. Staining with anti-αv, anti-β8, anti-BST2, followed by PE-conjugated anti-rabbit was conducted as described above. IFN-β was stained directly with FITC-conjugated anti-IFN-β (1:20 dilution; clone MMHB-3, 21400-3, PBL Assay Science), while FITC-conjugated IgG1κ mouse antibody (1:250 dilution; clone P3.6.2.8.1, 11-4714-81, eBioscience) was used as isotype control. Expression was analyzed on a FACS Calibur or Canto (BD Biosciences). All flow cytometry data analyses were performed using FlowJo v.10.8.1 (BD Biosciences). Total TGF-β levels in DC culture supernatants that were collected 24 h post-stimulation were determined by ELISA with anti-pan TGF-β (2 μg ml$^{-1}$, coating antibody; clone 1D11, MAB1835, R&D Systems) and biotinylated anti-TGF-β1 (0.2 μg ml$^{-1}$, detecting antibody; BAF240, R&D Systems), followed by streptavidin-HRP (Invitrogen). Supernatants were treated with 0.2 N HCl before ELISA to expose epitopes obscured within the LLC. Bioactive TGF-β levels in DC culture supernatants were measured by transferring supernatant at 4, 8 or 24 h post-stimulation to HEK-Blue TGF-β reporter cells (Invivogen), after which secreted SEAP in the reporter cell culture supernatants was determined 24 h later using the SEAP Reporter Assay kit (Invivogen). HEK-Blue TGF-β reporter cells were also stimulated with a concentration range of rhTGF-β1 (R&D Systems) to generate a standard curve.

### IRF activation and DNA binding

Nuclear and cytoplasmic extracts of DCs were prepared 2 or 6 h post-stimulation using NucBuster protein extraction kit (Novagen) as described by the manufacturer. Then, 20 μg of extract was resolved by SDS−PAGE and cellular localization of IRF1, IRF3, IRF5 or IRF7 was determined by immunoblotting with anti-IRF1 (1:1,000 dilution; 8478, Cell Signaling), anti-IRF3 (1:1,000 dilution; sc-9082, Santa Cruz), anti-IRF5 (1:1,000 dilution; ab124792, Abcam) or anti-IRF7 (1:1,000 dilution; 4920, Cell Signaling), followed by incubation with HRP-conjugated Clean-Blot IP Detection Reagent (1:2,500 dilution; 21230, Pierce) and ECL detection (Pierce). Membranes were also probed with anti-β-actin

(1:2,000 dilution; clone ACTBD11B7, sc-81178, Santa Cruz), followed by HRP-conjugated anti-mouse (1:1,000 dilution; sc-2314, Santa Cruz) to ensure equal protein loading among cytoplasmic and nuclear extracts, respectively. Nuclear IRF1, IRF5 and IRF7 levels were further quantified by ELISA (USCN Life Science). IRF1 and IRF5 localization was also determined by immunofluorescence staining; DCs were fixated with 4% paraformaldehyde 2 h post-stimulation, permeabilized with 0.2% (v/v) Triton X-100 in PBS, stained with anti-IRF1 (1:100 dilution; ab26109, Abcam) or anti-IRF5 (1:100 dilution; ab2932, Abcam), followed by incubation with Alexa Fluor 546-conjugated anti-rabbit (1:400 dilution; A10040, Invitrogen) or Alexa Fluor 546-conjugated anti-goat (1:400 dilution; A21085, Invitrogen), respectively. After staining nuclei with DAPI (300 nM; Invitrogen), cells were preserved in ProLong Diamond Antifade Mountant (Molecular probes) and IRF localization was visualized with a TCS SP8 X confocal microscope (Leica).

Binding of IRF factors to the *IFNB* promoter was assessed by ChIP assays using the ChIP-IT Express Enzymatic Shearing and ChIP-IT Express HT kits (both from Active Motif). Briefly, cells were fixated with 1% (v/v) paraformaldehyde 2 or 6 h post-stimulation, nuclei were isolated and chromatin DNA fragmented by enzymatic shearing (10 min, 37 °C). Protein-DNA complexes were immunoprecipitated using 2 µg anti-IRF1 (sc-640X, Santa Cruz), anti-IRF5 (abcam2932, Abcam), anti-IRF7 (sc-9083X, Santa Cruz) or negative control IgG (sc-2025; Santa Cruz) and protein G-coated magnetic beads. DNA was purified after reversal of crosslinks. Quantitative real-time PCR using SYBR green (Applied Biosystems) as described above was performed with a primer set spanning the ISRE of the *IFNB* promoter (Supplementary Table 1). Negative Control Primer Set 1 (active motif) was used as a negative control. To normalize for DNA input, a sample for each condition was taken that had not undergone immunoprecipitation ('input DNA') and results wre expressed as percentage input DNA.

## MMP14 activity
MMP14 activity was measured by adding 4 µM of the MMP14-specific FRET peptide substrate MMP14 substrate I (Calbiochem) to DCs 4, 8 or 24 h post-stimulation for 30 min at 37 °C. Supernatant containing (cleaved) substrate was then transferred to a black 96-well plate and fluorescence intensity was measured at 320 nm excitation and 420 nm emission wavelength using a Synergy HT reader (Biotek); the detected relative fluorescence units are a measure for MMP14 activity.

## T$_H$17 cell isolation, in vitro outgrowth and characterization
CD4$^+$ T cells were isolated by negative selection from blood donated by healthy donors or Crohn's disease patients with the human CD4$^+$ T cell isolation kit II (Miltenyi) and then stimulated for 6 h with 100 ng ml$^{-1}$ PMA (Sigma) and 1 µg ml$^{-1}$ ionomycin (Sigma), the last 4 h in the presence of 10 µg ml$^{-1}$ brefeldin A (Sigma). After fixation in 4% paraformaldehyde for 10 min, permeabilization with 0.5% (v/v) saponin and staining with APC-conjugated anti-IL-17 (1:25 dilution; clone eBio64DEC17, 17-7179-42, eBioscience), IL-17$^+$ cells were selected by sorting on a FACSAria IIu Cell Sorter (BD Biosciences). After reversal of crosslinks by proteinase K (QIAGEN) treatment at 56 °C for 16 h in PKD buffer (QIAGEN), mRNA isolation and complementary DNA synthesis was performed as described above for DCs. Transcript levels in T cells were measured with quantitative real-time PCR using SYBR green as described above. Specific primers were designed using Primer Express 2.0 (Applied Biosystems; Supplementary Table 1). A standard curve was created for all target transcripts from known quantities of human TrueClone or TrueORF expression plasmids (all from Origene). The absolute amount of target mRNA was normalized based on the absolute amount of *ACTB* mRNA within each sample.

For in vitro T$_H$17 cell outgrowth assays, both naive and memory CD4$^+$ T cells were isolated from buffy coats of healthy blood donors

(Sanquin) with the human CD4$^+$ T cell isolation kit II (Miltenyi), combined with staining with PE-conjugated anti-CD45RO (200 µg ml$^{-1}$; clone UCHL1, R084301-2, Agilent), captured on anti-PE beads (Miltenyi). Isolated naive T cells were routinely >99% CD4$^+$ (stained with Alexa Fluor 488-conjugated anti-CD4, 1:50 dilution; clone RPA-T4, 300519, BioLegend) and contained <1% CD45RO$^+$ cells, whereas memory T cells were routinely >99% CD4$^+$ and contained <1% CD45RA$^+$ cells (stained with APC-conjugated anti-CD45RA, 1:50 dilution; clone HI100, 550855, BD). DCs were either silenced for indicated proteins or preincubated for 2 h with blocking antibodies and then activated for 48 h as indicated, as described above. When DCs were activated for only 3 h before co-culture, this is mentioned explicitly in the text. Primed DCs were co-cultured with either naive CD4$^+$ T cells (200,000 T cells to 200,000 DCs) for 5 d before restimulation or memory CD4$^+$ T cells (20,000 T cells to 5,000 DCs) in the presence of 10 pg ml$^{-1}$ SEB (Sigma) and 10 U ml$^{-1}$ IL-2 (Chiron) (added after 5 d) until they became quiescent, typically after 11–14 d, before restimulation. During co-culture, cells were treated with either rhIL-27 (300 ng ml$^{-1}$; R&D Systems), rhTGF-β1 (20 ng ml$^{-1}$; R&D Systems), neutralizing antibodies (5 µg ml$^{-1}$) to IL-1β (AF-201-NA, R&D Systems), IL-23 (AF1716, R&D Systems), IL-12 (AF-219-NA, R&D Systems), IL-27 (AF2526, R&D Systems) or TGF-βRII (AF-241-NA, R&D Systems) or normal goat IgG (AB-108-C; R&D Systems) as a control. Proliferation and survival of co-cultures was monitored by loading naive or memory T cells with 2 µM CFSE (Molecular Probes) and measured every other day in combination with Fixable Viability Dye eFluor 780 (1:833 dilution; 65-0865-14, eBioscience) staining by flow cytometry on a FACS Canto (BD Biosciences). All flow cytometry data analyses were performed using FlowJo v.10.8.1 (BD Biosciences). Additionally, cell counts were determined every other day using the CASY cell counter and analyzer (OMNI Life Science). Restimulation of T cells occurred for 6 h with PMA and ionomycin in the presence of brefeldin A as described above. After fixation, permeabilization and staining with either APC-conjugated anti-IL-17 or eFluor506-conjugated anti-IL-17 (1:50 dilution; clone eBio64DEC17, 69-7179-42, eBioscience) alone or in combination with FITC-conjugated anti-IFN-γ (1:5 dilution; clone 25723.11, 340449, BD), total IL-17$^+$ or separate IFN-γ$^-$IL-17$^+$ and IFN-γ$^+$IL-17$^+$ cells were measured by flow cytometry and subsequently selected by sorting and used for either mRNA isolation, cDNA synthesis and quantitative real-time PCR as described above. Gating strategies for analyses and sorting are shown in Extended Data Fig. 3. Sorted total IL-17$^+$ cells were also subjected to further flow cytometry analyses, after permeabilization using the Foxp3/transcription factor staining kit (eBioscience), by staining with (combinations of) the following antibodies: FITC-conjugated anti-IFN-γ, PE-conjugated anti-IL-10 (1:10 dilution; clone JES3-9D7, 12-7108-82, Invitrogen), Alexa Fluor 647-conjugated anti-RORγt (1:10 dilution; clone Q21-559, 563620, BD), APC-conjugated anti-T-bet (1:10 dilution; clone 4B10, 644814, BioLegend), eFluor660-conjugated anti-c-Maf (1:10 dilution; clone sym0F1, 50-9855-82, Invitrogen), APC-conjugated anti-IL-1R1 (1:10 dilution; FAB269A, R&D Systems) and APC-conjugated anti-GM-CSF (1:10 dilution; clone BVD2-21C11, 502310, BioLegend) or matching isotype control antibodies: Alexa Fluor 647-conjugated mouse IgG2b (1:5 dilution; clone 27–35, 558713, BD), APC-conjugated mouse IgG1 (10 µg ml$^{-1}$; clone P3.6.2.8.1, 17-4714-42, Invitrogen), eFluor660-conjugated mouse IgG2b (10 µg ml$^{-1}$; clone eBMG2b, 50-4732-82, Invitrogen), APC-conjugated goat IgG (1:5 dilution; IC108A, R&D Systems) and APC-conjugated rat IgG2a (2.5 µg ml$^{-1}$; 402305, BioLegend).

## Statistical analysis
Statistical analyses were performed using the Student's *t*-test for paired and two-tailed observations using Office Professional Plus 2019, Excel (Microsoft) and GraphPad Prism v.8.3.1 (Dotmatics). Statistical significance was set at *P* < 0.05.

**Reporting summary**

Further information on research design is available in the Nature Portfolio Reporting Summary linked to this article.

## Data availability

Source data are provided with this paper.

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

## Acknowledgements

We thank X. Yocarini and R.K. Ramkisoen (both Amsterdam University Medical Center) for providing blood from patients with CD. The pCG–BST2–IRES–eGFP expression plasmid was kindly provided by F. Kirchhoff (Ulm University Medical Center). β8 (B5) antibodies were a kind gift from S.L. Nishimura (University of California). This work was supported by the European Research Council (advanced grant 670424 to T.B.H.G.). The funders had no role in study design, data collection and analysis, decision to publish or preparation of the manuscript.

## Author contributions

S.I.G. designed and supervised the research, performed experiments and wrote the manuscript. T.M.K., A.D. and B.A.W. performed experiments. E.B.M.R. assisted with flow cytometry experiments. G.R.A.M. D'H. provided blood from patients with CD. B.T. and T.B. provided fungal preparations. T.B.H.G. supported the research, provided feedback and helped prepare the manuscript.

## Competing interests

The authors declare no competing interests.

## Additional information

**Extended data** is available for this paper at https://doi.org/10.1038/s41590-022-01348-2.

**Correspondence and requests for materials** should be addressed to Sonja I. Gringhuis or Teunis B. H. Geijtenbeek.

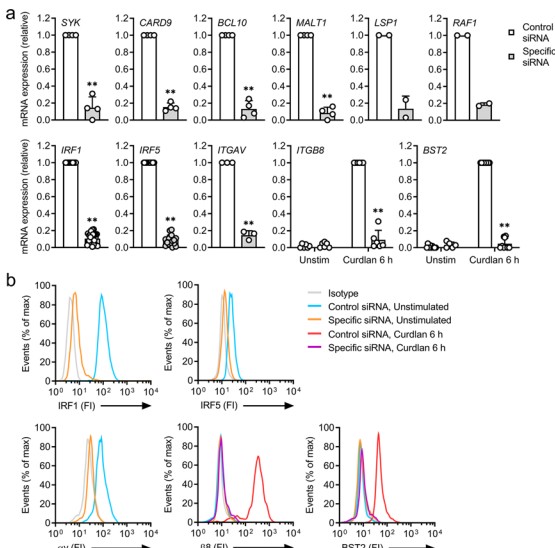

**Extended Data Fig. 1 | Silencing of proteins in human DCs by RNA interference. a**, Real-time PCR analyses of relative mRNA levels of indicated genes in unstimulated DCs or after 6 h stimulation of DCs with curdlan, after transduction with non-targeting (control) or specific siRNAs to silence protein expression (Syk, CARD9, Bcl10, Malt1 siRNA $n = 4$; LSP1, Raf-1 siRNA $n = 2$; IRF1 siRNA $n = 26$; IRF5 siRNA $n = 18$; ITGAV siRNA $n = 3$; ITGB8 siRNA $n = 6$; BST2 siRNA $n = 9$). Results from real-time PCR were normalized to the expression of reference household gene *GAPDH* and shown relative to either unstimulated or 6 h curdlan stimulation. Data represent mean ± s.d. of independent donors. **$P < 0.01$ (Student's *t*-test, paired, two-tailed), calculated between control and specific siRNA-transduced samples that were likewise stimulated. **b**, Flow cytometry analyses by staining for specific protein expression (FI, fluorescence intensity) in unstimulated DCs or after 6 h stimulation of DCs with curdlan, after silencing of specific protein expression ($n = 3$). Isotype control indicates negative control staining. Representative histograms for independent donors are shown.

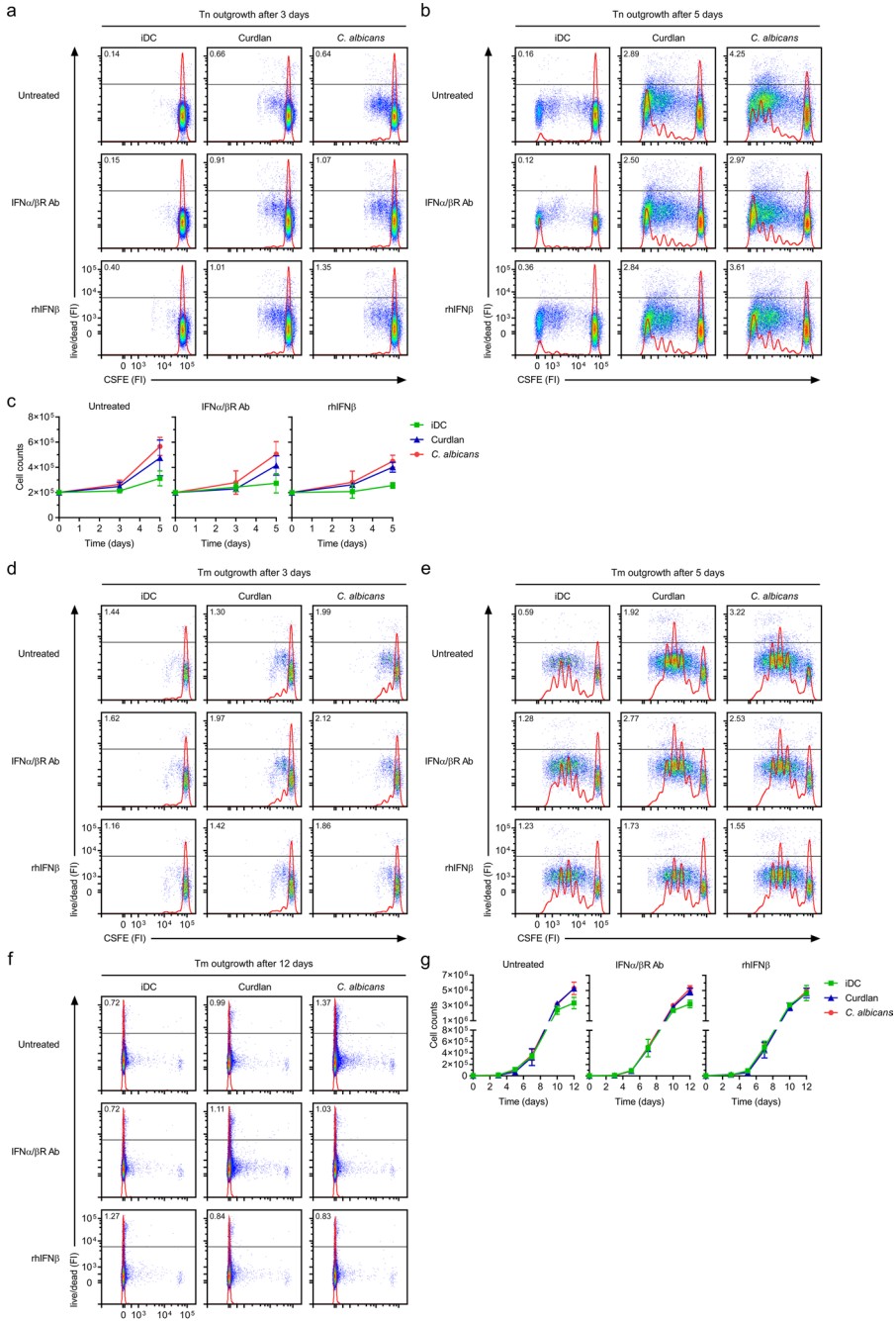

**Extended Data Fig. 2 | Proliferation and survival during co-culture of curdlan- or *C. albicans*-primed DCs with naive or memory CD4⁺ T cells are not altered by either attenuated or enhanced type I IFN responses. a,b,d-f**, Flow cytometry analyses of proliferation and survival by carboxyfluorescein succinimidyl ester (CFSE) dilution (FI, fluorescence intensity) and staining with a live/dead dye after T cells were outgrown in vitro by co-culture of either naive (Tn, **a,b**) or memory (Tm, **c-e**) CD4+ T cells with immature DCs (iDC) or DCs primed for 48 h with curdlan or *C. albicans*, in the presence of blocking IFN-α/βR antibodies (Ab) or recombinant human (rh) IFN-β, at day 3 (**a,d**, n = 3), day 5

(**b,e**, n = 3) or day 12 (**f**, n = 3). DC-memory T cell co-cultures (**c-e**) were done in the presence of bacterial superantigen *Staphylococcus aureus* enterotoxin B, resulting in antigen-independent T cell activation. Representative dot plots for independent donors are shown, with the percentage dead cells indicated in the upper part. The red histogram plots show the calculated CFSE division peaks. **c,g**, T cell counts at indicated time points after T cells were outgrown *in vitro* by co-culture of either naive (**c**) or memory (**g**) CD4⁺ T cells with immature DCs (iDC) or primed DCs, as described for **a,b,d-f** (n = 3). Data represent mean ± s.d. of independent donors.

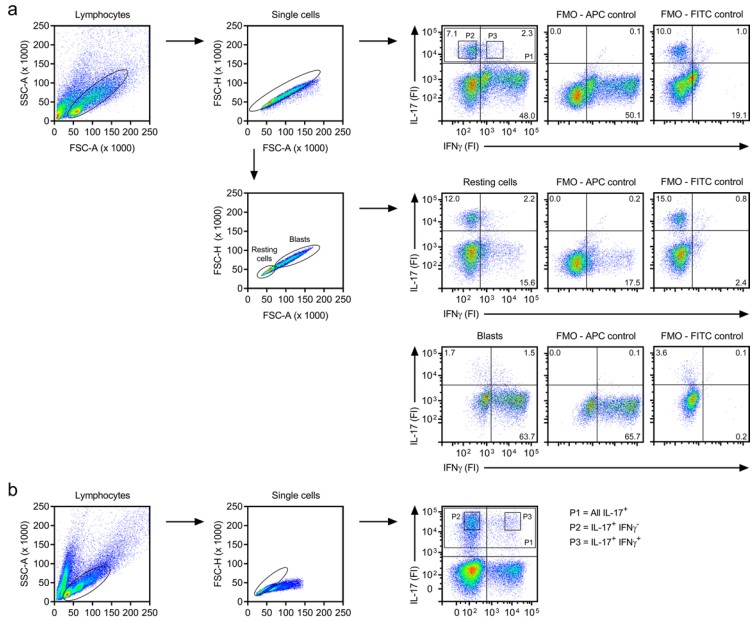

**Extended Data Fig. 3 | Flow cytometry analysis strategies. a, b,** Strategies for flow cytometry analyses of $T_H$ polarization by staining for intracellular IL-17 (APC) and/or IFN-γ (FITC) expression (FI, fluorescence intensity) in restimulated T cells, outgrown in vitro by co-culture of naive (**a**) or memory (**b**) CD4[+] T cells with immature or primed DCs: 1. Live lymphocytes were selected based on FSC-A/ SSC-A, followed by 2. single cell selection based on FSC-A/FSC-H. 3. During sorting, all IL-17[+], IFNγ[-]IL-17[+] or IFN-γ[+]IL-17[+] T cells were selected via respectively gate P1, P2 or P3. In **a**, restimulated T cells after DC-naive T cell co-cultures were further divided in resting and blast cell populations and analyzed separately

for IL-17 and IFN-γ expression. Fluorescence minus one (FMO) stainings were performed to determine gate borders. This was necessary as IFN-γ[-]IL-17[+] blast T cells coincided with IFN-γ[+]IL-17[+] resting T cells. These separate analyses were used to calculate the total percentages of IFN-γ[-]IL-17[+] and IFN-γ[+]IL-17[+] T cells present after restimulation, while accounting for the percentages of resting and blast cells. The percentage blast cells varied between 10-40% between experiments. In the IL-17/IFN-γ dot plots, the percentage positive cells are indicated in each quadrant.

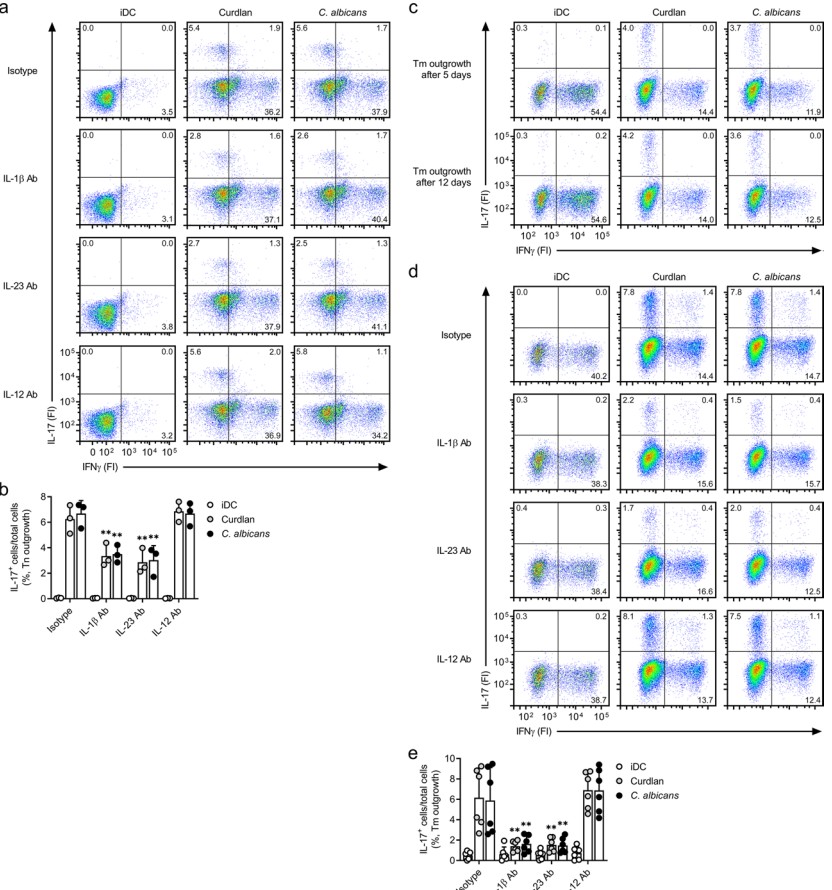

**Extended Data Fig. 4 | Dectin-1 directs T$_H$17 polarization via release of IL-1β and IL-23. a-e**, Flow cytometry analyses of T$_H$ polarization by staining for intracellular IL-17 and IFN-γ expression (FI, fluorescence intensity) in restimulated T cells, outgrown in vitro by co-culture of either naive (Tn, **a,b,**) or memory (Tm, **c-e,**) CD4$^+$ T cells with immature DCs (iDC) or DCs primed for 48 h with curdlan or *C. albicans*, in the presence of blocking IL-1β, IL-23 or IL-12 antibodies (Ab) or isotype IgG control antibodies, at day 5 (**a,b**, *n* = 3; **c-e**, *n* = 6) or

day 11–14 (**c-e**, *n* = 6). In **a,c,d**, representative dot plots for independent donors are shown, with the percentage positive cells indicated in each quadrant. In **b,e**, the percentage IL-17$^+$ cells per total amount of T cells are shown. Data in **b,e** represent mean ± s.d. of independent donors. **\**P* < 0.01 (Student's *t*-test, paired, two-tailed), calculated between untreated and treated samples that were likewise stimulated.

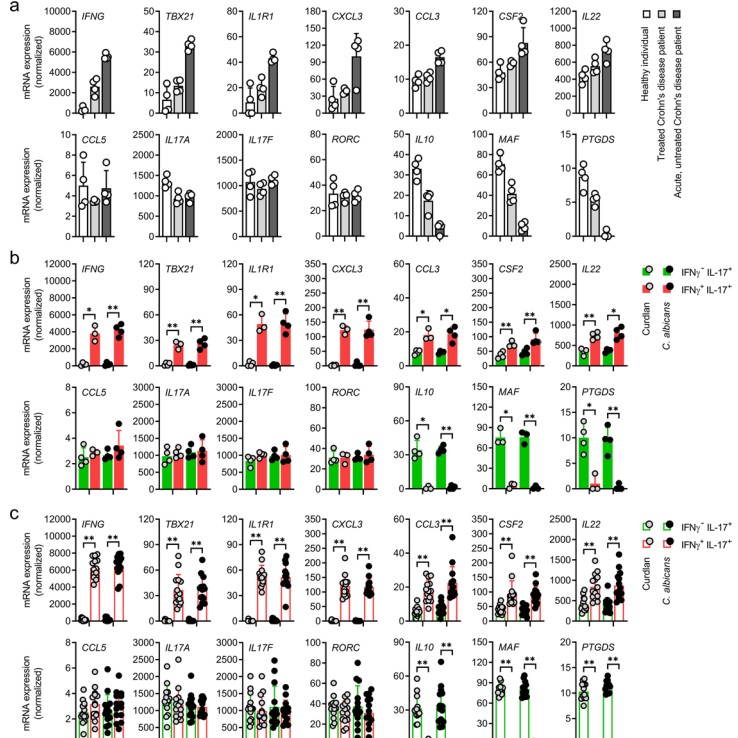

**Extended Data Fig. 5 | Dectin-1 directs polarization of predominantly T_H17 cells with high *IL10, MAF* and *PTGDS* and low *IFNG, TBX21, IL1R1* and *CXCL3* mRNA expression. a-c**, Real-time PCR analyses of *IFNG, TBX21, IL1R1, CXCL3, CCL3, CSF2, IL22, CCL5, IL17A, IL17F, RORC, IL10, MAF* and *PTGDS* normalized mRNA levels in IL-17⁺ T cells, isolated by flow cytometry-based sorting from peripheral blood of healthy donors (**a**, *n* = 4) or active Crohn's disease patients, either treated (**a**, *n* = 4) or acute and untreated (**a**, *n* = 4), or after restimulation of T cells, outgrown in vitro by co-culture of either naive (**b**) or memory (**c**) CD4⁺ T cells with immature DCs (iDC) or DCs primed for 48 h with curdlan or *C. albicans* (**c**), at day 5 (**b**, *n* = 4) or day 11–14 (**c**, *n* = 12). In **b,c**, IL-17⁺ cells were further separated during sorting based on intracellular IFN-γ expression. Results from real-time PCR were quantified using standard curves for all genes and normalized to the expression of reference household gene *ACTB*. These data are an addendum to the data presented in Fig. 2i (**a**) and Fig. 2j (**b,c**). Data represent mean ± s.d. of independent donors. **P < 0.01, *P < 0.05 (Student's *t*-test, paired, two-tailed), calculated between IFN-γ⁻IL-17⁺ and IFN-γ⁻IL-17⁺ T cells as indicated by brackets (**b,c**).

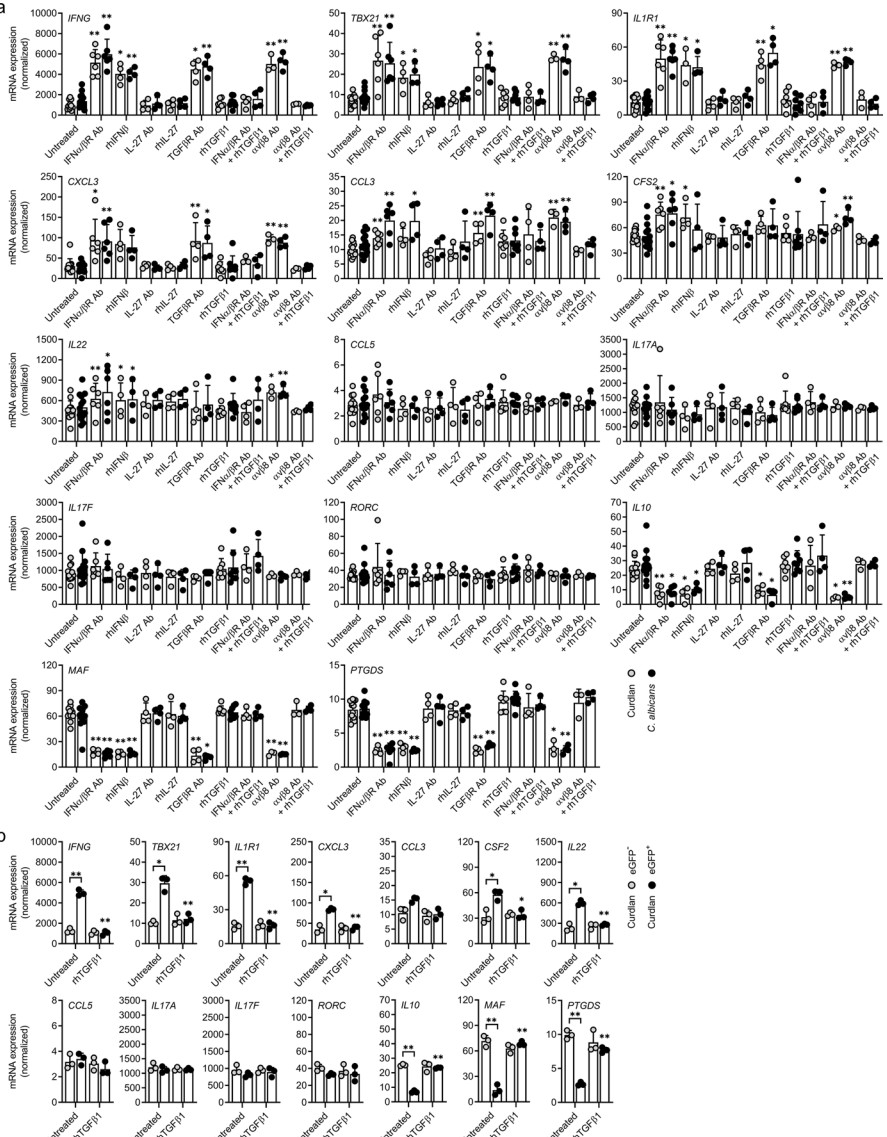

**Extended Data Fig. 6 | Dectin-1 directs non-pathogenic T_H17 polarization via tightly regulated expression of the IFN-stimulated genes β8 and BST2, that are crucial for TGF-β activation. a,b**, Real-time PCR analyses of *IFNG, TBX21, IL1R1, CXCL3, CCL3, CSF2, IL22, CCL5, IL17A, IL17F, RORC, IL10, MAF* and *PTGDS* normalized mRNA levels in IL-17+ T cells, isolated by flow cytometry-based sorting after restimulation of T cells, outgrown in vitro by co-culture of memory CD4+ T cells with immature DCs (iDC) or DCs primed for 48 h with curdlan (**a,b**) or *C. albicans* (**a**), in the presence of blocking IFN-α/βR antibodies (Ab, **a**), rhIFN-β (**a**) (both during DC stimulation), neutralizing IL-27 antibodies (**a**), rhIL-27 (**a**), blocking TGF-βR antibodies (**a**), rhTGF-β1 (**a,b**) (all during DC-T co-culture), blocking αvβ8 antibodies (during DC stimulation) (**a**), or after overexpression

of BST2 (eGFP+ cells) by transfecting DCs with mammalian expression plasmid pCG−BST2-IRES−eGFP and subsequent sorting based on eGFP expression (eGFP−, eGFP+) 24 h later (**b**), at day 11–14 (**a**, untreated n = 12; **d**, IFN-α/βR Ab n = 11, rhIFN-β n = 4; **e**, n = 4; **f**, TGF-βR Ab n = 6, rhTGF-β1 n = 4, αvβ8 Ab n = 3; **b**, n = 3). Results from RT−PCR were quantified using standard curves for all genes and normalized to the expression of reference household gene *ACTB*. These data are an addendum to the data presented in Fig. 2k (**a**), Fig. 4c (**a**), Fig. 5h (**a**), Fig. 6i (**a**) and Fig. 8k (**b**). Data represent mean ± s.d. of independent donors. **P < 0.01, *P < 0.05 (Student's t-test, paired, two-tailed), calculated between untreated and treated samples that were likewise stimulated (**a,b**) and eGFP− and eGFP+ DCs primed with curdlan as indicated by brackets (**b**).

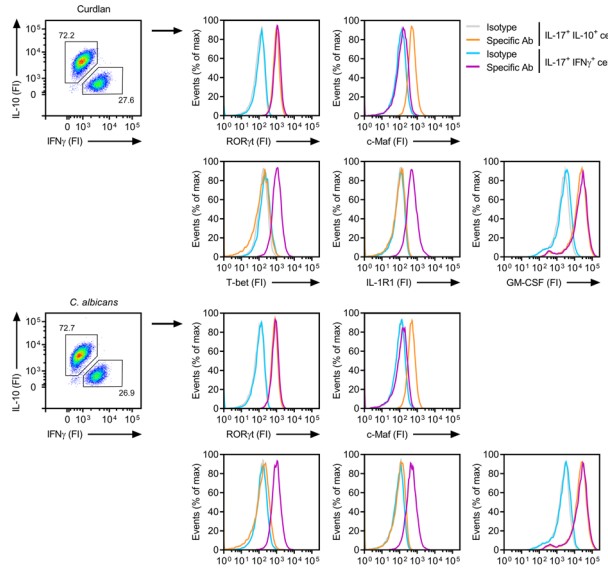

**Extended Data Fig. 7 | Flow cytometry analyses of IL-17⁺ T cells induced by curdlan- or *C. albicans*-primed DCs.** Flow cytometry analyses by staining for intracellular expression (FI, fluorescence intensity) of IL-10, IFN-γ, RORγt, c-Maf, T-bet, IL-1R1 and GM-CSF in restimulated IL-17⁺ T cells, by flow cytometry-based sorting after in vitro T$_H$ polarization by co-culture of memory CD4⁺ T cells with DCs primed for 48 h with curdlan or *C. albicans* at day 11–14. Isotype indicates negative control staining. Representative dot plots - with the percentage positive cells indicated - or histograms for independent donors are shown.

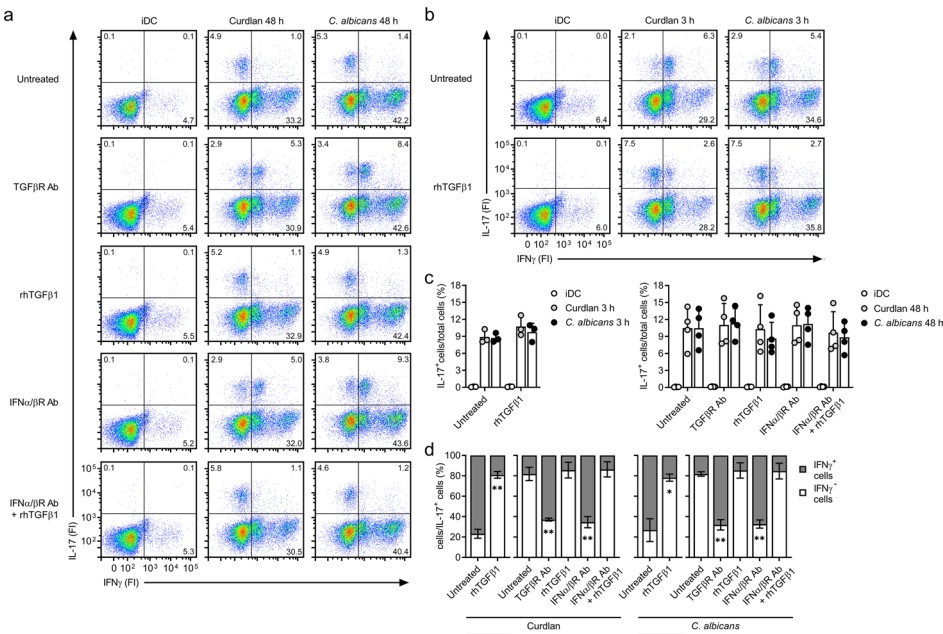

**Extended Data Fig. 8 | Dectin-1-mediated IFNβ expression regulates TGFβ activation that is required for non-pathogenic T_H17 polarization. a-d,** Flow cytometry analyses of T_H polarization by staining for intracellular IL-17 and IFN-γ expression (FI, fluorescence intensity) in restimulated T cells, outgrown in vitro by co-culture of naive CD4+ T cells with immature DCs (iDC) or DCs primed for either 48 h (**a,c,d**, n = 4) or 3 h (**b−d**, n = 3) with curdlan or *C. albicans*, in the presence of blocking IFN-α/βR antibodies (Ab; during DC stimulation) (**a,c,d**) and/or blocking TGFβR antibodies (**a,c,d**) or rhTGFβ1 (**a-d**) (both during DC-T co-culture), at day 5. In **a,b**, representative dot plots for independent donors are shown, with the percentage positive cells indicated in each quadrant. In **c**, the percentage IL-17+ cells per total amount of T cells are shown and in **d**, the percentage IFNγ- and percentage IFNγ+ cells per IL-17+ T cells are shown. Data in **c,d** represent mean ± s.d. of independent donors. **P < 0.01, *P < 0.05 (Student's t-test, paired, two-tailed), calculated between untreated and treated samples that were likewise stimulated.

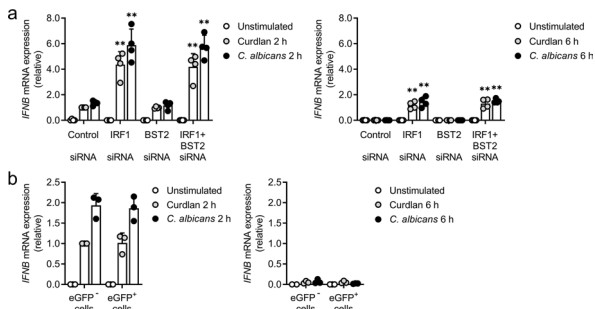

**Extended Data Fig. 9 | BST2 expression has no effect on dectin-1-induced *IFNB* expression. a,b**, Real-time PCR analyses of *IFNB* relative mRNA levels in unstimulated DCs after stimulation of DCs with curdlan or *C. albicans*, after transduction with non-targeting (control) or specific siRNAs to silence expression of BST2 and/or IRF1 (**a**) or after overexpression of BST2 (eGFP⁺ cells) by transfecting DCs with mammalian expression plasmid pCG–BST2-IRES–eGFP and subsequent sorting based on eGFP expression 24 h later (**b**), at 2 or 6 h (**a**, *n* = 4; **b**, *n* = 3). Results from real-time PCR were normalized to the expression of reference household gene *GAPDH* and shown relative to 2 h curdlan stimulation. Data represent mean ± s.d. of independent donors. \*\**P* < 0.01 (Student's *t*-test, paired, two-tailed), calculated between control and specific siRNA-transduced samples that were likewise stimulated.

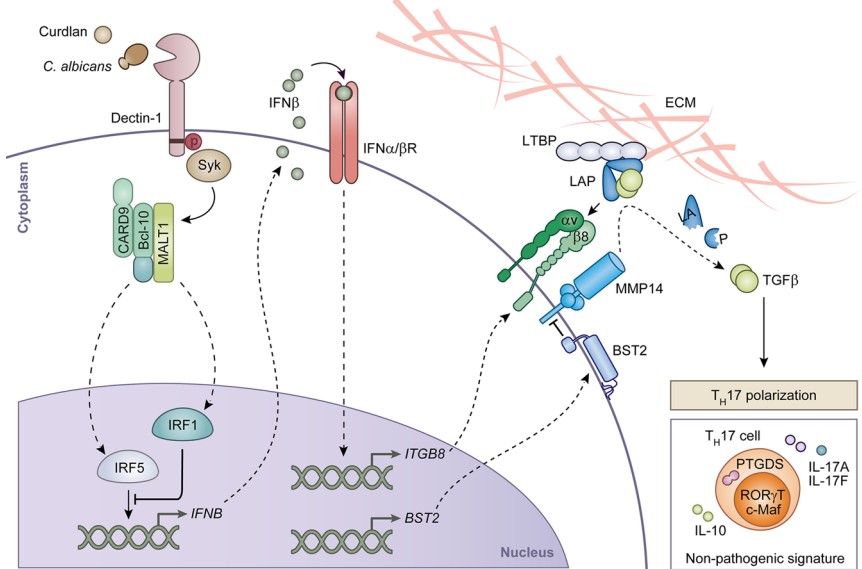

**Extended Data Fig. 10 | Dectin-1 signaling tightly regulates IFNβ expression to control the release of active TGFβ for non-pathogenic T<sub>H</sub>17 polarization.** Dectin-1 ligation by its specific ligand curdlan or fungi like *Candida albicans* induce Syk-CARD9-Bcl10-Malt1-mediated signaling pathways, which lead to activation of the transcription factors IRF1 and IRF5. The antagonistic actions of IRF1 and IRF5 on the promoter of the *IFNB* gene result in strictly controlled expression of IFNβ. Autocrine signaling through the IFN-α/βR induces expression of various IFN-stimulated genes, such as integrin chain β8 and BST2. Expression of αvβ8 integrin on the outer membrane of DCs allows binding of the large latent complex (LLC) consisting of TGFβ, latency-associated peptide (LAP) and latent TGFβ-binding protein (LTBP), that is sequestered within the extracellular matrix (ECM) after secretion. Upon binding of the LLC by αvβ8, matrix metalloproteinase MMP14 cleaves LAP, which frees the bio-active TGF-β dimer from the restraints of the LLC and allows it to bind to the TGF-βR expressed on T cells, thereby influencing the polarization of T helper (T<sub>H</sub>) cells towards T<sub>H</sub>17 cells with a non-pathogenic signature, characterized by expression of the transcription factor c-Maf, anti-inflammatory cytokine IL-10 and effector protein prostaglandin D2 synthase (PTGDS). Expression of IFN-induced BST2 interferes with the proteolytic activity of MMP14, thereby attenuating the release of active TGF-β. An imbalance in IFN-β expression will result in loss of TGF-β activation and differentiation of T<sub>H</sub>17 cells with a pathogenic molecular signature, when either αvβ8 expression is lost or when enhanced BST2 expression increasingly blocks MMP14 activity.

# nature research

# Reporting Summary

Nature Research wishes to improve the reproducibility of the work that we publish. This form provides structure for consistency and transparency in reporting. For further information on Nature Research policies, see our Editorial Policies and the Editorial Policy Checklist.

## Statistics

For all statistical analyses, confirm that the following items are present in the figure legend, table legend, main text, or Methods section.

| n/a | Confirmed | |
|---|---|---|
| ☐ | ☒ | The exact sample size ($n$) for each experimental group/condition, given as a discrete number and unit of measurement |
| ☐ | ☒ | A statement on whether measurements were taken from distinct samples or whether the same sample was measured repeatedly |
| ☐ | ☒ | The statistical test(s) used AND whether they are one- or two-sided<br>*Only common tests should be described solely by name; describe more complex techniques in the Methods section.* |
| ☒ | ☐ | A description of all covariates tested |
| ☒ | ☐ | A description of any assumptions or corrections, such as tests of normality and adjustment for multiple comparisons |
| ☐ | ☒ | A full description of the statistical parameters including central tendency (e.g. means) or other basic estimates (e.g. regression coefficient) AND variation (e.g. standard deviation) or associated estimates of uncertainty (e.g. confidence intervals) |
| ☐ | ☒ | For null hypothesis testing, the test statistic (e.g. $F$, $t$, $r$) with confidence intervals, effect sizes, degrees of freedom and $P$ value noted<br>*Give P values as exact values whenever suitable.* |
| ☒ | ☐ | For Bayesian analysis, information on the choice of priors and Markov chain Monte Carlo settings |
| ☒ | ☐ | For hierarchical and complex designs, identification of the appropriate level for tests and full reporting of outcomes |
| ☒ | ☐ | Estimates of effect sizes (e.g. Cohen's $d$, Pearson's $r$), indicating how they were calculated |

*Our web collection on statistics for biologists contains articles on many of the points above.*

## Software and code

Policy information about availability of computer code

| Data collection | Microsoft Office Professional Plus 2019 - Excel |
|---|---|
| Data analysis | Microsoft Office Professional Plus 2019 - Excel, GraphPad Prism 8.3.1, FlowJo 10.8.1, Primer Express 2.0 |

For manuscripts utilizing custom algorithms or software that are central to the research but not yet described in published literature, software must be made available to editors and reviewers. We strongly encourage code deposition in a community repository (e.g. GitHub). See the Nature Research guidelines for submitting code & software for further information.

## Data

Policy information about availability of data

All manuscripts must include a data availability statement. This statement should provide the following information, where applicable:
- Accession codes, unique identifiers, or web links for publicly available datasets
- A list of figures that have associated raw data
- A description of any restrictions on data availability

The authors declare that the data supporting the findings of this study are available within this paper and its source data files.

# Field-specific reporting

Please select the one below that is the best fit for your research. If you are not sure, read the appropriate sections before making your selection.

☒ Life sciences          ☐ Behavioural & social sciences          ☐ Ecological, evolutionary & environmental sciences

For a reference copy of the document with all sections, see nature.com/documents/nr-reporting-summary-flat.pdf

# Life sciences study design

All studies must disclose on these points even when the disclosure is negative.

| | |
|---|---|
| Sample size | No statistical methods were used to predetermine sample size. |
| Data exclusions | No data were excluded. We did routinely screen our buffy coat donors before experiments for dectin-1 single nucleotide polymorphism rs16910526 using TaqMan SNP Genotyping Assays (Assay ID C_33748481_10; Applied Biosystems) and only used dectin-1 wild-type donors for experiments, unless otherwise indicated. |
| Replication | Typically, we performed all experiments at least 3 or more times to confirm robustness of biological observations, with a few exceptions when results were 'black and white' (no change or complete block). |
| Randomization | No randomization was used. |
| Blinding | Experiments were not blinded but kept as unbiased as possible. |

# Reporting for specific materials, systems and methods

We require information from authors about some types of materials, experimental systems and methods used in many studies. Here, indicate whether each material, system or method listed is relevant to your study. If you are not sure if a list item applies to your research, read the appropriate section before selecting a response.

### Materials & experimental systems

| n/a | Involved in the study |
|---|---|
| ☐ | ☒ Antibodies |
| ☐ | ☒ Eukaryotic cell lines |
| ☒ | ☐ Palaeontology and archaeology |
| ☒ | ☐ Animals and other organisms |
| ☐ | ☒ Human research participants |
| ☒ | ☐ Clinical data |
| ☒ | ☐ Dual use research of concern |

### Methods

| n/a | Involved in the study |
|---|---|
| ☒ | ☐ ChIP-seq |
| ☐ | ☒ Flow cytometry |
| ☒ | ☐ MRI-based neuroimaging |

## Antibodies

| | |
|---|---|
| Antibodies used | CD4 T cell isolation:<br>Alexa Fluor 488-conjugated anti-CD4 (1:50; clone RPA-T4, 300519, Biolegend)<br>APC-conjugated anti-CD45RA (1:50; clone HI100, 550855, BD Bioscience).<br>PE-conjugated anti-CD45RO-PE (200 µg/ml; clone UCHL1, R084301-2, Agilent)<br><br>During culture:<br>blocking antibodies against dectin-1 (20 µg/ml; clone #259931, MAB1859, R&D Systems)<br>blocking antibodies against IFNα/βR2 (20 µg/ml; clone MMHAR-2, PBL Assay Science)<br>anti-αvβ1 (10 µg/ml; clone #P5D2, MAB17781, R&D Systems)<br>anti-αvβ3 (10 µg/ml; clone #23C6, MAB3050, R&D Systems)<br>anti-αvβ5 (10 µg/ml; clone #P5H9, MAB2528, R&D Systems)<br>anti-αvβ6 (10 µg/ml; clone 10D5, ab77906, Abcam)<br>anti-αvβ8 (10 µg/ml; kind gift from S. L. Nishimura) (not commercially available)<br>mouse IgG1 isotype control (20 µg/ml; clone MOPC-21, 555746, BD Pharmingen)<br>mouse IgG2a isotype control (20 µg/ml; clone G155-178, 555571, BD Bioscience)<br>mouse IgG2b isotype control (20 µg/ml; clone #20116, MAB004, R&D Systems)<br>neutralizing antibodies against IL-1β (5 µg/ml, AF-201-NA, R&D Systems)<br>neutralizing antibodies against IL-23 (5 µg/ml, AF1716, R&D Systems)<br>neutralizing antibodies against IL-12 (5 µg/ml, AF-219-NA, R&D Systems)<br>neutralizing antibodies against IL-27 (5 µg/ml, AF2526, R&D Systems)<br>neutralizing antibodies TGFβRII (5 µg/ml, AF-241-NA, R&D Systems)<br>normal goat IgG (AB-108-C; R&D Systems) as a control |

FACS staining:
anti-IRF1 (1:50; ab26109, Abcam)
anti-IRF5 (1:50; ab124792, Abcam)
anti-αv (1:50; AF1219, R&D Systems)
anti-β8 (1:50; ab80673, Abcam)
anti-BST2 (1:100; NIH AIDS Reagent program #11721)
PE-conjugated anti-rabbit (1:200; 711-116-152, Jackson Immunoresearch)
Alexa Fluor 488-conjugated anti-goat (1:400; A11055, Invitrogen)
FITC-conjugated anti-IFNβ (1:20; clone MMHB-3, 21400-3, PBL Assay Science)
FITC-conjugated IgG1κ isotype control mouse Ab (1:250; clone P3.6.2.8.1, 11-4714-81, eBioscience)
APC-conjugated anti-IL-17 (1:25; clone eBio64DEC17, 17-7179-42, eBioscience)
eFluor506-conjugated anti-IL-17 (1:50; clone eBio64DEC17, 69-7179-42, eBioscience)
FITC-conjugated anti-IFN-γ (1:5; clone 25723.11, 25723.11, BD)
PE-conjugated anti-IL-10 (1:10; clone JES3-9D7, 12-7108-82, Invitrogen)
Alexa Fluor 647-conjugated anti-RORγt (1:10; clone Q21-559, 563620, BD)
APC-conjugated anti-T-bet (1:10; clone 4B10, 644814, Biolegend)
eFluor660-conjugated anti-c-Maf (1:10; clone symOF1, 50-9855-82, Invitrogen)
APC-conjugated anti-IL-1R1 (1:10; FAB269A, R&D Systems)
APC-conjugated anti-GM-CSF (1:10; clone BVD2-21C11, 502310, Biolegend)
Alexa Fluor 647-conjugated mouse IgG2b (1:5; clone 27-35, 558713, BD)
APC-conjugated mouse IgG1 (10 µg/ml; clone P3.6.2.8.1, 17-4714-42, Invitrogen)
eFluor660-conjugated mouse IgG2b (10 µg/ml; clone eBMG2b, 50-4732-82, Invitrogen),
APC-conjugated goat IgG (1:5; IC108A, R&D Systems)
APC-conjugated rat IgG2a (2.5 µg/ml; 402305, Biolegend)

ELISA:
anti-pan TGFβ (2 µg/ml, coating Ab; clone 1D11, MAB1835, R&D Systems)
biotinylated anti-TGFβ1 (0.2 µg/ml, detecting Ab; BAF240, R&D Systems)

Immunoblotting:
anti-IRF1 (1:1000; 8478, Cell Signaling)
anti-IRF3 (1:1000, sc-9082, Santa Cruz)
anti-IRF5 (1:1000, ab124792, Abcam)
anti-IRF7 (1:1000; 4920, Cell Signaling)
HRP-conjugated secondary antibody (1:2500; Clean-Blot IP Detection Reagent 21230, Pierce)
anti-β-actin (1:2000; clone ACTBD11B7, sc-81178, Santa Cruz)
HRP-conjugated anti-mouse (1:1000; sc-2314, Santa Cruz)

Immunofluorescense staining:
anti-IRF1 (1:100; ab26109, Abcam)
anti-IRF5 (1:100; ab2932, Abcam)
Alexa Fluor 546-conjugated anti-rabbit (1:400; A10040, Invitrogen)
Alexa Fluor 546-conjugated anti-goat (1:400; A21085, Invitrogen)

ChIP assay:
anti-IRF1 (sc-640X, Santa Cruz)
anti-IRF5 (ab2932, Abcam)
anti-IRF7 (sc-9083X, Santa Cruz)
negative control IgG (sc-2025; Santa Cruz)

Validation

Validation of each primary antibody was based on the information provided by the manufacturer:

CD4 T cell isolation:
Alexa Fluor 488-conjugated anti-CD4 (1:50; clone RPA-T4, 300519, Biolegend): https://www.biolegend.com/nl-nl/products/alexa-fluor-488-anti-human-cd4-antibody-2727
APC-conjugated anti-CD45RA (1:50; clone HI100, 550855, BD Bioscience): https://www.bdbiosciences.com/en-fr/products/reagents/flow-cytometry-reagents/research-reagents/single-color-antibodies-ruo/APC-Mouse-Anti-Human-CD45RA.550855
PE-conjugated anti-CD45RO-PE (200 µg/ml; clone UCHL1, R084301-2, Agilent): https://www.agilent.com/store/en_US/Prod-R084301-2/R084301-2

During culture:
blocking antibodies against dectin-1 (20 µg/ml; clone #259931, MAB1859, R&D Systems): https://www.rndsystems.com/products/human-dectin-1-clec7a-antibody-259931_mab1859
blocking antibodies against IFNα/βR2 (20 µg/ml; clone MMHAR-2, PBL Assay Science): https://www.pblassaysci.com/antibodies/anti-human-ifnar2-antibody-clone-mmhar-2-neutralizing-mab-21385
anti-αvβ1 (10 µg/ml; clone #P5D2, MAB17781, R&D Systems): https://www.rndsystems.com/products/human-integrin-beta1-cd29-antibody-p5d2_mab17781
anti-αvβ3 (10 µg/ml; clone #23C6, MAB3050, R&D Systems): https://www.rndsystems.com/products/human-integrin-alphavbeta3-antibody-23c6_mab3050
anti-αvβ5 (10 µg/ml; clone #P5H9, MAB2528, R&D Systems): https://www.rndsystems.com/products/human-integrin-alphavbeta5-antibody-p5h9_mab2528
anti-αvβ6 (10 µg/ml; clone 10D5, ab77906, Abcam): https://www.abcam.com/integrin-alpha-v--beta-6-antibody-10d5-bsa-and-azide-free-ab77906.html
anti-αvβ8 (10 µg/ml; kind gift from S. L. Nishimura) (not commercially available)

mouse IgG1 isotype control (20 μg/ml; clone MOPC-21, 555746, BD Pharmingen): https://www.bdbiosciences.com/en-au/products/reagents/flow-cytometry-reagents/research-reagents/flow-cytometry-controls-and-lysates/purified-mouse-igg1-isotype-control.555746

mouse IgG2a isotype control (20 μg/ml; clone G155-178, 555571, BD Bioscience): https://www.bdbiosciences.com/en-ca/products/reagents/flow-cytometry-reagents/research-reagents/flow-cytometry-controls-and-lysates/Purified-Mouse-IgG2a,-%CE%BA-Isotype-Control.555571

mouse IgG2b isotype control (20 μg/ml; clone #20116, MAB004, R&D Systems): https://www.rndsystems.com/products/mouse-igg2b-isotype-control_mab004

neutralizing antibodies against IL-1β (5 μg/ml, AF-201-NA, R&D Systems): https://www.rndsystems.com/products/human-il-1beta-il-1f2-antibody_af-201-na

neutralizing antibodies against IL-23 (5 μg/ml, AF1716, R&D Systems): https://www.rndsystems.com/products/human-il-23-p19-antibody_af1716

neutralizing antibodies against IL-12 (5 μg/ml, AF-219-NA, R&D Systems): https://www.rndsystems.com/products/human-il-12-antibody_af-219-na

neutralizing antibodies against IL-27 (5 μg/ml, AF2526, R&D Systems): https://www.rndsystems.com/products/human-il-27-antibody_af2526

neutralizing antibodies TGFβRII (5 μg/ml, AF-241-NA, R&D Systems): https://www.rndsystems.com/products/human-tgf-beta-rii-antibody_af-241-na

normal goat IgG (AB-108-C; R&D Systems) as a control: https://www.rndsystems.com/products/normal-goat-igg-control_ab-108-c

FACS staining:
anti-IRF1 (1:50; ab26109, Abcam): https://www.abcam.com/irf1-antibody-ab26109.html
anti-IRF5 (1:50; ab124792, Abcam): https://www.abcam.com/irf5-antibody-epr6094-ab124792.html
anti-αv (1:50; AF1219, R&D Systems): https://www.rndsystems.com/products/human-mouse-rat-integrin-alphav-cd51-antibody_af1219
anti-β8 (1:50; ab80673, Abcam): https://www.abcam.com/integrin-beta-8-antibody-ab80673.html
anti-BST2 (1:100; NIH AIDS Reagent program #11721)
PE-conjugated anti-rabbit (1:200; 711-116-152, Jackson Immunoresearch): https://www.jacksonimmuno.com/catalog/products/711-116-152/Donkey-Rabbit-IgG-HL-R-Phycoerythrin
Alexa Fluor 488-conjugated anti-goat (1:400; A11055, Invitrogen): https://www.thermofisher.com/antibody/product/Donkey-anti-Goat-IgG-H-L-Cross-Adsorbed-Secondary-Antibody-Polyclonal/A-11055
FITC-conjugated anti-IFNβ (1:20; clone MMHB-3, 21400-3, PBL Assay Science): https://www.pblassaysci.com/antibodies/anti-human-ifn-beta-antibody-clone-mmhb-3-mab-21400 (conjugated no longer available)
FITC-conjugated IgG1κ isotype control mouse Ab (1:250; clone P3.6.2.8.1, 11-4714-81, eBioscience): https://www.thermofisher.com/antibody/product/Mouse-IgG1-kappa-clone-P3-6-2-8-1-Isotype-Control/11-4714-81
APC-conjugated anti-IL-17 (1:25; clone eBio64DEC17, 17-7179-42, eBioscience): https://www.thermofisher.com/antibody/product/IL-17A-Antibody-clone-eBio64DEC17-Monoclonal/17-7179-42
eFluor506-conjugated anti-IL-17 (1:50; clone eBio64DEC17, 69-7179-42, eBioscience): https://www.thermofisher.com/antibody/product/IL-17A-Antibody-clone-eBio64DEC17-Monoclonal/69-7179-42
FITC-conjugated anti-IFN-γ (1:5; clone 25723.11, 25723.11, BD): https://www.bdbiosciences.com/en-us/products/reagents/flow-cytometry-reagents/clinical-discovery-research/single-color-antibodies-ruo-gmp/fitc-mouse-anti-human-ifn.340449
PE-conjugated anti-IL-10 (1:10; clone JES3-9D7, 12-7108-82, Invitrogen): https://www.thermofisher.com/antibody/product/IL-10-Antibody-clone-JES3-9D7-Monoclonal/12-7108-82
Alexa Fluor 647-conjugated anti-RORγt (1:10; clone Q21-559, 563620, BD): https://www.bdbiosciences.com/en-au/products/reagents/flow-cytometry-reagents/research-reagents/single-color-antibodies-ruo/alexa-fluor-647-mouse-anti-human-ror-t.563620
APC-conjugated anti-T-bet (1:10; clone 4B10, 644814, Biolegend): https://www.biolegend.com/en-us/search-results/apc-anti-t-bet-antibody-7120?GroupID=BLG6433
eFluor660-conjugated anti-c-Maf (1:10; clone sym0F1, 50-9855-82, Invitrogen): https://www.thermofisher.com/antibody/product/c-MAF-Antibody-clone-sym0F1-Monoclonal/50-9855-82
APC-conjugated anti-IL-1R1 (1:10; FAB269A, R&D Systems): https://www.rndsystems.com/products/human-il-1-ri-apc-conjugated-antibody_fab269a
APC-conjugated anti-GM-CSF (1:10; clone BVD2-21C11, 502310, Biolegend): https://www.biolegend.com/en-gb/products/apc-anti-human-gm-csf-antibody-7777
Alexa Fluor 647-conjugated mouse IgG2b (1:5; clone 27-35, 558713, BD): https://www.bdbiosciences.com/en-au/products/reagents/flow-cytometry-reagents/research-reagents/single-color-antibodies-ruo/alexa-fluor-647-mouse-igg2b-isotype-control.558713
APC-conjugated mouse IgG1 (10 μg/ml; clone P3.6.2.8.1, 17-4714-42, Invitrogen): https://www.thermofisher.com/antibody/product/Mouse-IgG1-kappa-clone-P3-6-2-8-1-Isotype-Control/17-4714-42
eFluor660-conjugated mouse IgG2b (10 μg/ml; clone eBMG2b, 50-4732-82, Invitrogen): https://www.thermofisher.com/antibody/product/Mouse-IgG2b-kappa-clone-eBMG2b-Isotype-Control/50-4732-82
APC-conjugated goat IgG (1:5; IC108A, R&D Systems): https://www.rndsystems.com/products/goat-igg-apc-conjugated-antibody_ic108a
APC-conjugated rat IgG2a (2.5 μg/ml; 402305, Biolegend): https://www.biolegend.com/fr-fr/products/apc-rat-igg2a-lambda-isotype-control-antibody-19307

ELISA:
anti-pan TGFβ (2 μg/ml, coating Ab; clone 1D11, MAB1835, R&D Systems): https://www.rndsystems.com/products/tgf-beta1-2-3-antibody-1d11_mab1835
biotinylated anti-TGFβ1 (0.2 μg/ml, detecting Ab; BAF240, R&D Systems): https://www.rndsystems.com/products/tgf-beta1-biotinylated-antibody_baf240

Immunoblotting:
anti-IRF1 (1:1000; 8478, Cell Signaling): https://www.cellsignal.com/products/primary-antibodies/irf-1-d5e4-xp-rabbit-mab/8478
anti-IRF3 (1:1000, sc-9082, Santa Cruz): https://www.scbt.com/p/irf-3-antibody-fl-425
anti-IRF5 (1:1000, ab124792, Abcam): https://www.abcam.com/irf5-antibody-epr6094-ab124792.html
anti-IRF7 (1:1000; 4920, Cell Signaling): https://www.cellsignal.com/products/primary-antibodies/irf-7-antibody/4920
HRP-conjugated secondary antibody (1:2500; Clean-Blot IP Detection Reagent 21230, Pierce): https://www.thermofisher.com/order/

catalog/product/21230
anti-β-actin (1:2000; clone ACTBD11B7, sc-81178, Santa Cruz): https://www.scbt.com/p/beta-actin-antibody-actbd11b7?requestFrom=search
HRP-conjugated anti-mouse (1:1000; sc-2314, Santa Cruz): https://www.scbt.com/p/donkey-anti-mouse-igg-hrp

Immunofluorescense staining:
anti-IRF1 (1:100; ab26109, Abcam): https://www.abcam.com/irf1-antibody-ab26109.html
anti-IRF5 (ab2932, Abcam): https://www.abcam.com/irf5-antibody-ab2932.html
Alexa Fluor 546-conjugated anti-rabbit (1:400; A10040, Invitrogen): https://www.thermofisher.com/antibody/product/Donkey-anti-Rabbit-IgG-H-L-Highly-Cross-Adsorbed-Secondary-Antibody-Polyclonal/A10040
Alexa Fluor 546-conjugated anti-goat (1:400; A21085, Invitrogen): https://www.thermofisher.com/antibody/product/Rabbit-anti-Goat-IgG-H-L-Cross-Adsorbed-Secondary-Antibody-Polyclonal/A-21085

ChIP assay:
anti-IRF1 (sc-640X, Santa Cruz): https://www.scbt.com/p/irf-1-antibody-m-20
anti-IRF5 (ab2932, Abcam): https://www.abcam.com/irf5-antibody-ab2932.html
anti-IRF7 (sc-9083X, Santa Cruz): https://www.scbt.com/p/irf-7-antibody-h-246
negative control IgG (sc-2025; Santa Cruz): https://www.scbt.com/p/normal-mouse-igg

# Eukaryotic cell lines

Policy information about cell lines

| | |
|---|---|
| Cell line source(s) | HEK-Blue™ TGF-β reporter cells: Invivogen |
| Authentication | This cell line was not authenticated |
| Mycoplasma contamination | The cell line was tested negative for mycoplasma contamination. |
| Commonly misidentified lines (See ICLAC register) | No misidentified lines are used. |

# Human research participants

Policy information about studies involving human research participants

| | |
|---|---|
| Population characteristics | Healthy donors: F 27 yrs, F 47 yrs, M 27 yrs, M 61 yrs; acute, untreated CD donors: F 20 yrs, F 56 yrs, M 40 yrs, M 50 yrs; treated CD donors: F 37 yrs, F 59 yrs, M 27 yrs, M 70 yrs. |
| Recruitment | Patients that were seen at the IBD clinic of Amsterdam UMC, location AMC, and were diagnosed with Crohn's disease, but had not yet started treatment were selected for the 'acute, untreated' group, while patients that had been diagnosed previously and had gone into remission as a result of treatment were selected for the 'treated' group. We had no criteria on age or gender. We tried to match the age and gender of healthy subjects to the CD donors. |
| Ethics oversight | This study was done in accordance with the ethical guidelines of the Amsterdam UMC, location AMC and human material was obtained in accordance with the AMC Medical Ethics Review Committee (i.e. Institutional Review Committee) according to the Medical Research Involving Human Subjects Act. Buffy coats obtained after blood donation (Sanquin) are not subjected to informed consent according to the Medical Research Involving Human Subjects Act and the AMC Medical Ethics Review Committee. Blood obtained from healthy volunteers was covered by the BACON protocol. Patients with active Crohn's disease were recruited at the IBD clinic of Amsterdam UMC, location AMC. After providing written informed consent, an additional blood sample was drawn in addition to routine blood draws. The project was covered by the Future-IBD biobank protocol. All samples were handled anonymously. |

Note that full information on the approval of the study protocol must also be provided in the manuscript.

# Flow Cytometry

## Plots

Confirm that:

☒ The axis labels state the marker and fluorochrome used (e.g. CD4-FITC).

☒ The axis scales are clearly visible. Include numbers along axes only for bottom left plot of group (a 'group' is an analysis of identical markers).

☒ All plots are contour plots with outliers or pseudocolor plots.

☒ A numerical value for number of cells or percentage (with statistics) is provided.

## Methodology

| | |
|---|---|
| Sample preparation | FACS analyses:<br>Human monocyte-derived dendritic cells were differentiated from monocytes isolated from buffy coats. |

CD4+ T cells were isolated by negative selection from buffy coats, after which naive and memory T cells were separated by positive selection for CD45RO, and used in cocultures with moDCs. Cells were fixed with para-formaldehyde and permeabilized using either saponin or MeOH.

FACS sorting:
moDCs were transfected with a pCG-BST2-IRES-eGFP expression plasmid and sorted based on low and high eGFP.
CD4+ T cells were isolated by negative selection from blood donated by healthy donors or Crohn's disease patients, stimulated for 6 h with PMA/ionomycin, fixed with para-formaldehyde, stained with APC- or eFluor506-conjugated anti-IL-17 and sorted based on IL-17- and IL-17+.

| Instrument | FACS Canto (BD Biosciences), FACS Calibur (BD Biosciences) and FACSAria IIu Cell Sorter (BD Biosciences) |
| --- | --- |
| Software | FlowJo 10.7.1 |
| Cell population abundance | The complete post-sort fraction was relevant and used for further experiments. |

Gating strategy

FACS analyses:
moDCs and T cells were gated on live cells and size based on FCS-A vs. SSC-A. Differentiated T cells were further gated on single cells based on FSC-H vs. FSC-A.

FACS sorting:
For sorting, moDCs cells were further gated on single cells based on FSC-H vs. FSC-W and SSC-H vs. FCS-W. These single cells were plotted GFP vs. a dump channel and the populations were sorted in GFP- and GFP++.
For sorting, T cells cells were further gated on single cells based on FSC-H vs. FSC-W and SSC-H vs. FCS-W. These single cells were plotted IL-17 APC vs. a dump channel or IFNg FITC and the populations were sorted for IL-17+, IL-17+IFNg- or IL-17+IFNg+.

☒ Tick this box to confirm that a figure exemplifying the gating strategy is provided in the Supplementary Information.

