## [Peer Review File · Nature Immunology]

Peer Review Information

Journal: Nature Immunology

Manuscript Title: Dectin-1 directs non-pathogenic TH17 polarization by regulating release of active TGF β via tightly controlled type I IFN responses

Corresponding author name(s): Sonja I. Gringhuis, Teunis B.H. Geijtenbeek

Reviewer Comments & Decisions:

Decision Letter, initial version:

Subject: Decision on Nature Immunology submission NI-A32231

Message: 22nd Jun 2021

Age:

Dear Dr. Geijtenbeek,

We have now finished reviewing your manuscript entitled "Dectin-1 directs non-pathogenic TH17 differentiation by regulating release of active TGF β via tightly controlled type I IFN responses", reference number NI-A32231. Although the editors thought that the manuscript was interesting enough to send out for in-depth review, the reviewers were not in favor of publishing the paper in Nature Immunology. It remains unclear if the conclusions are appropriately supported by the current data set. As such, we cannot accept the manuscript for publication.

Should you find yourself able to thoroughly address the referees' concerns, please let me know. If the novelty of the paper has not been compromised in the interim, you could submit a new paper, although we hope you understand that until we have read the revised manuscript in its entirety we cannot promise that it will be sent back for peer review.

If you decide to resubmit, please include a "Response to referees" detailing, point-by-point, how you addressed each referee comment. If we decide to send the manuscript back to review, this response will be sent back to the referees along with the revised manuscript. In addition, please include a revised version of any required reporting checklist. It will be available to referees to aid in their evaluation if the manuscript goes back for peer review. A revised checklist is essential for re-review of the paper.

The Reporting Summary can be found here:
<https://www.nature.com/authors/policies/ReportingSummary.pdf>

We realize that this is disappointing. I hope that you continue to consider Nature Immunology for your results most significant for the immunology community and wish you well in your future investigations.

Thank you for the opportunity to review your work.

Sincerely,

Ioana Visan, Ph.D.
Senior Editor
Nature Immunology

Tel: 212-726-9207
Fax: 212-696-9752
www.nature.com/ni

Reviewers' comments:

Reviewer #1 (Remarks to the Author):

Major findings of the study:

- The authors propose a model in which Dectin-1 controls the development of non-pathogenic (defined as IL-17 only producing) vs pathogenic (defined as IL-17 plus IFN γ -double producing) Th17 cells via the amount of IFN- β induced in DCs. Low amounts of IFN- β and excess amounts of IFN- β would promote the generation of pathogenic Th17 cells at the expense of non-pathogenic Th17 cells.
- The graded production of IFN- β depended on the ratio of IRF1 vs IRF5 nuclear translocation in DCs: while IRF5 promoted IFN- β expression, IRF1 inhibited it. Dectin-1 induced IFN- β , which in turn controlled the release of TGF- β by DCs by inducing the expression of the integrin $\alpha\text{v}\beta\text{8}$ that primed the latency-associated peptide for cleavage by MMP14, and thus the release of active TGF- β . Exogenous TGF- β could rescue the disruption of the IFN- β -driven release of TGF- β in terms of non-pathogenic Th17 differentiation in a DC-TC co-culture system.
- While TGF- β was released by DCs through Dectin-1/IFN- β -mediated induction of $\alpha\text{v}\beta\text{8}$ integrin, high concentrations of IFN- β (which were facilitated by enhanced IRF5 action through inhibition of IRF1) induced BST2, an inhibitor of MMP14, which prevented TGF- β release by DCs.

General critique:

This is a study that is fully conducted with human DCs and T cells. The study introduces the concept that DCs translate the graded induction of IFN- β by pattern recognition receptor ligands (in particular dectin-1) into differential amounts of TGF- β released in its active form. While low and high amounts of IFN- β that act back on DCs in an autocrine fashion would prevent the release of TGF- β through insufficient induction of $\alpha\text{v}\beta\text{8}$ integrin and inhibition of MMP14, respectively, medium amounts of IFN- β would facilitate the release of TGF- β . Sufficient availability of TGF- β would be necessary (and sufficient) to skew memory T cells into "non-pathogenic" Th17 cells (more IL-17 single producers than IL-17/IFN- γ double producers). This concept provides a TGF- β centered view on Th17 skewing in the context of a relevant danger signal in fungal host defence. Some fundamental findings (IRF1 vs IRF5 mediated IFN- β induction, antagonism of downstream pathways of IFN- β , i. e. integrin $\alpha\text{v}\beta\text{8}$ vs BST2 induction) are brought into a comprehensive theory here. While this is all pretty attractive, some conceptual problems remain that are testable in the experimental set-up that the

authors decided to use for their study (see specific comments).

Specific comments:

1. The concept of pathogenic vs non-pathogenic Th17 cells is still a matter of debate since it has not yet been decided whether these are rather states of Th17 cells than traits. Therefore, it might make a difference whether cytokine phenotypes (IL-17 vs IFN γ) are tested in T cell cultures starting with naive T cells vs T cell cultures where T cells that already have been committed are tested (CD45RO positive). Key experiments should be repeated with naive T cells to see whether the concept of dectin-1 dependent graded expression of IFN- γ also controls the "priming" (initial commitment) of T cells to IL-17 only producers vs IL-17/IFN γ double producers.
2. The present study is difficult to reconcile with the results published by Zielinski et al. 2012 (Zielinski et al., 2012) who propose that *C. albicans* would promote IL-17/IFN- γ double producers (due to IL-1 induction in their co-culture system) while *S. aureus* would induce non-pathogenic Th17 cells that lack IFN- γ production but co-produce IL-10. However, the co-culture systems in both studies are quite different. While in that previous study irradiated autologous monocytes were used as APCs, GM-CSF/IL-4 induced DCs are used in the present study. Furthermore, Zielinski looked primarily at the "priming" of T cells and put "naive" T cells (CD45RO-) in co-culture, in the present study memory T cells (CD4+CD45RO+) were used. In addition, while Zielinski investigated the pathogen specific T cells (combining *C. albicans* as a source of antigens and adjuvants), the present study applies an oligoclonal stimulation system (SEB) in combination with curdlan or heat-killed *C. albicans*. Again, key experiments should be repeated in this prior set-up in order to show the robustness of the authors' concept.
3. IFN- γ (in excess) curtailed MMP14 activity and thus prevented TGF- β release by DCs (resulting in loss of non-pathogenic Th17 cells in favor of pathogenic Th17 cells). While the BST2 mediated inhibition of MMP14 is convincing, the net effect of this pathway is confusing. It is not sufficiently clear how IFN- γ (which promotes the production of IL-27 by myeloid cells) could still enhance pathogenic Th17 cells when IL-27 has been shown to dampen Th17 responses and most importantly, is a strong inducer of IL-10 in T cells (Guo et al., 2008; Stumhofer et al., 2007). What about IL-10 in the "excess" IFN- γ condition? Other transcription factors (except c-MAF) might drive IL-10 here (such as Blimp-1 (encoded by PRMD1)).

Minor comments:

- p. 14, l. 331: wrong Fig. reference (should be Fig. 6g, h).
 p. 14, l. 336 and 339: wrong Fig. reference (should be Fig. 6i instead of Fig. 6j).
 p. 15, l. 357: in the Fig. 7, panel d is missing. In the text, the description refers to Fig. 7e (not 7d, which is missing), and Fig. 7e (ll. 359, 360, 388) refers to the actual panel Fig. 7f.

References

- Guo, B., Chang, E.Y., and Cheng, G. (2008). The type I IFN induction pathway constrains Th17-mediated autoimmune inflammation in mice. *J Clin Invest* 118, 1680-1690. 10.1172/JCI33342.
- Stumhofer, J.S., Silver, J.S., Laurence, A., Porrett, P.M., Harris, T.H., Turka, L.A., Ernst, M., Saris, C.J., O'Shea, J.J., and Hunter, C.A. (2007). Interleukins 27 and 6 induce STAT3-mediated T cell production of interleukin 10. *Nat Immunol* 8, 1363-1371. 10.1038/ni1537.
- Zielinski, C.E., Mele, F., Aschenbrenner, D., Jarrossay, D., Ronchi, F., Gattorno, M., Monticelli, S., Lanzavecchia, A., and Sallusto, F. (2012). Pathogen-induced human TH17 cells produce IFN- γ or IL-10 and are regulated by IL-1 β . *Nature* 484, 514-518. 10.1038/nature10957.

Reviewer #2 (Remarks to the Author):

General comment:

In this manuscript, the authors report that a regulated type I IFN production following dectin-1 stimulation is responsible for the generation of non-pathogenic (i.e., non-IFN-g-producing) Th17 cells. The mechanism underlying this phenomenon is dependent on the two IFN-responsive genes b8 (subunit of the avb8 integrin) and BST2. The avb8 integrin is essential for generation of active TGFb, which is the polarizing factor leading to non-pathogenic Th17 cells. On the other hand, BST2 is induced following high levels of type I IFN and contributes (through MMP14) to the degradation of active TGFb and thus to the generation of pathogenic (i.e., co-expressing IFN-g) Th17. Therefore, dectin-1 fine-tunes type I IFN production to ensure enough active TGFb and the consequent development of non-pathogenic Th17 cells.

Overall, the Authors are to be commended for dissecting in depth the molecular mechanism underlying their observation. Experiments are in general technically sound, with proper controls, and the results clear and easy to interpret. However, the pathophysiological relevance of the Authors' findings is less clear to this reviewer. What would be the in vivo setting whereby excessive type I interferon leads to pathogenic Th17 development? Is this related to fungal infection or (absence of) dectin-1 signaling?

Specific comments:

- Why were the co-cultures of DC performed with memory (rather than naïve) CD4+ T cells? Wouldn't this reflect plasticity rather than polarization?
- The expression of the master transcriptional regulators for Th1 and Th17 (T-bet and Rorg-t, respectively) should be shown.
- In all settings in which pathogenic Th17 were increased, Th1 cells were also significantly increased. This could lead to the conclusion that the effect of regulated type I IFN is on IFN-g production (regardless of the T helper subset). Do the authors have a hypothesis for this increase in IFN-g-producing cells?
- Is IFN-a also induced in this system (in addition to IFN-b)?
- Fig S1A: has LPS been used as an alternative stimulus here and if so, why weren't ISGs induced?

Reviewer #3 (Remarks to the Author):

Summary of the key results

ok

Originality and significance: if not novel, please include reference
see report

Data & methodology: validity of approach, quality of data, quality of presentation
see report

Appropriate use of statistics and treatment of uncertainties

ok

Conclusions: robustness, validity, reliability
see report

Suggested improvements: experiments, data for possible revision
the T cell analyses requires significant additional experiments as indicated below

References: appropriate credit to previous work?
ok

Clarity and context: lucidity of abstract/summary, appropriateness of abstract, introduction and conclusions
all ok

Review

Gringhuis et al describe dectin-1 as a pathogen recognition receptor regulating the development of non-pathogenic Th17. They describe that dectin1 triggering by its ligands induces a type 1 IFN response which initiates the release of active TGFbeta and subsequently "non-pathogenic" Th17 cells, which are mainly defined by the absence of IFNg and a Th1 transcriptional profile whereas anti-inflammatory mediators, such as IL-10 and c-MAF are enhanced. Interestingly either blocking the type 1 IFNR or adding IFNbeta leads to induction of so called "pathogenic" Th17 cells coexpressing IFNg. The authors come to the conclusion that dectin-1 regulation of the IFN β response at an intermediate level generates non-pathogenic TH17 differentiation and either too low or too high IFN β results in pathogenic TH17 polarization.

The manuscript has two main stories: 1. Dectin-1 signaling and regulation of active TGF-beta release 2. The effect of IFN/TGFbeta on human memory T cell differentiation.

In my view the first point regarding transcriptional regulators downstream of dectin-1 in DC, leading to type IFN production and the biochemistry of active TGFbeta release have been nicely analyzed and the data look convincing to me, although this is clearly not my expertise and I cannot judge what is really new, since the pathways analyzed were all described before although maybe not in the same cellular context. Although the induction of IFN by dectin1 is known (e.g. Fresno et al Immunity 38, 1176-1186 (2013)), the fine-tuned IFN induction by dectin1 is a new concept in my view. However, some questions remain regarding this dose effect of dectin-1 signal and IFN (see below).

The second and in my view not well characterized part is that the fine balance of type I IFN decides over pathogenic or non-pathogenic Th17 responses and its molecular regulation via TGF-beta. Here I have several major concerns about the validity of the data and the experimental systems.

1. Pathogenic and non-pathogenic Th17 cells: The role of TGF-beta for non-pathogenic Th17 is known and even led to the term pathogenic versus non-pathogenic Th17. However, this has never been firmly established especially in humans. E.g. I am not aware that strong pathogen-specific Th17 responses in humans e.g. against *Candida* or *S. aureus* recapitulate such pathogenic or non-pathogenic phenotypes as described in the original publication by Zielinski et al Nature 2012. There *Candida* specific responses were claimed to induce pathogenic Th17 cells co-expressing IFNg whereas *S. aureus* induces non-pathogenic Th17 cell producing IL-10 which is actually in contrast to the data from Gringhuis et al. In my view protective or

pathogenic function of a T cell subsets cannot be defined without the context of antigen-specificity., i.e. the same subset may have divergent function when targeting different pathogens.

Therefore I would avoid overuse of the terms pathogenic/non pathogenic but stick to the actual facts, i.e. the induction of IFN γ and other alterations of the transcriptional profile, which are actually not restricted to the Th17 cells in the cultures (see below). Also the term differentiation should be used for stable alterations which occur especially in naïve T cells, whereas transient "modulation" of gene expression may better describe the analyses in the manuscript.

2. The T cell stimulation system in general is unclear to me.

a. The main effects on DC throughout the paper occur during the first hours of stimulation. However, all coculture experiments are done after 48 hours of stimulation. I am wondering what signals DC can still provide at this late time point.

b. The assay also relies on a rather long in vitro expansion of memory T cells, which may introduce another strong experimental bias and leaves completely unclear which cells are targets or may simply survive under these conditions. The effect of DC modulation should be directly visible during the early interaction with DCs within hours or a few days.

c. The memory cells used for coculture are quite undefined making it unclear which subsets are affected. It would be required to use defined starting populations: sorted Th1, Th17, Th17*(Sallusto definition), Th2 and naïve T cells to define which cell types are modulated (e.g. subsets defined by chemokine receptor patterns) or whether we talk about differentiation of naïve T cells.

d. Overall it appears to me that the different conditions mainly modulate the Th1 profile of the culture, rather than inducing IFN γ (or pathogenic phenotype) in a certain memory subset, such as Th17 cells. With TGF β repressing Th1 signatures but enhancing a "non pathogenic" signature, c-MAF, IL-10, which is well known and not restricted to Th17 cells. So we may look on two overlapping effects here which has to be addressed.

e. It remains unclear what induces/recalls actually the IL-17 production in this culture system since it is not blocked by any of the tested factors.

Specific points:

Fig.1 Dosage effect: The authors provide the view that dectin-1 signals always lead to the ideal IFN/TGF β levels. Is there no dose effect on dectin-1 signaling? Also, if the authors believe that high and low concentrations of IFN induce TGF- β and subsequently IFN- γ coexpressing Th17 cells and repress IL-10/c-MAF they should show it, e.g. IFN titration with DC matured in the absence of fungal stimulation.

Fig. 2 Memory cell culture: sorted memory + DC primed with curd/cand + SEB 14 days

There are two effects of the co-culture: strong IL-17 induction or outgrowth of th17 cells but also massive reduction of IFN γ (70 to 20%) in all cells. With anti IFNR and addition IFN there is less decrease of IFN γ . Th17 are not affected.

- What induces IL-17

- The claim that IFN γ /Th17 are specifically induced is not clear. The culture system is very undefined (total memory, 14 days), so it completely unclear which cells are modulated or survive. Also they should not call this Th17 differentiation since they start with memory cells. They should take either naïve T cells, sorted Th17 and Th17* cells (Sallusto protocol) and use Candida specific T cell directly.

- Much shorter stimulation times seem to be relevant: why 14 days? Is this physiologic?

Modulation of memory cells/effect by short lived DC should be visible immediately (hours/few days) after stimulation.

- Also the effects are not consistent, while Candida induces Th17 (not sensitive to aIFNR) and reduces massively IFNg (restored by anti IFNR) addition of IFN reduces IL-17 and increases IFNg.
- Although under both conditions the relative frequency of IL17/IFNg double pos is indeed higher it is completely unclear what happens especially in quantitative terms (induction or selective survival). The authors should report about the quality of the culture, survival, expansion rates. Under both conditions (antiIFNR & IFNbeta) IFNg is massively induced which explains the increase in IL17/IFNg double pos.
- If this is a dosing effect of type 1 IFN (low dose: no effect on IL17 strong block of IFNg – high dose: block of IL-17 + induction of IFNg?) they should show it.
- IN 2C/D they focus on IL-17/IFNg: other cytokines should be included such as IL-10, IL-22, GM-CSF or transcription factors such as c-MAF which are all analysed below on the transcriptional level.

Fig 2 E/F: RNA profile of candidate genes in sorted Th17 cells from CD patients.

- The clear link to the role of type 1 IFN is missing and therefore the relevance of the data is unclear to me:
 - o Remarkable RNA levels of IFNg, but only two CD patients were analysed
 - o FACS stainings should be shown to confirm RNA analysis where possible, especially the critical ones: IL-10, IFNg, c-MAF, GM-CSF, ...
 - o Are there for example more IFNg/IL-17 producers within candida albicans specific T cells (maybe data from literature).

Fig 2G/H: RNA analysis of in vitro cultured total IL-17 or sorted IFNg+ IL-17+ or IFNg- IL-17+ cells

- Sorted IFNg+/-: results confirm expression profile e.g. enriched “pathogenic” genes in IFNg+ and non pathogenic genes in IFNg-. Which is a nice technical control but not surprising since they were sorted for IFNg.
- No difference between cells cultured with and without Curdlan/Candida, suggesting that the in vitro conditions alter the balance between the cells but not the cells per se (again indicating the critical role of the culture conditions as discussed for 2 A/B).
- Since the main effect of the antiIFNR and IFNbeta is actually on IFNg production, I am missing the single IL-17 negative cells and/or IFNg pos cells as a control, i.e. are TH17/IFNg cell unique or just reflecting a mixture between IFNg+ and IL-17+.

Fig. 2H: Confirmation of IFNg/IL-17 results from Fig 2A/B on RNA level.

- These markers should be confirmed by flow cytometry, in addition cytokine production in the SN should be integrated to support the overall effect on cytokine balance.
- Again I am missing the IFNg only and the T cell culture without Curdlan/Candida modulated DCs as a control such as in Fig 2A/B since they have already huge IFNg production,

Fig4: IFNR induces IL-27 (IL1, 6, 23 not affected by aIFNR) but this has no functional impact.

- Are there any high controls for the functionality of the anti IL-27 and blocking antibodies, since IL-27 has been described as a T cell modulator, e.g. to induce IL-10 production and Tr1 differentiation. This should be included.

Fig5: Dectin1/IFN induces release of active TGFbeta

Curd/Cand induces release of active TGFbeta, blocked by aIFNR and anti dectin1

IFN blocks TGFbeta release in a dose dependent fashion and IRF1 or IRF5 knockdown in DC

blocks active TGFbeta release. Claim: dectin1 generates optimal IFN whereas high and low dose IFN prevent the release of active TGF-beta:

- No direct evidence for such a dose effect of IFN, this has to be shown as indicated above by titration of IFN in the absence of Curd/Cand

Fig 5E,F,G TGF beta effect on "pathogenic" Th17 development?

- Again as seen in Fig 2 blockade of TGFbeta, just like aIFNR, mainly leads to a drastic increase in IFNg but not restricted to Th17 cells. Therefore the increase in double positive IFNg/IL17 is rather a consequence of IFNg increase. Addition of TGFbeta has no direct effect but blocks the aIFNR induced IFNg induction, suggesting that TGFbeta in the culture released by Curd/Cand inhibits IFN-g production, as has been suggested before. Overall this does not suggest specific conversion into IFNg+/IL-17+ T cells by TGFbeta.

Fig5H mRNA profile of TGFbeta modulated Th17

- The mRNA supports in my view that TGFbeta mainly suppresses the overall Th1 signature and increases anti-inflammatory signature IL-10/cMAF without specifically affecting Th17 cells. That TGFbeta has this activity on IL-10 is already known.

Although we cannot offer to publish your paper in Nature Immunology, the work may be appropriate for another journal in the Nature Research portfolio. If you wish to explore suitable journals and transfer your manuscript to a journal of your choice, please use our <https://mts-ni.nature.com/cgi-bin/main.plex?el=A6E4OMB1A7LFV3X3A9fdhKquxpaeHTr79FPtucQwZ> manuscript transfer portal. If you transfer to Nature-branded journals or to the Communications journals, you will not have to re-supply manuscript metadata and files. This link can only be used once and remains active until used.

All Nature Research journals are editorially independent, and the decision to consider your manuscript will be taken by their own editorial staff. For more information, please see our http://www.nature.com/authors/author_resources/transfer_manuscripts.html?WT.mc_id=EMI_NPG_1511_AUTHORTRANSF&WT.ec_id=AUTHOR manuscript transfer FAQ page. Note that any decision to opt in to In Review at the original journal is not sent to the receiving journal on transfer. You can opt in to <https://www.nature.com/nature-research/for-authors/in-review> In Review at receiving journals that support this service by choosing to modify your manuscript on transfer. In Review is available for primary research manuscript types only.

Author Rebuttal to Initial comments

Response to Reviewer #1

Specific comments:

1. *The concept of pathogenic vs non-pathogenic Th17 cells is still a matter of debate since it has not yet been decided whether these are rather states of Th17 cells than traits. Therefore, it might make a difference whether cytokine phenotypes (IL-17 vs IFN γ) are tested in T cell cultures starting with naive T cells vs T cell cultures where T cells that already have been committed are tested (CD45RO positive). Key experiments should be repeated with naive T cells to see whether the concept of dectin-1 dependent graded expression of IFN- β also controls the "priming" (initial commitment) of T cells to IL-17 only producers vs IL-17/IFN γ double producers.*

We thank the reviewer for the comment. As requested, we have repeated key T_H outgrowth experiments using naive T cells. These new data support our findings and confirm that the dectin-1-dependent graded expression of IFN β critically determines the phenotype of the *in vitro* differentiated T_H17 cells: limited IFN β expression controls the development of T cells that are mainly IL-17 only producers, whereas blocking of IFN γ signaling or excess IFN β leads primarily to IL-17/IFN γ double producers. We have included these new data in **Fig. 2, Supplementary Fig. 2** and **Supplementary Fig. 5** and discussed the findings in the manuscript.

2. *The present study is difficult to reconcile with the results published by Zielinski et al. 2012 (Zielinski et al., 2012) who propose that C. albicans would promote IL-17/IFN- γ double producers (due to IL-1 induction in their co-culture system) while S. aureus would induce non-pathogenic Th17 cells that lack IFN- γ production but co-produce IL-10. However, the coculture systems in both studies are quite different. While in that previous study irradiated autologous monocytes were used as APCs, GM-CSF/IL-4 induced DCs are used in the present study. Furthermore, Zielinski looked primarily at the "priming" of T cells and put "naive" T cells (CD45RO-) in co-culture, in the present study memory T cells (CD4+CD45RO+) were used. In addition, while Zielinski investigated the pathogen specific T cells (combining C. albicans as a source of antigens and adjuvants), the present study applies an oligoclonal stimulation system (SEB) in combination with curdlan or heat-killed C. albicans. Again, key experiments should be repeated in this prior set-up in order to show the robustness of the authors' concept.*

We have repeated key T_H outgrowth experiments using naive T cells (see comment 1 above) in the absence of SEB and observe similar results as obtained with memory T cells in the presence of SEB, strongly suggesting that neither the use of naive vs memory T cells nor antigen specificity variances account for the observed differences.

However, another major difference between the setup of the experiments – which we had not previously realized – might explain the observed differences and would actually unite our results: while we primed our DCs for 48 h before co-culture with either naive or memory T cells, Zielinski *et al.*

stimulated their APCs for only 3 h prior to addition of naive T cells. In our case, this is a deliberate time frame as it should closely resemble the physiological time frame in which activated DCs will transfer their message to the CD4 T cells, as DCs require 24-72 h to reach lymph nodes once they have entered the lymphatic system (de Winde et al. 2020). This time period will also allow the DCs to develop a cytokine expression profile so they are appropriately equipped to deliver their message to T cells: our results show that it takes at least 8 h before $\alpha\beta 8$ integrin is present on DCs after recognition of ligands by dectin-1 and thus able to generate active TGF β . This would mean that in the experiments of Zielinski *et al.*, after only 3 h APC stimulation, naive T cells were subjected to differentiation signals in the absence of active TGF β , which would explain the development of pathogenic T_H17 cells. To test this hypothesis, we have performed experiments with 3 h curdlan- or *C. albicans*-primed DCs and naive T cells and we observed that the majority of the differentiated T_H17 cells have a pathogenic phenotype, i.e. are IL-17/IFN γ double producers. The addition of rhTGF β during co-culture resulted in the development of non-pathogenic T_H17 cells (**Reviewer Fig. 1-1**). We would like to point out that another study by Bacher *et al.* showed that *C. albicans*-specific circulating memory T_H17 cells in healthy persons have a nonpathogenic phenotype, in line with the results of our study (Bacher et al. 2019). We have now discussed this in the manuscript and if requested we can include the data presented in the reviewer figure in the manuscript in order to reconcile our data with the study by Zielinski *et al.*

3. *IFN- β (in excess) curtailed MMP14 activity and thus prevented TGF- β release by DCs (resulting in loss of non-pathogenic Th17 cells in favor of pathogenic Th17 cells). While the BST2 mediated inhibition of MMP14 is convincing, the net effect of this pathway is confusing. It is not sufficiently clear how IFN- β (which promotes the production of IL-27 by myeloid cells) could still enhance pathogenic Th17 cells when IL-27 has been shown to dampen Th17 responses and most importantly, is a strong inducer of IL-10 in T cells (Guo et al., 2008; Stumhofer et al., 2007). What about IL-10 in the "excess" IFN- β condition? Other transcription factors (except c-MAF) might drive IL-10 here (such as Blimp-1 (encoded by PRMD1)).*

We thank the reviewer for noting this interesting observation. We observe that in the 'excess' IFN β condition – which indeed results in excess IL-27 expression (we have now included these data in **Fig. 4b**) – expression of IL-10, and also transcriptional regulator c-Maf, is suppressed in the outgrown T_H17 cells (**Fig. 2i** and **Supplementary Fig. 2d**), suggesting that IL-27 is not the only factor controlling *IL10* transcription in these T cells. IFN γ is a known suppressor of *IL10* transcription (Hu et al. 2006). As the 'excess' IFN β condition strongly induced IFN γ expression in the outgrown T_H17 cells, we hypothesize that IFN γ counteracts the effects of IL-27R signaling on *IL10* transcription. We have clarified this in the manuscript.

Minor comments:

p. 14, l. 331: wrong Fig. reference (should be Fig. 6g, h).

p. 14, l. 336 and 339: wrong Fig. reference (should be Fig. 6i instead of Fig. 6j).

p. 15, l. 357: in the Fig. 7, panel d is missing. In the text, the description refers to Fig. 7e (not 7d, which is missing), and Fig. 7e (ll. 359, 360, 388) refers to the actual panel Fig. 7f.

We thank the reviewer for pointing out these errors and apologize that we did not triple check the figure references before submitting.

References

de Winde, C. M., Munday, C. & Acton, S. E. Molecular mechanisms of dendritic cell migration in immunity and cancer. *Med. Microbiol. Immunol.* **209**, 515-529 (2020).

Bacher, P. et al. Human Anti-fungal Th17 Immunity and Pathology Rely on Cross-Reactivity against *Candida albicans*. *Cell* **176**, 1340-1355 (2019).

Hu, X. et al. IFN- γ Suppresses IL-10 Production and Synergizes with TLR2 by Regulating GSK3 and CREB/AP-1 Proteins. *Immunity* **24**, 563-574 (2006).

Reviewer Fig. 1-1 | a,b, Flow cytometry analyses of T_H polarization by staining for intracellular IL-17 and IFN γ expression (FI, fluorescence intensity) in restimulated T cells, outgrown *in vitro* by coculture of naive CD4⁺ T cells with immature DCs (iDC) or DCs primed for 3 h (**a,b**) or 48 h (**b**) with curdlan or *C. albicans* CBS2712, in the absence or presence of recombinant human (rh) TGF β 1 ($n = 3$). Representative dot plots for independent donors are shown, with % positive cells indicated in each quadrant. In (**b**), the % IFN γ ⁻ and % IFN γ ⁺ cells per IL-17⁺ T cells are shown. Data in (**b**) represent mean \pm s.d. of independent donors.

Response to Reviewer #2:

Overall, the Authors are to be commended for dissecting in depth the molecular mechanism underlying their observation. Experiments are in general technically sound, with proper controls, and the results clear and easy to interpret. However, the pathophysiological relevance of the Authors' findings is less clear to this reviewer. What would be the *in vivo* setting whereby

excessive type I interferon leads to pathogenic Th17 development? Is this related to fungal infection or (absence of) dectin-1 signaling?

We thank the reviewer for the compliments and interesting comment. Our study shows that dectin-1 signaling strictly controls type I IFN responses to ensure the development of nonpathogenic T_H17 responses during *C. albicans* infection. This allows fungal infections to be cleared without tissue damage and excessive inflammation. Bacher *et al.* have previously shown that circulating *C. albicans*-specific T_H17 cells in healthy donors indeed carry the nonpathogenic phenotype (Bacher et al. 2019). In contrast, T_H17-driven pathologies are observed with other pathogenic fungal species such as *Fonsecaea* and *Malassezia* species, which are associated with excessive skin inflammation. Thus, it is possible that these pathogens interfere with dectin-1 signaling and as such dectin-1-induced type I IFN responses. For example, we have previously shown that *Fonsecaea* species also trigger Mincle signaling, besides dectin-1, which leads to IRF1 degradation (Wevers et al. 2014). As IRF1 is involved in limiting type I IFN responses in response to dectin-1 triggering, our data suggest that the degradation of IRF1 might underlie the induction of pathogenic T_H17 responses to *Fonsecaea* species as a result of excess IFN β . Moreover, pathogenic T_H17 responses induced by excess IFN β might be important in autoimmune diseases such as Crohn's disease where there is a strong IFN signature and pathogenic T_H17 responses (Andreou et al. 2020). Thus, our findings that type I IFN responses play an essential role in dictating pathogenic versus non-pathogenic T_H17 responses has broader implications and might even be important in viral infections. We have clarified this in the manuscript.

Specific comments:

- *Why were the co-cultures of DC performed with memory (rather than naïve) CD4+ T cells? Wouldn't this reflect plasticity rather than polarization?*

We agree with the reviewer that the T_H outgrowth experiments using memory T cells reflect phenotypic plasticity rather than differentiation. We have now clarified this in the manuscript. Furthermore, we have now also included T_H outgrowth assays using naïve T cells. These new data support our findings and confirm that the dectin-1-dependent graded expression of IFN β critically determines the phenotype of the *in vitro* differentiated T_H17 cells: limited IFN β expression controls the development of T cells that are mainly IL-17 only producers, whereas blocking of IFNR signaling or excess IFN β leads primarily to IL-17/IFN γ double producers. We have included these new data in **Fig. 2**, **Supplementary Fig. 2** and **Supplementary Fig. 5** and discussed the findings in the manuscript.

- *The expression of the master transcriptional regulators for Th1 and Th17 (T-bet and Rorγt, respectively) should be shown.*

As requested, we have included flow cytometry analyses that show the expression of RORγt and T-bet in the outgrown T_H17 cells in **Fig. 2m**. These data confirm our transcriptional analyses.

- *In all settings in which pathogenic Th17 were increased, Th1 cells were also significantly increased. This could lead to the conclusion that the effect of regulated type I IFN is on IFN-γ production (regardless of the T helper subset). Do the authors have a hypothesis for this increase in IFN-γ-producing cells?*

We agree with the reviewer that the increase in pathogenic T_H17 cells coincides with the increase in T_H1 cells in our T_H outgrowth assays using memory T cells. TGFβR signaling is known to block T-bet expression and hence *IFNG* transcription, while stimulating c-Maf expression and therefore *IL10* transcription (Lin et al. 2005, Rutz, Noubade et al. 2011). Also, IFNγ expression in turn blocks *IL10* transcription (Hu et al. 2006). As such it is not surprising, and even to be expected, that expression of these genes is affected in both T_H17 and T_H1 cells under the influence of (dys)regulated type I IFN responses and TGFβ activation by DCs. We have now clarified this in the manuscript.

Furthermore, we have now included T_H outgrowth assays using naive T cells, in which we observe that the regulated type I IFN responses specifically control the phenotype of the *in vitro* differentiated T_H17 cells, while hardly affecting T_H1 cells; even during dysregulated type I IFN responses, the level of T_H1 cell induction remains the same. These data indicate that the effects observed in the T_H outgrowth assays using memory T cells reflect the prior presence of differentiated T_H1 cells. We have discussed this in the manuscript.

- *Is IFN-α also induced in this system (in addition to IFN-β)?*

We thank the reviewer for this interesting question. No, IFNα is not induced when dectin-1 signaling is triggered by curdlan or *C. albicans*. This is due to the presence of IRF1 that prevents binding of IFNβ-induced IRF7 to the *IFNA* promoter, similar as occurs on the *IFNB* promoter (**Reviewer Fig. 2-1**).

- *Fig S1A: has LPS been used as an alternative stimulus here and if so, why weren't ISGs induced?*

No, the LPS control was only used when measuring *IFNB* expression after 2 h. We agree with the reviewer that it was unclear this way that LPS was not included with the ISG expression at 6 h, for which we apologize. We have changed the figure for clarity. We have previously shown that indeed LPS induces ISG expression after 6 h (Gringhuis et al. 2014).

References

Bacher, P. et al. Human Anti-fungal Th17 Immunity and Pathology Rely on Cross-Reactivity against *Candida albicans*. *Cell* **176**, 1340-1355 (2019).

Wevers, B. A. et al. Fungal engagement of the C-type lectin mincle suppresses dectin-1-induced antifungal immunity. *Cell Host Microbe* **15**, 494-505 (2014).

Andreou, N.-P., Legaki, E. & Gazouli, M. Inflammatory bowel disease pathobiology: the role of the interferon signature. *Ann Gastroenterol.* **33**, 125-133 (2020).

Lin, J. T., Martin, S. L., Xia, L. & Gorham, J. D. TGF- β 1 Uses Distinct Mechanisms to Inhibit IFN- γ Expression in CD4+ T Cells at Priming and at Recall: Differential Involvement of Stat4 and T-bet. *J. Immunol.* **174**, 5950-5958 (2005).

Rutz, S. et al. Transcription factor c-Maf mediates the TGF- β -dependent suppression of IL-22 production in TH17 cells. *Nat. Immunol.* **12**, 1238-1245 (2011).

Hu, X. et al. IFN- γ Suppresses IL-10 Production and Synergizes with TLR2 by Regulating GSK3 and CREB/AP-1 Proteins. *Immunity* **24**, 563-574 (2006).

Gringhuis, S. I. et al. Fucose-based PAMPs prime dendritic cells for follicular T helper cell polarization via DC-SIGN-dependent IL-27 production. *Nat. Commun.* **5**, 5074 (2014).

Reviewer Fig. 2-1 | RT-PCR analyses of *IFNB* or *IFNA* relative mRNA levels in DCs at indicated times after stimulation with curdlan or *C. albicans* CBS2712, after silencing (siRNA) of IRF1 and/or IRF7 expression (***IFNB* - curdlan**, $n = 4$; **others** $n = 2$). Results from RT-PCR were normalized to the expression of reference household gene *GAPDH* and shown relative to 2 h curdlan stimulation (*IFNB*) or 6 h curdlan stimulation after IRF1 silencing (*IFNA*). Data represent mean \pm s.d. of independent donors.

Response to Reviewer #3:

Major concerns

1. Pathogenic and non-pathogenic Th17 cells: The role of TGF-beta for non-pathogenic Th17 is known and even led to the term pathogenic versus non-pathogenic Th17. However, this has never been firmly established especially in humans. E.g. I am not aware that strong pathogen-specific Th17 responses in humans e.g. against *Candida* or *S. aureus* recapitulate such pathogenic or non-pathogenic phenotypes as described in the original publication by Zielinski et al Nature 2012. There *Candida* specific responses were claimed to induce pathogenic Th17

*cells co-expressing IFN γ whereas *S. aureus* induces non-pathogenic Th17 cell producing IL-10 which is actually in contrast to the data from Gringhuis *et al.* In my view protective or pathogenic function of a T cell subsets cannot be defined without the context of antigen-specificity., i.e. the same subset may have divergent function when targeting different pathogens. Therefore I would avoid overuse of the terms pathogenic/non pathogenic but stick to the actual facts, i.e. the induction of IFN γ and other alterations of the transcriptional profile, which are actually not restricted to the Th17 cells in the cultures (see below).*

We agree with the reviewer that the concept of pathogenic and non-pathogenic T_H17 cells is not as well established in humans as in mice. However, the presence of pathogenic and nonpathogenic T_H17 in humans has been shown in various studies and pathogenic T_H17 cells are observed in various diseases (Annunziato *et al.* 2007, Lazarevic and Glimcher 2011, Patel and Kuchroo 2015, Bsai *et al.* 2019, Kamali *et al.* 2019). Indeed, we have analyzed circulating T_H17 cells isolated from Crohn's disease patients and observed a pathogenic phenotype. Furthermore, we have extended these analyses to healthy donors and established an overall phenotype of human pathogenic and non-pathogenic T_H17 cells. These analyses enabled us to interpret the molecular signatures obtained from sorted IL-17⁺ cells in culture and label them as either 'pathogenic' or 'non-pathogenic' without relying on merely absence or presence of IFN γ expression. Moreover, with regard to *C. albicans* infections, Bacher *et al.* showed that *C. albicans*-specific circulating memory T_H17 cells in healthy persons have a non-pathogenic phenotype, in line with the results of our study (Bacher *et al.* 2019). As requested, we have limited the use of pathogenic and nonpathogenic T_H17 cells and more often refer to IL-17⁺IFN γ ⁺ and IL-17⁺IL-10⁺ Th17 cells.

We are not aware of any reports that show that IFN γ -expressing T_H17 cells (beside Zielinski *et al.*) and/or subsequent tissue damage accompanies *Candida albicans* infections, of course in situations where no further underlying pathologies exist. In fact, we hypothesize that a major difference in the setup between the experiments of Zielinski *et al.* and our experiments might explain the apparent discrepancies and would actually unite our results: while we prime our DCs for 48 h before co-culture with either naive or memory T cells, Zielinski *et al.* stimulate their APCs for only 3 h prior to addition of the naive T cells. As our results show that it takes at least 8 h before $\alpha\text{v}\beta\text{3}$ integrin is present on DCs after recognition of ligands by dectin-1 and thus able to generate active TGF β , this would mean that in the experiments of Zielinski *et al.*, after only 3 h APC stimulation, the naive T cells are subjected to differentiation/polarization signals in the absence of active TGF β and thus elicit a pathogenic T_H17 response. To test this hypothesis, we have performed experiments with 3 h curdlan- or *C. albicans*-primed DCs and naive T cells and we observed that the majority of the differentiated T_H17 cells are IL-17/IFN γ double producers. The addition of rhTGF β during co-culture resulted in the development of non-pathogenic T_H17 cells (**Reviewer Fig. 3-1**). We have now discussed this in the manuscript and if requested we can include the data presented in the reviewer figure in the manuscript in order to reconcile our data with the study by Zielinski *et al.*

Also the term differentiation should be used for stable alterations which occur especially in naïve T cells, whereas transient “modulation” of gene expression may better describe the analyses in the manuscript.

We agree with the reviewer that the experiments with memory T cells support the role of the defined mechanism in determining the phenotype of the outgrown T_H17 cells, thereby reflecting phenotypic plasticity, i.e. 'modulation', rather than differentiation, i.e. stable alterations. We have now clarified this in the manuscript. Furthermore, we have now also included T_H outgrowth assays using naïve T cells. These new data support our findings and confirm that the dectin-1-dependent graded expression of IFN β and subsequent TGF β activation critically determines the phenotype of the *in vitro* differentiated T_H17 cells: limited IFN β expression controls the development of T cells that are mainly IL-17 only producers, whereas blocking of IFN β signaling or excess IFN β leads primarily to IL-17/IFN γ double producers. We have included these new data in **Fig. 2, Supplementary Fig. 2 and Supplementary Fig. 5** and discussed the findings in the manuscript.

2. The T cell stimulation system in general is unclear to me.

a. The main effects on DC throughout the paper occur during the first hours of stimulation. However, all coculture experiments are done after 48 hours of stimulation. I am wondering what signals DC can still provide at this late time point.

We determine transcriptional effects early after stimulation (2-6 h), however the presence of proteins like cytokines (e.g. active TGF β) and ISGs (β 8, BST2) - and also co-stimulatory molecules - is detected much later (24-48 h). We have chosen this later time point also as DC-T cell encounters in lymphoid tissues will only occur after DCs encounter pathogens at mucosal sites, after which DCs require 24-72 h to reach lymph nodes once they have entered the lymphatic system (de Winde et al. 2020). Therefore, we feel that 48 h poststimulation reflects a more physiological time frame that allows DCs to establish a protein expression profile so they are appropriately equipped to deliver their message to T cells. Moreover, our experiments with 3 h primed DCs as described above (see comment 1) and shown in **Reviewer Figure 3-1** suggest that shorter time periods are not sufficient for the development of a full-sized protein expression profile.

b. The assay also relies on a rather long in vitro expansion of memory T cells, which may introduce another strong experimental bias and leaves completely unclear which cells are targets

or may simply survive under these conditions. The effect of DC modulation should be directly visible during the early interaction with DCs within hours or a few days.

We respectfully disagree with the reviewer that the long expansion period might introduce experimental bias as we use the same assay for all conditions and any experimental bias that for example favored survival of a certain T_H17 subtype would eclipse any differences between our various conditions. We have now included new data using naive T cells in an adapted T_H outgrowth assay with a short 5 days co-culture period, which recaptures our data obtained after prolonged memory T cell expansion. We have included these new data in **Fig. 2, Supplementary Fig. 2** and **Supplementary Fig. 5** and discussed the findings in the manuscript.

c. The memory cells used for coculture are quite undefined making it unclear which subsets are affected. It would be required to use defined starting populations: sorted Th1, Th17, Th17(Sallusto definition), Th2 and naive T cells to define which cell types are modulated (e.g. subsets defined by chemokine receptor patterns) or whether we talk about differentiation of naive T cells.*

We agree with the reviewer that it is unclear what memory subsets are affected in the T_H outgrowth experiments using memory T cells. Flow cytometry analyses show that the isolated memory T cells contained <1% naive T cells. However, the focus of our manuscript is on the regulation of the phenotypic traits of T_H17 cells and as such we have not further investigated the origin of the outgrown T_H17 cells.

Moreover, we have now included T_H outgrowth assays using naive T cells and these new data confirm that the dectin-1-dependent regulated expression of IFN β and subsequent TGF β activation critically determines the phenotype of the *in vitro* differentiated T_H17 cells. We have included these new data in **Fig. 2, Supplementary Fig. 2** and **Supplementary Fig. 5** and discussed the findings in the manuscript.

d. Overall it appears to me that the different conditions mainly modulate the Th1 profile of the culture, rather than inducing IFN γ (or pathogenic phenotype) in a certain memory subset, such as Th17 cells. With TGFbeta repressing Th1 signatures but enhancing a “non pathogenic” signature, c-MAF, IL-10, which is well known and not restricted to Th17 cells. So we may look on two overlapping effects here which has to be addressed.

The reviewer raises an interesting point. Indeed, as the reviewer mentions, it is well known that TGF β R signaling blocks T-bet expression and hence *IFNG* transcription, while stimulating c-Maf expression and therefore *IL10* transcription (Lin et al. 2005, Rutz et al. 2011), while in turn IFN γ blocks

IL10 transcription (Hu et al. 2006). As such it is not surprising, and even to be expected, that expression of these genes is affected in both T_H17 and T_H1 cells under the influence of type I IFN responses and $TGF\beta$ activation by DCs. We have now clarified this in the manuscript. Thus, we completely agree with the reviewer that the increase in pathogenic T_H17 cells coincides with the increase in T_H1 cells in our T_H outgrowth assays using memory T cells. The focus of our manuscript is on the regulation of the phenotypic traits of T_H17 cells and as such we have not further investigated the regulation of T_H1 cells.

Importantly, we have now included T_H outgrowth assays using naive T cells, in which we observe that the regulated type I IFN responses specifically control the phenotypic traits of T_H17 cells, while hardly affecting T_H1 cells. We have included these new data in **Fig. 2, Supplementary Fig. 2** and **Supplementary Fig. 5**. These data indicate that the effects observed in the T_H outgrowth assays using memory T cells reflect the prior presence of differentiated T_H1 cells. Moreover, the effect of regulated type I IFN is not merely on IFN γ and IL-10 expression – via regulated $TGF\beta R$ signaling – as during excess IFN β conditions – that lead to excess IL-27 (we have now added those data in **Fig. 4b**) – IL-27R signaling interferes with IL-17 production in T cells (**Fig. 2** and **Fig. 4**). We have discussed this in the manuscript.

e. It remains unclear what induces/recalls actually the IL-17 production in this culture system since it is not blocked by any of the tested factors.

We and others have previously shown that IL-23 and IL-1 β expression is required for IL-17 production in the T_H outgrowth assay with memory T cells (van Beelen et al. 2007, Gringhuis et al. 2011). As the levels of IFN β induced by dectin-1 triggering do not affect IL-23 and IL-1 β expression (**Fig. 4a**), it would have been unexpected if IL-17 production had been blocked. We have clarified this in the manuscript.

Specific points:

Fig.1 Dosage effect: The authors provide the view that dectin-1 signals always lead to the ideal IFN/TGFbeta levels. Is there no dose effect on dectin-1 signaling?

The ligands we have used in the current research are strong ligands of dectin-1 and there is no dose effect on dectin-1 signaling (**Reviewer Fig. 3-2**).

Also, if the authors believe that high and low concentrations of IFN induce TGF-beta and subsequently IFN-g coexpressing Th17 cells and repress IL-10/c-MAF they should show it, e.g. IFN titration with DC matured in the absence of fungal stimulation.

The reviewer raises an interesting question. However, either differentiation of naive T cells to T_H17 cells or recall of IL-17 production in memory T cells requires dectin-1-induced cytokines such as IL-23 and IL-1 β (see above, comment 2e). As such, treatment of DCs with IFN β alone will not provide the signals necessary for induction of T_H17 responses. This is also apparent in the experiments shown in **Fig. 2b** and **Fig. 2f**. Additionally, stimulation of DCs with IFN β alone (concentration range) does not result in release of active TGF β : only during co-stimulation with curdlan or *C. albicans* all required signals that lead to TGF β activation are present (**Fig. 5b**). Thus, without dectin-1 triggering (either via fungal stimulation or curdlan) we are not able to mimic the effects of regulated IFN β expression on T_H17 polarization. Replacing dectin-1 triggering by a random PRR ligand to mature DCs would result in unpredictable expression patterns of both the T_H17-inducing cytokines IL-23, IL-6 and IL-1 β as well as IFN β .

Instead, we mimic low and high IFN β concentrations by blocking IFNR antibodies (**Fig. 2l** and **Supplementary Fig. 2, Fig. 5b**)/blocking $\alpha\beta\delta$ antibodies (**Fig. 6a, Fig. 6i** and **Supplementary Fig. 2**) as well as the addition of excess IFN β (**Fig. 2l** and **Supplementary Fig. 2, Fig. 5c**)/overexpression of BST2 (**Fig. 8j, Fig. 8k** and **Supplementary Fig. 2**), respectively, and analyzed their effects on TGF β activation by DCs and T-bet/IFN γ and cMaf/IL-10 expression in outgrown T_H17 cells.

Fig. 2 Memory cell culture: sorted memory + DC primed with curd/cand + SEB 14 days There are two effects of the co-culture: strong IL-17 induction or outgrowth of th17 cells but also massive reduction of IFN γ (70 to 20%) in all cells. With anti IFNR and addition IFN there is less decrease of IFN γ . Th17 are not affected.

- What induces IL-17

Dectin-1-induced IL-23 and IL-1 β expression is required for IL-17 production in the T_H outgrowth assay with memory T cells (see comment 2e) (van Beelen et al. 2007, Gringhuis et al. 2011). As the levels of IFN β induced by dectin-1 triggering do not affect IL-23 and IL-1 β expression (**Fig. 4a**), IL-17 production in outgrown T_H 17 cells is not affected.

The lower % of IFN γ -expressing cells in our T_H outgrowth assays using memory T cells likely reflects the levels of TGF β in the culture. As discussed above (see comment 2d), TGF β R signaling blocks T-bet expression and hence *IFNG* transcription, while stimulating cMaf expression and therefore *IL10* transcription (Lin et al. 2005, Rutz et al. 2011), while in turn IFN γ blocks *IL10* transcription (Hu et al. 2006). As TGF β activation is blocked in the presence of blocking IFNR antibodies as well as excess IFN β , *IFNG* transcription is no longer blocked in both T_H17 and T_H1 cells, hence the observed 'less decrease'.

- *The claim that IFN γ /Th17 are specifically induced is not clear. The culture system is very undefined (total memory, 14 days), so it completely unclear which cells are modulated or survive. Also they should not call this Th17 differentiation since they start with memory cells. They should take either naïve T cells, sorted Th17 and Th17* cells (Sallusto protocol) and use Candida specific T cell directly.*

We agree with the reviewer that the experiments with memory T cells support the role of the defined mechanism in determining the phenotype of the outgrown T_H17 cells, thereby reflecting phenotypic plasticity, i.e. 'modulation', rather than differentiation, i.e. stable alterations. We have now clarified this in the manuscript.

Furthermore, we have now also included T_H outgrowth assays using naive T cells with a short 5-days co-culture period, which recaptures our data obtained after prolonged memory T cell expansion and confirms that the dectin-1-dependent graded expression of IFN β and subsequent TGF β activation critically determines the phenotype of the *in vitro* differentiated T_H17 cells. In the T_H outgrowth assays using naive T cells, we observe that the regulated type I IFN responses specifically control the phenotypic traits of T_H17 cells, while hardly affecting T_H1 cells. We have included these new data in **Fig. 2**, **Supplementary Fig. 2** and **Supplementary Fig. 5** and discussed the findings in the manuscript.

- *Much shorter stimulation times seem to be relevant: why 14 days? Is this physiologic? Modulation of memory cells/effect by short lived DC should be visible immediately (hours/few days) after stimulation.*

We agree with the reviewer that effects of DC-induced protein expression profiles are visible after 4-5 days (not hours), as has been reported before. We have now included new data using naive T cells in an adapted T_H outgrowth assay with a short 5 days co-culture period, which recaptures our data obtained after prolonged memory T cell expansion. The long expansion time of the memory T cells is a practical issue to obtain enough cells to perform our analyses. (Of note, we do not expand the memory T cells for a set time period of 14 days, but we keep them in culture until they reach a resting state, which is typically between 11-14 days after the start of the co-culture). The 5 days co-cultures with naive T cells are started with 10 times more T cells than the co-cultures with memory T cells to overcome this obstacle.

- *Also the effects are not consistent, while Candida induces Th17 (not sensitive to α IFNR) and reduces massively IFN γ (restored by anti IFNR) addition of IFN reduces IL-17 and increases IFN γ .*

We are afraid that the reviewer has misunderstood, and we will further clarify this in the manuscript. It is to be expected that the conditions in which IFNR signaling is blocked or excess IFN β is present result in different outcomes as both conditions have distinct effects on the DC-derived cytokines (IL-23, IL-1 β , IL-6, IL-27 and TGF β) that affect IL-17 and IFN γ expression in T cells:

(a) during the 'normal' situation with curdlan- or *C. albicans*-primed DCs, dectin-1 signaling induces IL-23, IL-1 β and IL-6 expression (**Fig. 4a**), which induce IL-17 production in T cells. Simultaneously, dectin-1-induced graded type I IFN responses lead to TGF β activation (**Fig. 5b-d**) and subsequent TGF β R signaling in differentiated T cells blocks T-bet expression and as such IFN γ expression. The type I IFN responses also lead to IL-27 expression (**Fig. 1g, Fig. 4b**), however these levels remain below the threshold levels that are required for IL-27 to block IL-17 production in T cells (**Fig. 4c,d**).

(b) when IFNR signaling is blocked, this has no effect on IL-23, IL-1 β and IL-6 expression (**Fig. 4a**), while IL-27 expression is attenuated (**Fig. 4b**) and thus IL-17 production in T cells remains unaltered. Concurrently, the release of active TGF β is blocked (**Fig. 5b**) – as this depends on IFNR-induced $\alpha\beta 8$ expression (**Fig. 6a-h**) – and as such TGF β R signaling can no longer block IFN γ expression in T cells.

(c) during 'excess' IFN β condition (either by addition of rhIFN β or silencing of IRF5), IL-27 expression by DCs is increased (we have added those data to **Fig. 4b**), now reaching levels that do interfere with IL-17 production in T cells, as also seen in the 'excess' rhIL-27 condition (**Fig. 4d**). Concurrently, the release of active TGF β is blocked (**Fig. 5d**) – as enhanced type I IFN responses also lead to increased levels of BST2 expression (**Fig. 8b,e**), which interferes with MMP14-mediated release of active TGF β (**Fig. 8i,j**) – and as such TGF β R signaling can no longer block IFN γ expression in T cells, similar as seen during IFNR block (b).

- Although under both conditions the relative frequency of IL17/IFN γ double pos is indeed higher it is completely unclear what happens especially in quantitative terms (induction or selective survival). The authors should report about the quality of the culture, survival, expansion rates. Under both conditions (antiIFNR & IFNbeta) IFN γ is massively induced which explains the increase in IL17/IFN γ double pos.

We have now included new data using naive T cells in an adapted T_H outgrowth assay with a short 5 days co-culture period, which recaptures our data obtained after prolonged memory T cell expansion. These data confirm that the dectin-1-dependent regulated expression of IFN β and subsequent TGF β activation critically determines the phenotype of the *in vitro* differentiated T_H17 cells and is not a result of selective cell survival or a bias in the culture system. Interestingly, in these T_H outgrowth assays using naive T cells, we observe that the regulated type I IFN responses specifically control the

phenotypic traits of T_H17 cells, while hardly affecting T_H1 cells. We have included these new data in **Fig. 2**, **Supplementary Fig. 2** and **Supplementary Fig. 5** and discussed the findings in the manuscript.

- If this is a dosing effect of type 1 IFN (low dose: no effect on IL17 strong block of IFN γ – high dose: block of IL-17 + induction of IFN γ ?) they should show it.

As mentioned above (see 2nd specific comment), the reviewer raises an interesting question. However, either differentiation of naive T cells to T_H17 cells or recall of IL-17 production in memory T cells requires dectin-1-induced cytokines such as IL-23 and IL-1 β (see above, comment 2e). As such, treatment of DCs with IFN β alone will not provide the signals necessary for induction of T_H17 responses. This is also apparent in the experiments shown in **Fig. 2b** and **Fig. 2f**. Additionally, stimulation of DCs with IFN β alone (concentration range) does not result in release of active TGF β : only during co-stimulation with curdlan or *C. albicans* all required signals that lead to TGF β activation are present (**Fig. 5b**). Thus, without dectin-1 triggering (either via fungal stimulation or curdlan) we are not able to mimic the effects of regulated IFN β expression on T_H17 polarization. Replacing dectin-1 triggering by a random PRR ligand to mature DCs would result in unpredictable expression patterns of both the T_H17-inducing cytokines IL-23, IL-6 and IL-1 β as well as IFN β .

Instead, we mimic low and high IFN β concentrations by other means: blocking IFN β antibodies (**Fig. 2l** and **Supplementary Fig. 2, Fig. 5b**) and blocking $\alpha\beta 8$ antibodies (**Fig. 6a, Fig. 6i** and **Supplementary Fig. 2**) mimic a no/low IFN β environment, resulting in no TGF β activation by DCs and thus IFN γ induction in T cells, without affecting IL-17 production in T cells that is mediated by dectin-1-induced IL-23 and IL-1 β , which are not affected by low IFN β levels (**Fig. 4a**). On the other hand, excess IFN β (**Fig. 2l** and **Supplementary Fig. 2, Fig. 5c**) creates a high IFN β environment that interferes with TGF β activation by DCs (due to high BST2 levels) and thus IFN γ induction in T cells, while at the same time high levels of IFN β -induced IL-27 interfere with IL-17 production in T cells. Excess IL-27 (**Fig. 4c-e** and **Supplementary Fig. 2**) mimics the part of a high IFN β environment that interferes with IL-17 induction in T cells, without interfering with TGF β activation and thus IFN γ induction in T cells remains suppressed. Overexpression of BST2 (**Fig. 8j, Fig. 8k** and **Supplementary Fig. 2**) mimics the part of a high IFN β environment that results in no TGF β activation by DCs and thus allows IFN γ induction in T cells, without affecting IL-17 production in T cells.

- IN 2C/D they focus on IL-17/IFN γ : other cytokines should be included such as IL-10, IL-22, GM-CSF or transcription factors such as c-MAF which are all analysed below on the transcriptional level.

As requested, we have added flow cytometry analyses that show the expression of transcription factors ROR γ t, T-bet and c-Maf, as well as other cytokines (IL-10 and GM-CSF) and IL-1R1 by the outgrown T_H17 cells in **Fig. 2m**.

Fig 2 E/F: RNA profile of candidate genes in sorted Th17 cells from CD patients. - The clear link to the role of type 1 IFN is missing and therefore the relevance of the data is unclear to me:

We included the analyses of circulating T_H17 cells isolated from Crohn's disease patients to set the 'baseline' for pathogenic T_H17 cells so we could demonstrate the similarities with T_H17 cells that developed in our *in vitro* T_H outgrowth assays. We have clarified this in the manuscript.

o Remarkable RNA levels of IFN γ , but only two CD patients were analysed

For further confirmation, we have added data from two more CD patients. As these are diagnosed but yet untreated patients, they are hard to find.

o FACS stainings should be shown to confirm RNA analysis where possible, especially the critical ones: IL-10, IFN γ , c-MAF, GMCSF, ...

As requested, we have added flow cytometry analyses that show the expression of transcription factors ROR γ t, T-bet and c-Maf, as well as other cytokines (IL-10 and GM-CSF) and IL-1R1 by the outgrown T_H17 cells in **Fig. 2m**.

o Are there for example more IFN γ /IL-17 producers within candida albicans specific T cells (maybe data from literature).

We are not aware of any reports that show that IFN γ -expressing T_H17 cells (beside Zielinski *et al.*) and/or subsequent tissue damage accompanies *Candida albicans* infections, of course in situations where no further underlying pathologies exist. In contrast, Bacher *et al.* show that *C. albicans*-specific circulating memory T_H17 cells in healthy persons have a nonpathogenic phenotype, in line with the results of our study (Bacher et al. 2019).

Fig 2G/H: RNA analysis of in vitro cultured total IL-17 or sorted IFN γ + IL-17+ or IFN γ - IL-17+ cells

- *Sorted IFN γ +/-: results confirm expression profile e.g. enriched “pathogenic” genes in IFN γ + and non pathogenic genes in IFN γ -. Which is a nice technical control but not surprising since they were sorted for IFN γ .*

The gene expression profile of human pathogenic and non-pathogenic T_H17 cells is less well established than that of mice T_H17 cells. Moreover, distinct markers that are linked to one of these phenotypes have been described in separate papers. We have therefore performed these analyses to establish the overall phenotype of human pathogenic and non-pathogenic T_H17 cells. Moreover, these analyses allowed us to interpret the molecular signature obtained from overall T_H17 cells of blood of patients/healthy donors and T_H outgrowth cultures.

- *No difference between cells cultured with and without Curdlan/Candida, suggesting that the in vitro conditions alter the balance between the cells but not the cells per se (again indicating the critical role of the culture conditions as discussed for 2 A/B).*

We are unsure to what the reviewer is referring as we clearly show that memory T cells, and now also naive T cells, that are co-cultured in the presence of curdlan- or *C. albicans*-primed DCs demonstrate IL-17 production, while immature DCs do not induce this response.

- *Since the main effect of the antiIFN γ and IFN β is actually on IFN γ production, I am missing the single IL-17 negative cells and/or IFN γ pos cells as a control, i.e. are TH17/IFN γ cell unique or just reflecting a mixture between IFN γ + and IL-17+.*

The IFN γ + T_H17 cells have a unique phenotype and are not a mixture of IFN γ + and IL-17+ cells: all molecular analyses in **Fig. 2i** (previously **Fig. 2h**) – after IFN γ block or excess IFN β treatment of DCs - were performed on sorted IL-17+ cells and as such do not contain single IFN γ + cells. As the focus of our manuscript is on the regulation of the phenotypic traits of T_H17 cells, we have not further investigated the expression of the various genes in T_H17 cells.

2H: Confirmation of IFN γ /IL-17 results from Fig 2A/B on RNA level. - These markers should be confirmed by flow cytometry, in addition cytokine production in the SN should be integrated to support the overall effect on cytokine balance.

As requested, we have added flow cytometry analyses that show the expression of transcription factors ROR γ t, T-bet and c-Maf, cytokines IFN γ , IL-10 and GM-CSF as well as IL-1R1 by the outgrown T_H17 cells in **Fig. 2m**.

We have also measured IL-17A and IFN γ in the supernatant: the ELISA data corroborate our flow cytometry analyses and demonstrate that only excess IFN β levels reduce IL-17 production, while blocking IFNR or TGF β R signaling greatly enhances IFN γ production by T cells (**Reviewer Fig. 3-3**). If requested we can include the data presented in the reviewer figure in the manuscript.

- Again I am missing the IFN γ only and the T cell culture without Curdlan/Candida modulated DCs as a control such as in Fig 2A/B since they have already huge IFN γ production,

The molecular analyses in **Fig. 2l** (previously **Fig. 2h**) – after IFNR block or excess IFN β treatment of DCs - were performed on sorted IL-17⁺ cells and as such do not contain single IFN γ ⁺ cells. As the focus of our manuscript is on the regulation of the phenotypic traits of T_H17 cells, we have not further investigated the expression of the various genes in T_H1 cells. However, we have now included T_H outgrowth assays using naive T cells, in which we observe that the regulated type I IFN responses specifically control the phenotypic traits of T_H17 cells, while hardly affecting T_H1 cells. In co-cultures of immature DCs with naive T cells, hardly any IFN γ -producing cells are induced, supporting an important role for DC maturation and the establishment of a full-sized cytokine expression profile. We have included these new data in **Fig. 2**, **Supplementary Fig. 2** and **Supplementary Fig. 5**. These data indicate that the effects observed in the T_H outgrowth assays using memory T cells reflect the prior presence of differentiated T_H1 cells.

Fig4: IFNR induces IL-27 (IL1, 6, 23 not affected by α IFNR) but this has no functional impact. - Are there any high controls for the functionality of the anti IL-27 and blocking antibodies, since IL-27 has been described as a T cell modulator, e.g. to induce IL-10 production and Tr1 differentiation. This should be included.

The antibody is functional as we have shown previously (Gringhuis et al. 2014).

Fig5: Dectin1/IFN induces release of active TGFbeta

Curd/Cand induces release of active TGFbeta, blocked by α IFNR and anti dectin1 IFN blocks TGFbeta release in a dose dependent fashion and IRF1 or IRF5 knockdown in DC blocks active

TGFbeta release. Claim: dectin1 generates optimal IFN whereas high and low dose IFN prevent the release of active TGF-beta:

- *No direct evidence for such a dose effect of IFN, this has to be shown as indicated above by titration of IFN in the absence of Curd/Cand*

As mentioned above (see 2nd specific comment), the reviewer raises an interesting question. However, either differentiation of naive T cells to T_H17 cells or recall of IL-17 production in memory T cells requires dectin-1-induced cytokines such as IL-23 and IL-1 β (see above, comment 2e). As such, treatment of DCs with IFN β alone will not provide the signals necessary for induction of T_H17 responses. This is also apparent in the experiments shown in **Fig. 2b** and **Fig. 2f**. Additionally, stimulation of DCs with IFN β alone (concentration range) does not result in release of active TGF β : only during co-stimulation with curdlan or *C. albicans* all required signals that lead to TGF β activation are present (**Fig. 5b**). Thus, without dectin-1 triggering (either via fungal stimulation or curdlan) we are not able to mimic the effects of regulated IFN β expression on T_H17 polarization. Replacing dectin-1 triggering by a random PRR ligand to mature DCs would result in unpredictable expression patterns of both the T_H17-inducing cytokines IL-23, IL-6 and IL-1 β as well as IFN β .

Instead, we mimic low and high IFN β concentrations by other means: blocking IFNR antibodies (**Fig. 2l** and **Supplementary Fig. 2, Fig. 5b**) and blocking $\alpha\beta 8$ antibodies (**Fig. 6a, Fig. 6i** and **Supplementary Fig. 2**) mimic a no/low IFN β environment, resulting in no TGF β activation by DCs and thus IFN γ induction in T cells, without affecting IL-17 production in T cells that is mediated by dectin-1-induced IL-23 and IL-1 β , which are not affected by low IFN β levels (**Fig. 4a**). On the other hand, excess IFN β (**Fig. 2l** and **Supplementary Fig. 2, Fig. 5c**) creates a high IFN β environment that interferes with TGF β activation by DCs (due to high BST2 levels) and thus IFN γ induction in T cells, while at the same time high levels of IFN β -induced IL-27 interfere with IL-17 production in T cells. Excess IL-27 (**Fig. 4c-e** and **Supplementary Fig. 2**) mimics the part of a high IFN β environment that interferes with IL-17 induction in T cells, without interfering with TGF β activation and thus IFN γ induction in T cells remains suppressed. Overexpression of BST2 (**Fig. 8j, Fig. 8k** and **Supplementary Fig. 2**) mimics the part of a high IFN β environment that results in no TGF β activation by DCs and thus allows IFN γ induction in T cells, without affecting IL-17 production in T cells.

Fig 5E,F,G TGF beta effect on "pathogenic" Th17 development?

- *Again as seen in Fig 2 blockade of TGbeta, just like α IFNR, mainly leads to a drastic increase in IFNg but not restricted to Th17 cells. Therefore the increase in double positive IFNg/IL17 is rather a consequence of IFNg increase. Addition of TGFbeta has no direct effect but blocks the α IFNR induced IFNg induction, suggesting that TGFbeta in the culture released by Curd/Cand inhibits IFN-g production, as has been suggested before. Overall this does not suggest specific conversion into IFNg+/IL-17+ T cells by TGFbeta.*

Fig5H mRNA profile of TGFbeta modulated Th17

- *The mRNA supports in my view that TGFbeta mainly suppresses the overall Th1 signature and increases anti-inflammatory signature IL-10/cMAF without specifically affecting Th17 cells. That TGFbeta has this activity on IL-10 is already known.*

Indeed, as discussed above (see comment 2d), it is well known that TGF β R signaling blocks T-bet expression and hence *IFNG* transcription, while stimulating c-Maf expression and therefore *IL10* transcription (Lin et al. 2005, Rutz et al. 2011). Also, IFN γ expression in turn blocks *IL10* transcription (Hu et al. 2006). As such it is not surprising, and even to be expected, that expression of these genes is affected in both T_H17 and T_H1 cells under the influence of type I IFN responses and TGF β activation by DCs. We have now clarified this in the manuscript. The focus of our manuscript is on how dectin-1 signaling specifically directs various processes to make sure that active TGF β is released to ensure development of IL10-producing T_H17 cells for fungal clearance and as such we have not further investigated the regulation of T_H1 cells. Furthermore, we have now included T_H outgrowth assays using naive T cells, in which we observe that the regulated type I IFN responses specifically control the phenotypic traits of T_H17 cells, while hardly affecting T_H1 cells. We have included these new data in **Fig. 2, Supplementary Fig. 2** and **Supplementary Fig. 5**. These data indicate that the effects observed in the T_H outgrowth assays using memory T cells reflect the prior presence of differentiated T_H1 cells.

References

Annunziato, F. et al. Phenotypic and functional features of human Th17 cells. *J. Exp. Med.* **204**, 1849-1861 (2007).

Lazarevic, V. & Glimcher, L. H. T-bet in disease. *Nat. Immunol.* **12**, 597-606 (2011).

Patel, D. D. & Kuchroo, V. K. Th17 Cell Pathway in Human Immunity: Lessons from Genetics and Therapeutic Interventions. *Immunity* **43**, 1040-1051 (2015).

Bsat, M. et al. Differential Pathogenic Th17 Profile in Mesenteric Lymph Nodes of Crohn's Disease and Ulcerative Colitis Patients. *Front. Immunol.* **10** (2019).

Kamali, A. N. et al. A role for Th1-like Th17 cells in the pathogenesis of inflammatory and autoimmune disorders. *Mol. Immunol.* **105**, 107-115 (2019).

Bacher, P. et al. Human Anti-fungal Th17 Immunity and Pathology Rely on Cross-Reactivity against *Candida albicans*. *Cell* **176**, 1340-1355 (2019).

de Winde, C. M., Munday, C. & Acton, S. E. Molecular mechanisms of dendritic cell migration in immunity and cancer. *Med. Microbiol. Immunol.* **209**, 515-529 (2020).

Lin, J. T., Martin, S. L., Xia, L. & Gorham, J. D. TGF- β 1 Uses Distinct Mechanisms to Inhibit IFN- γ Expression in CD4+ T Cells at Priming and at Recall: Differential Involvement of Stat4 and T-bet. *J. Immunol.* **174**, 5950-5958 (2005).

Rutz, S. et al. Transcription factor c-Maf mediates the TGF- β -dependent suppression of IL-22 production in TH17 cells. *Nat. Immunol.* **12**, 1238-1245 (2011).

Hu, X. et al. IFN- γ Suppresses IL-10 Production and Synergizes with TLR2 by Regulating GSK3 and CREB/AP-1 Proteins. *Immunity* **24**, 563-574 (2006).

van Beelen, A. J. et al. Stimulation of the intracellular bacterial sensor NOD2 programs dendritic cells to promote interleukin-17 production in human memory T cells. *Immunity* **27**, 660-669 (2007).

Gringhuis, S. I. et al. Selective c-Rel activation via Malt1 controls anti-fungal T(H)-17 immunity by dectin-1 and dectin-2. *PLoS Pathog.* **7**, e1001259 (2011).

Gringhuis, S. I. et al. Fucose-based PAMPs prime dendritic cells for follicular T helper cell polarization via DC-SIGN-dependent IL-27 production. *Nat. Commun.* **5**, 5074 (2014).

Reviewer Fig. 3-1 | a,b, Flow cytometry analyses of T_H polarization by staining for intracellular IL-17 and IFN γ expression (FI, fluorescence intensity) in restimulated T cells, outgrown *in vitro* by coculture of naive CD4⁺ T cells with immature DCs (iDC) or DCs primed for 3 h (**a,b**) or 48 h (**b**) with curdlan or *C. albicans* CBS2712, in the absence or presence of recombinant human (rh) TGF β 1 ($n = 3$). Representative dot plots for independent donors are shown, with % positive cells indicated in each quadrant. In (**b**), the % IFN γ ⁻ and % IFN γ ⁺ cells per IL-17⁺ T cells are shown. Data in (**b**) represent mean \pm s.d. of independent donors.

Reviewer Fig. 3-2 | RT-PCR analyses of *IFNB*, *IL6* and *IL1B* relative mRNA levels in DCs at 2 or 6 h after stimulation with a concentration range of curdlan ($n = 3$). Results from RT-PCR were normalized to the expression of reference household gene *GAPDH* and shown relative to 2 h curdlan (10 µg/ml) stimulation (*IFNB*) or 6 h curdlan (10 µg/ml) stimulation (*IL6*, *IL1B*). Data represent mean \pm s.d. of independent donors.

Reviewer Fig. 3-3 | ELISA for quantification of IL-17A or IFN γ in the supernatant of restimulated T cells, outgrown *in vitro* by coculture of memory CD4⁺ T cells with immature DCs (unstimulated) or DCs primed for 48 h with curdlan or *C. albicans* CBS2712, in the presence of blocking IFN α/β R antibodies or rhIFN β (during DC stimulation) and/or blocking TGF β R antibodies or rhTGF β 1 (during DC-T coculture) ($n = 3$). Data represent mean \pm s.d. of independent donors. ** $P < 0.01$, * $P < 0.05$ (Student's t -test).

Point-by-point reply to Reviewer #3:

The authors have addressed most of my points raised in the first review. My biggest concern was about the actual mechanisms underlying the induction of IFN γ /IL-17 in human T cells. The addition of data on naive Th cell differentiation was an important and essential point. To my knowledge the mechanisms of dectin-1 mediated induction of human Th17 cells is also new and could be followed in more detail. However, the data provided so far still do not fully convince me and I have especially some technical concerns regarding intracellular cytokine stainings, which in my eyes prevent publication, if not clarified.

1. General quality of the culture system:

In my previous review I raised the point that it is difficult to understand the co-culture system the authors call "T cell outgrowth assay" (14 days, 2 day preactivation of DCs). The authors now show that DCs stimulated for 3h and 48h have different effects on T cells. This is ok and interesting. However, the authors still do not provide any data about the growth/survival of the T cells throughout the culture. It is well known that during culture small subsets may grow out especially if the overall proliferation is not homogenous. Thus standard parameters should be cell survival, expansion, viability allowing to estimate whether the cultures are comparable.

The new short term co-cultures of DCs with naïve T cells corroborate the results obtained with the long term co-cultures (T cell outgrowth assay). These data clearly show that short or long term assays are suitable to investigate the signals required for induction of pathogenic and non-pathogenic T cells by DCs.

We agree with the reviewer that during culture small subsets may grow out, but that is exactly the idea behind the DC-T cell co-cultures and it is therefore also called T cell outgrowth assay. In the DC-T cell co-cultures, only T cells that react to the alloantigens on DCs will survive and expand over 11-14 days allowing them to become the majority of the culture. During that time (but also as we have shown within 5 days) they develop their T_H17 signature profile depending on the cues provided by the DCs. Thus, overall proliferation will indeed not be homogenous, and as such survival data will not add to the conclusions that can be drawn. When the signal from the mature DCs changes, so will the response of the T cells, meaning that cell survival/expansion/viability of the cultures will not be comparable, but reflect the change of the provided DC signal. This is particularly clear for our co-cultures in the presence of rhIFN β or rhIL-27 that demonstrated significantly less T cell proliferation, which is exactly in line with our hypothesis and known literature.

We can provide the viability and cell survival if necessary but it will not add to the conclusions as there will be differences between conditions. We can clarify this in the discussion and also emphasize that the short term DC-T cell co-cultures corroborate the long term T cell outgrowth assay. These assays have been performed by various groups including Zielinski *et al.* and are considered the best assays to investigate differentiation cues.

I still disagree with the authors and would say that cytokine induction should be visible early, after a few days. Otherwise it would make biologically no sense.

We agree with the reviewer that cytokine induction is visible early, after a few days as we indeed observe a strong cytokine induction in the short 5 days time period of our DC-naïve T cell co-cultures (**Fig. 2a,b** and **Supplementary Fig. 5**). The data with the new short term DC-T cell co-cultures corroborate our results obtained with the long term DC-memory T cell cultures. We can also perform a short term culture with memory T cells, but all reviewers seem to agree that the inclusion of the naïve T cell-DC co-cultures support our hypothesis.

This is important to understand whether we really talk about induction of IFN γ /IL-17 or just better survival, e.g. due to more costimulation by dectin1 matured DC. As previously mentioned, I don't understand why human memory T cells activated for 14 days should not contain any Th17 cells (Fig 2 e, f). Sallusto and colleagues showed nicely that Th17 and Th17(IFN γ) cells and other Th subsets can be directly sorted from blood according to chemokine receptor expression profile, and these cells produce a high amount of signature cytokines. I still think it is required to sort different memory subsets before coculture to distinguish between induction or survival.*

We completely agree with the reviewer that T_H17 cells can be directly sorted from blood – although we prefer using IL-17 stainings over chemokine receptor stainings. We used exactly that strategy to analyze IL-17+ CD4 T cells from blood of healthy persons and Crohn's disease patients (**Fig. 2i,j**). However, these cells constitute only a small fraction: we isolate ~ 100.000 IL-17+ cells from

15 million CD4⁺ T cells (this varies per donor), which is < 1%. These numbers are in line with literature [Simplified assay for enrichment of primed human Th17 and Tc17 lymphocytes from peripheral blood | Translational Medicine Communications | Full Text (biomedcentral.com)]. Obviously, during our memory T cell outgrowth assay, immature DCs do not provide the signals for T cells to proliferate and after re-stimulation only those <1% will produce IL-17, which is exactly what we see (**Fig. 2e,f**).

Instead of sorting different memory T cells, we have repeated the experiments as requested with naive T cells and we observe the same changes in differentiation profiles. These data support our hypothesis that IFN signaling controls T_H17 differentiation. We would like to stress that the focus of the manuscript is not what cells are triggered to become IL-17-producing cells – although certainly an interesting research question – but that dectin-1 signaling specifically induces IL-17+IFN γ - cells. Therefore we believe that sorting different memory T cells and determine which subset is more efficiently induced into T_H17 cells is beyond the scope of our manuscript.

2. Th17 induction

Furthermore I still don't understand which signals the 48h activated DC mediate to induce IL-17 (or alternatively: support better expansion of pre-existing Th17 cells). The authors write in the response letter that this has been previously been shown to depend on IL-1,-23 but then they should show that DCs after 48h still produce these cytokines and that they are relevant for the Th17 induction. By selectively blocking these cytokines and also IL-12 it can also be excluded that IFN γ induction is not driven by IL-12 (as mentioned before to me it looks overall like strong induction of IFN γ in all cells not only in the Th17 cells).

We have performed the suggested experiments in the past (see figure below), and if required we can repeat these experiments and include these in the manuscript.

Reviewer Figure 1. IL-1b and IL-23 induce Th17 differentiation. Curdlan-stimulated DCs induce Th17 differentiation, which is inhibited by antibodies against IL-23 and IL-1b, whereas antibodies against IL-12 did not block Th17 differentiation.

As we have mentioned in the response letter, other groups have performed similar experiments with the same setup and same blocks, although the primary signal was then provided by NOD2 ligands, also showing that IL-1 β and IL-23 are required for DC-mediated IL-17 production in memory T cells (Beelen *et al.*, Immunity 2007). Obviously, it has been known for a long time that IL-1 β and IL-23 are required for this response. We regret that the reviewer does not understand how 48 h activated DCs can still produce cytokines, but our experiments clearly show that this is the case or otherwise there would not be any T_H cell development at all, either in our co-cultures with memory T cells or naive T cells.

The same is true for the naïve T cell culture, the authors write that many more naïve T cells have to be used to set up the culture and to receive sufficient amounts for analysis. In my experience this indicates that the T cells poorly grow and many may eventually die.

The reviewer has misunderstood our comment – we meant that when we analyze our outgrown T cells after 5 days of co-culture (as we did for the naive T cells) instead of allowing an 11-14 days proliferation period (as we did for the memory T cells), we need to start our experiments with more T cells to obtain the same amount of cells for our analyses. So it is not the case that naive T cells grow poorly compared to memory T cells but that the short culture period of 5 days results in less cells for analysis. If required we can also perform the DC-memory T cell co-cultures in 5 days to show that there is no difference between outgrowth of memory or naive T cells.

3. Stainings

- In fact the stainings of the naïve T cell cultures also appear untypically to me: in my experience cytokine stainings give rarely such clear round shaped populations (Fig 2a, b) compare also to Fig 2 e, f). Have the same antibodies been used in 2 a,b vs e.f ? Have doublets been excluded? Here I would like to see the original gates, scatter gates, ...

We have provided all the original scatter plots and gating strategies for **Fig. 2a,b** and **Fig. 2e,f** as pdf attachments and we can include these in the manuscript. We have used the same antibodies throughout the entire manuscript and thus also in **Fig. 2a,b** vs **Fig. 2e,f**. Indeed, when analyzing the outgrown T cells after 5 days DC-naïve T cell co-cultures, we had to exclude doublets as these cells are still very activated and cluster together. When analyzing the outgrown T cells after 11-14 day DC-memory T cell co-cultures, there are hardly any doublets present as these cells have been cultured until becoming quiescent.

- Especially figure 2 m) appears very untypical to me but maybe I misunderstand it: the dot plots are intracellular staining for IFN γ and IL-10 from T cells from one culture? These two populations have then been further analyzed for the genes shown in the histograms? If this is the case I have severe concerns, whether these data are technically well generated. I never saw IL-10 and IFN-g staining in T cells in two separated populations like this, not to say that this is not possible. This must be an artefact.

Also intracellular stainings for transcription factors in our hands never give such homogenous peaks. Also here the authors have to provide the gating, dot plots (double stainings?), etc to make that plausible.

The dot plots are indeed intracellular triple stainings for IFN γ and IL-10 from IL-17+ cells sorted from one culture and these two populations have been further analyzed for the genes shown in the histograms. Importantly, the FACS result is certainly not an artefact, as the two separated populations of IL-10+ and IFN γ + cells within the IL-17+ cells coincide with the real time PCR analyses we did on sorted IL-17+IFN γ - and IL-17+IFN γ + cells that likewise show that IL-10 expression is mostly absent in the IL-17+IFN γ + fraction (**Fig. 2k**). We see this distinction also when analyzing by real-time PCR the sorted IL-17+IFN γ - and IL-17+IFN γ + cells after 5 days DC-naïve T cell co-cultures (**Fig. 2k**).

We are surprised that the reviewer doubts our transcription factor stainings. These stainings are done on cells that have been allowed to proliferate until becoming quiescent and have had time to develop the signature profiles to their full potential. Why would there need to be a broad distribution of transcription factor expression levels between these cells? The remarks of the reviewer indicate that they analyze transcription factor expression much sooner after the start of T_H cell development. This could easily explain why they observe more spread in their transcription factor expression patterns. We could even argue that this time effect is visible in the real time PCR analyses of the DC-naïve T cell co-cultures vs the DC-memory T cell co-cultures: the '5 days old' IL-17+IFN γ + cells produce significantly less *TBX21* (which encodes transcription factor T-bet) and *IFNG* than the '11-14 days old' IL-17+IFN γ + cells (**Fig. 2k** and

Supplementary Fig. 2b,c). Differences in culture conditions will affect T_H cell development, and we find it somewhat insulting to simply call this an artefact.

We have added all original scatter plots and gating strategies for **Fig. 2m** as pdf attachments.

Decision Letter, first revision:

Subject: Decision on Nature Immunology submission NI-A32231A-Z

Message: 22nd Mar 2022

Dear Dr. Geijtenbeek,

Thank you for your response to the referees' comments on your article, "Dectin-1 directs non-pathogenic TH17 polarization by regulating release of active TGFβ via tightly controlled type I IFN responses". Although we are interested in the possibility of publishing your study in Nature Immunology, the issues raised by the referees need to be addressed.

Please revise along the lines specified in your letter and per our conversation. At resubmission, please include a "Response to referees" detailing, point-by-point, how you addressed each referee comment. If no action was taken to address a point, you must provide a compelling argument. This response will be sent back to the referees along with the revised manuscript.

Please include a revised version of any required reporting checklist. It will be available to referees to aid in their evaluation. The Reporting Summary can be found here: <https://www.nature.com/documents/nr-reporting-summary.pdf>

Please use the link below to submit your revised manuscript and related files:
[REDACTED]

We hope to receive your revised manuscript within 2-3 months. If you cannot send it within this time, please let us know. We will be happy to consider your revision so long as nothing similar has been accepted for publication at Nature Immunology or published elsewhere.

Nature Immunology is committed to improving transparency in authorship. As part of our efforts in this direction, we are now requesting that all authors identified as 'corresponding author' on published papers create and link their Open Researcher and Contributor Identifier (ORCID) with their account on the Manuscript Tracking System (MTS), prior to acceptance. ORCID helps the scientific community achieve unambiguous attribution of all scholarly contributions. You can create and link your ORCID from the home page of the MTS by clicking on 'Modify my Springer Nature account'. For more information please visit www.springernature.com/orcid.

Sincerely,

Ioana Visan, Ph.D.
Senior Editor
Nature Immunology

Tel: 212-726-9207
Fax: 212-696-9752
www.nature.com/ni

Reviewers' Comments:

Reviewer #1:

Remarks to the Author:

The authors took a serious effort to repond to my comments. In particular the repetition of differentiation experiments with naive T cells (which was also suggested by the two other reviewers) has been performed. The rigor and the flow of the manuscript has much improved by this set of experiments.

I also appreciate the authors' effort to revisit the experimental set-up of Zielinski et al. The data that is provided in Reviewer Fig. 1-1. is important and should go into the manuscript (at least as supplementary data) since it will indeed help the field to understand these obvious discrepancies in both manuscripts.

Finally I have noticed a few minor mistakes in the revised manuscript:

Page 10: Second line: here it must read "... purified IFNg+IL-17+ cells...." instead of

"...IFNg-IL-17+..."

Page 14: Line 330: here it must read "... outgrown TH17 cells from IFNg-IL-17+ cells...." instead of "... IFNg+IL-17-....".

Reviewer #2:

Remarks to the Author:

The Authors have satisfactorily addressed the concerns raised by this reviewer in the first round of revision. Congratulations to the Authors on a nice study!

Reviewer #3:

Remarks to the Author:

Review Gringhuis et al

The authors have addressed most of my points raised in the first review. My biggest concern was about the actual mechanisms underlying the induction of IFNg/IL-17 in human T cells.

The addition of data on naïve Th cell differentiation was an important and essential point. To my knowledge the mechanisms of dectin-1 mediated induction of human Th17 cells is also new and could be followed in more detail.

However, the data provided so far still do not fully convince me and I have especially some technical concerns regarding intracellular cytokine stainings, which in my eyes prevent publication, if not clarified.

1. General quality of the culture system:

In my previous review I raised the point that it is difficult to understand the co-culture system the authors call "T cell outgrowth assay" (14 days, 2 day preactivation of DCs). The authors now show that DCs stimulated for 3h and 48h have different effects on T cells. This is ok and interesting.

However, the authors still do not provide any data about the growth/survival of the T cells throughout the culture. It is well known that during culture small subsets may grow out especially if the overall proliferation is not homogenous. Thus standard parameters should be cell survival, expansion, viability allowing to estimate whether the cultures are comparable.

I still disagree with the authors and would say that cytokine induction should be visible early, after a few days. Otherwise it would make biologically no sense.

This is important to understand whether we really talk about induction of IFNg/IL-17 or just better survival, e.g. due to more costimulation by dectin1 matured DC. As previously mentioned, I don't understand why human memory T cells activated for 14 days should not contain any Th17 cells (Fig 2 e, f). Sallusto and colleagues showed nicely that Th17 and Th17*(IFNg) cells and other Th subsets can be directly sorted from blood according to chemokine receptor expression profile, and these cells produce a high amount of signature cytokines. I still think it is required to sort different memory subsets before coculture to distinguish between induction or survival.

2. Th17 induction

Furthermore I still don't understand which signals the 48h activated DC mediate to induce

IL-17 (or alternatively: support better expansion of pre-existing Th17 cells). The authors write in the response letter that this has been previously been shown to depend on IL-1,-23 but then they should show that DCs after 48h still produce these cytokines and that they are relevant for the Th17 induction. By selectively blocking these cytokines and also IL-12 it can also be excluded that IFN γ induction is not driven by IL-12 (as mentioned before to me it looks overall like strong induction of IFN γ in all cells not only in the Th17 cells).

The same is true for the naïve T cell culture, the authors write that many more naïve T cells have to be used to set up the culture and to receive sufficient amounts for analysis. In my experience this indicates that the T cells poorly grow and many may eventually die.

3. Stainings

- In fact the stainings of the naïve T cell cultures also appear untypically to me: in my experience cytokine stainings give rarely such clear round shaped populations (Fig 2a, b) compare also to Fig 2 e, f). Have the same antibodies been used in 2 a,b vs e.f ?. Have doublets been excluded? Here I would like to see the original gates, scatter gates, ...

- Especially figure 2 m) appears very untypical to me but maybe I misunderstand it: the dot plots are intracellular staining for IFN γ and IL-10 from T cells from one culture? These two populations have then been further analyzed for the genes shown in the histograms? If this is the case I have severe concerns, whether these data are technically well generated. I never saw IL-10 and IFN-g staining in T cells in two separated populations like this, not to say that this is not possible. This must be an artefact. Also intracellular stainings for transcription factors in our hands never give such homogenous peaks. Also here the authors have to provide the gating, dot plots (double stainings?), etc to make that plausible.

Author Rebuttal, first revision:

Response to Reviewer #1

The authors took a serious effort to repond to my comments. In particular the repetition of differentiation experiments with naive T cells (which was also suggested by the two other reviewers) has been performed. The rigor and the flow of the manuscript has much improved by this set of experiments.

I also appreciate the authors' effort to revisit the experimental set-up of Zielinski et al. The data that is is provided in Reviewer Fig. 1-1. is important and should go into the manuscript (at least as supplementary data) since it will indeed help the field to understand these obvious discrepancies in both manuscripts.

We thank the reviewer for the positive comments. As requested, we have included the data from Reviewer Fig. 1-1 in the manuscript in **Supplementary Fig. 9**.

Finally I have noticed a few minor mistakes in the revised manuscript:

Page 10: Second line: here it must read "... purified IFNg+IL-17+ cells...." instead of "...IFNgIL-17+..."

Page 14: Line 330: here it must read ".... outgrown TH17 cells from IFNg-IL-17+ cells...." instead of "... IFNg+IL-17-....".

We thank the reviewer for pointing out these mistakes. We have rectified them in the manuscript.

Response to Reviewer #2

The Authors have satisfactorily addressed the concerns raised by this reviewer in the first round of revision. Congratulations to the Authors on a nice study!

We thank the reviewer for the positive comments!

Response to Reviewer #3

The authors have addressed most of my points raised in the first review. My biggest concern was about the actual mechanisms underlying the induction of IFNg/IL-17 in human T cells. The addition of data on naïve Th cell differentiation was an important and essential point. To my knowledge the mechanisms of dectin-1 mediated induction of human Th17 cells is also new and could be followed in more detail. However, the data provided so far still do not fully convince me and I have especially some technical concerns regarding intracellular cytokine stainings, which in my eyes prevent publication, if not clarified.

1. General quality of the culture system:

In my previous review I raised the point that it is difficult to understand the co-culture system the authors call “T cell outgrowth assay” (14 days, 2 day preactivation of DCs). The authors now show that DCs stimulated for 3h and 48h have different effects on T cells. This is ok and interesting.

However, the authors still do not provide any data about the growth/survival of the T cells throughout the culture. It is well known that during culture small subsets may grow out especially if the overall proliferation is not homogenous. Thus standard parameters should be cell survival, expansion, viability allowing to estimate whether the cultures are comparable.

As requested, we have now included proliferation and survival data as well as expansion curves for both naive and memory CD4⁺ T cells that were cocultured with curdlan- or *C. albicans*-primed DCs in **Supplementary Fig. 3**. These data show that primed but not immature DCs in the DC-naive T cell cocultures (in the absence of SEB) induced proliferation of a small percentage of T cells, likely in an Ag-dependent manner. During DC-memory T cell cocultures, which are performed in the presence of SEB, both immature and primed DCs induced proliferation of the majority of the T cells, in an Ag-independent manner. Either inhibition or enhancement of type I IFN responses during DC priming, despite changing the cytokine expression profile of the primed DCs, did not alter the observed proliferation and survival profiles of the activated T cells (either naive or memory) during cocultures. Since the cocultures are comparable with regard to proliferation and survival, it seems unlikely that changes in these parameters are responsible for the observed differences in IL-17/IFN γ expression by T cells. We have also clarified this in the manuscript.

I still disagree with the authors and would say that cytokine induction should be visible early, after a few days. Otherwise it would make biologically no sense.

We apologize for the confusion as we previously agreed with the reviewer that cytokine induction in T cells should be visible after a few days: we showed strong cytokine induction in naive T cells after 5 days of coculture with primed DCs (**Fig. 2a,b** and **Supplementary Fig. 9** – numbering corresponding to revised manuscript). We have now also included data after 5 days of DC-memory T cell cocultures, in which we also observed a clear cytokine induction (**Supplementary Fig. 5**). We have clarified this in the manuscript.

This is important to understand whether we really talk about induction of IFN γ /IL-17 or just better survival, e.g. due to more costimulation by dectin1 matured DC. As previously mentioned, I don't understand why human memory T cells activated for 14 days should not

contain any Th17 cells (Fig 2 e, f). Sallusto and colleagues showed nicely that Th17 and Th17(IFN γ) cells and other Th subsets can be directly sorted from blood according to chemokine receptor expression profile, and these cells produce a high amount of signature cytokines. I still think it is required to sort different memory subsets before coculture to distinguish between induction or survival.*

We apologize for the confusion but we do detect T_H17 cells after coculture of memory T cells with immature DCs, although at low levels (<1%) and with donor variation (compare **Fig. 2e** and **2f**). Like Sallusto and colleagues, we have sorted T_H17 cells from blood (**Fig. 2i,j**) and these cells constitute only a small fraction: we isolate ~ 100.000 IL-17⁺ cells from 15 million CD4⁺ T cells (this varies per donor), which is <1%. These numbers are also in line with literature (Dagur *et al.*, 2019). After homogenous proliferation due to the presence of SEB, but without specific induction (as observed with our immature DCs), only those <1% will produce IL-17 after re-stimulation, which is exactly what we see (**Fig. 2e,f**).

We agree with the reviewer that it is important to distinguish whether outgrowth of IL17/IFN γ -expressing cells is due to induction or better survival. Since proliferation and survival numbers did not differ between immature and primed DCs (**Supplementary Fig. 3**), our data suggest that the induction of IL-17⁺ cells after restimulation of memory T cells cocultured with primed DCs is due to the differentiation of memory T cells into T_H17 cells due to dectin-1 cues, and not due to differences in survival. It would indeed be interesting to further examine the plasticity of Th cells and investigate whether a specific memory T cell subset differentiates into T_H17 cells but we feel that this is beyond the scope of our manuscript.

2. Th17 induction

Furthermore I still don't understand which signals the 48h activated DC mediate to induce IL17 (or alternatively: support better expansion of pre-existing Th17 cells). The authors write in the response letter that this has been previously been shown to depend on IL-1,-23 but then they should show that DCs after 48h still produce these cytokines and that they are relevant for the Th17 induction. By selectively blocking these cytokines and also IL-12 it can also be excluded that IFN γ induction is not driven by IL-12 (as mentioned before to me it looks overall like strong induction of IFN γ in all cells not only in the Th17 cells).

We have determined the expression of IL-1 β , IL-23, IL-6 and IL-12 within the supernatant of (un)stimulated DCs over time. We have added these data in **Reviewer Fig. 1**. While IL-1 β and IL-6 expression can be detected after 8 h already, IL-23 and IL-12 expression takes longer. After 24 h, the expression levels of all four cytokines in the supernatant reach a plateau level. Interestingly, after washing the DCs after 48 h, we observe that primed DCs still produce all cytokines. Moreover, when we

add either naive or memory T cells, the induction of all cytokines is enhanced. Thus, DCs still produce all cytokines after 48 h and these cytokines will be involved in instructing the T cells. As requested, we have performed blocking experiments and our data show that both IL-1 β and IL-23, but not IL-12, produced as a result of dectin-1 signaling, are required for the induction of IL-17 expression in both naive and memory T cells. We have included these data (**Supplementary Fig. 5**) and clarified our findings in the manuscript.

The same is true for the naïve T cell culture, the authors write that many more naïve T cells have to be used to set up the culture and to receive sufficient amounts for analysis. In my experience this indicates that the T cells poorly grow and many may eventually die.

We are afraid that the reviewer has misunderstood our comment – we meant that when we analyze our outgrown T cells after 5 days of co-culture (as we did for the naive T cells) instead of allowing an 11-14 days proliferation period (as we did for the memory T cells), we need to start our experiments with more T cells to obtain the same amount of cells for our analyses. The expansion curves in **Supplementary Fig. 3c** and **g** showcase the differences in cell numbers we obtained. The proliferation data in **Supplementary Fig. 3** show that both our naive and memory T cell cocultures are growing nicely.

3. Stainings

- *In fact the stainings of the naïve T cell cultures also appear untypically to me: in my experience cytokine stainings give rarely such clear round shaped populations (Fig 2a, b) compare also to Fig 2 e, f). Have the same antibodies been used in 2 a,b vs e.f ?. Have doublets been excluded? Here I would like to see the original gates, scatter gates, ...*

We thank the reviewer for this question. We have now enlisted the help of Dr. Ester Remmerswaal within our department, who is an expert in both T lymphocytes and flow cytometry analyses, to ensure proper analyses of our data.

Previously, we did not exclude doublets or debris from our memory T cell analyses and this has caused an overestimation of double positive IL-17⁺IFN γ ⁺ cells in our experiments. We have added both our previous and new analyses, including gating strategies and scatter plots for **Fig. 2e** and **f** in **Reviewer Fig. 2** (added as a separate pdf file), to illustrate the differences. Interestingly, undisturbed dectin-1 signaling leads to even less pathogenic T_H17 cell induction. Disturbances in type I IFN responses still lead to a significant increase in IL-17⁺IFN γ ⁺ T cells. Thus, strictly regulated dectin-1 signaling seems even more important to prevent the generation of pathogenic T_H17 cells. We have now included our

gating strategy in the manuscript in **Supplementary Fig. 4**. This does not affect our analyses of sorted T_H17 cells after DC-memory T cell experiments (real-time PCR experiments, FACS stainings) as here the correct gating strategies had been applied (see also **Reviewer Fig. 2** (added as a separate pdf file)).

We have used the same antibodies throughout the entire manuscript and thus also in **Fig. 2a,b** vs **Fig. 2e,f**. We have no explanation for the different shape of the cytokine stainings and can only speculate that this is an intrinsic characteristic of the naive T cells. Also we observe differences in the shape of the cytokine stainings between the resting and blast cells after re-stimulation (**Supplementary Fig. 4**). We have also re-analyzed our DCnaive T cell coculture data as previously we excluded blast cells from our naive T cell analyses, erroneously identifying these as doublets. We have added both our previous and new analyses, including gating strategies and scatter plots for **Fig. 2a** and **b** in **Reviewer Fig. 2** (added as a separate pdf file), to illustrate the differences. As the IL-17 and IFN γ expression levels of the blast cells differ from that of the resting cells after re-stimulation, it became necessary to separate both populations before our analyses of the IL-17 $^+$ IFN γ^- and IL-17 $^+$ IFN γ^+ cells, otherwise IL-17 $^+$ IFN γ^- blast cells would overlap with IL-17 $^+$ IFN γ^+ resting cells. As the blast cell population varies per experiment (10-40% of the lymphocyte single cell gate), this effect has smaller or larger consequences depending on the experiment. We have now included our gating strategy in the manuscript in **Supplementary Fig. 4**.

- *Especially figure 2 m) appears very untypical to me but maybe I misunderstand it: the dot plots are intracellular staining for IFN γ and IL-10 from T cells from one culture? These two populations have then been further analyzed for the genes shown in the histograms? If this is the case I have severe concerns, whether these data are technically well generated. I never saw IL-10 and IFN-g staining in T cells in two separated populations like this, not to say that this is not possible. This must be an artefact. Also intracellular stainings for transcription factors in our hands never give such homogenous peaks. Also here the authors have to provide the gating, dot plots (double stainings?), etc to make that plausible.*

The dot plots are indeed intracellular triple stainings for IFN γ and IL-10 from IL-17 $^+$ cells sorted from one culture and these two populations have been further analyzed for the genes shown in the histograms. Importantly, the FACS results are certainly not an artefact, as the two separated populations of IL-10 $^+$ and IFN γ^+ cells within the IL-17 $^+$ cells coincide with the real time PCR analyses we did on sorted IL-17 $^+$ IFN γ^- and IL-17 $^+$ IFN γ^+ cells that likewise show that IL-10 expression is mostly absent in the IL-17 $^+$ IFN γ^+ fraction (**Fig. 2k**). We see this distinction also when analyzing mRNA expression of the sorted IL-17 $^+$ IFN γ^- and IL-17 $^+$ IFN γ^+ cells after 5 days DC-naive T cell co-cultures by real-time PCR (**Fig. 2k**).

We have included all original scatter plots and gating strategies for **Fig. 2m** in **Reviewer Fig. 2** (added as a separate pdf file). We can only speculate on the differences between our stainings and those of the reviewer. Possibly, the more homogenous expression of transcription factors is due to the fact that these T cells have been grown until they become quiescent and as such have developed the signature profiles to their full potential. It is possible that the reviewer analyses transcription factor expression much sooner after the start of T_H cell development, which would perhaps lead to more spread in their transcription factor expression patterns as the signature is not yet fully developed.

References

Dagur, P. K., Stansky, E., Saxena, A., Biancotto, A. & McCoy Jr, J. P. Simplified assay for enrichment of primed human Th17 and Tc17 lymphocytes from peripheral blood. *Transl. Med. Commun.* **4**, 11 (2019).

Reviewer Fig. 1 | IL-1 β , IL-23, IL-6 and IL-12 expression over time in supernatants of primed DCs. ELISA for quantification of IL-1 β , IL-6, IL-23 and IL-12 in the supernatant of DCs at the indicated times after stimulation with curdlan or *C. albicans* CBS2712. After 48 h, either naive (Tn) T cells were added to the medium or DCs was washed and resuspended in fresh medium, either without or with naive or memory (Tm) T cells. Data represent mean \pm s.d. of 3 independent donors.

Response to Reviewer #3

Thanks to the authors for providing all the additional data and controls and for answering all my questions. This was very helpful and improved the manuscript.

However, there are still two last points related to figure 2. The data in figure 2 provide such a black and white picture of pathogenic versus non pathogenic Th17 cells (which is a concept very much discussed whether these phenotypes exist in vivo) that I have difficulties to fully comprehend this. But this is more based on experience rather than on specific points which look critical, therefore I would like to state that as my personal opinion.

We thank the reviewer for the positive comments on the new data in our revised manuscript.

Fig 2 i, j

If I understand correctly: Here the authors analyse IL-17+ cells sorted ex vivo from PMA/iono stimulated T cells from healthy donors and CD patients, sorted for IL-17 and subsequently analyzed for the various genes by RT-PCR?

The authors find an almost black and white difference between IL-17+ cells from healthy donors and CD patients (n=4) with respect to genes expressed by so called pathogenic and non-pathogenic Th17 cells. The analysis judged from the gating strategies looks fine, but such a marked difference is surprising and I am not aware that something similarly drastic has previously been described.

My impression from the overall literature and personal experience is that only mild differences between patients and healthy can be observed. So the authors should at least comment on this striking observation in their assay system (which is sophisticated) and discrepancy to published data (or prove me wrong with examples from the literature). They should also discuss that additional functional data (e.g. cytokine secretion data, i.c. stainings, more and defined patients ...) may be necessary to support this finding in the future.

The reviewer is correct that we have analysed IL-17+ cells sorted from healthy donors and acute untreated CD patients. We have also sorted IL-17+ cells from CD patients that are in remission, and the differences with healthy donors are much milder (Reviewer Figure 1), suggesting that the striking differences between healthy and acute CD patients might be because the CD patients used in the manuscript are untreated and acute. We are also not aware of any study that has performed a similar technical challenging analysis, which makes it very interesting and we will further pursue this in another study.

As requested, we can discuss the observed differences between healthy and CD patients in the manuscript and also discuss that further study is necessary to investigate the striking Th17 phenotype in acute CD patients.

Fig 2m: These are stainings for IL-10 and IFN γ within T cells sorted for IL-17 + after culture (according to the raw data files provided). From the dot plots and gating strategies, I did not identify any obvious issues.

However, from all my experience of 25 years cytokine staining of human and murine T cells focusing on IL-10 from Th1/Th17 subsets, I have never observed such well separated IFN γ versus IL-10 producing populations of Th17 cells and such extremely high frequencies of 70% IL-10 (uniform staining). All other cytokine stainings in the manuscript look different, although there are no other IL-10 stainings provided in the manuscript.

The high frequencies of IL-10 population and clear separation might be due to our DC stimulation and T cell outgrowth assay. As our data show curdlan and *C. albicans* induce a very strong Th17 phenotype. The uniform staining of IL-10 might be caused by the procedure as the cells are permeabilized/fixed, sorted and permeabilized/fixed again (using a special kit for transcription factor staining) before staining. However, the IL-10 expression data are supported by the mRNA expression profiles (Fig. 2I). We can discuss this in the manuscript.

Decision Letter, second revision:

Subject: Decision on Nature Immunology submission NI-A32231B

Message: Dear Dr. Geijtenbeek,

Thank you for your response to the reviewers' comments on your manuscript "Dectin-1 directs non-pathogenic TH17 polarization by regulating release of active TGF β via tightly controlled type I IFN responses". We are happy to inform you that if you revise your manuscript appropriately in response to the referees' comments and our editorial requirements your manuscript should be publishable in Nature Immunology.

Please revise your manuscript to address the reviewer's comments and as outlined in your letter. At resubmission, please include a point-by-point response to the referees' comments, noting the pages and lines where the changes can be found in the revision. Please highlight the changes in the revised manuscript as well. Please note that articles for Nature Immunology have a word limit of 4000 words for the introduction, results and discussion and a limit of 50 references for the main text. The discussion should not exceed 800 words.

We are trying to improve the quality and transparency of methods and statistics reporting in our papers (please see our editorial in the May 2013 issue). Please update the Life Sciences Reporting Summary, and supplements if applicable, with any information

relevant to any new experiments and upload it (as a Related Manuscript File) along with the files for your revision. If nothing in the checklist has changed, please upload the current version again.

TRANSPARENT PEER REVIEW

Nature Immunology offers a transparent peer review option for new original research manuscripts submitted from 1st December 2019. We encourage increased transparency in peer review by publishing the reviewer comments, author rebuttal letters and editorial decision letters if the authors agree. Such peer review material is made available as a supplementary peer review file. **Please state in the cover letter 'I wish to participate in transparent peer review' if you want to opt in, or 'I do not wish to participate in transparent peer review' if you don't.** Failure to state your preference will result in delays in accepting your manuscript for publication.

ORCID

Nature Immunology is committed to improving transparency in authorship. As part of our efforts in this direction, we are now requesting that all authors identified as 'corresponding author' on published papers create and link their Open Researcher and Contributor Identifier (ORCID) with their account on the Manuscript Tracking System (MTS), prior to acceptance. ORCID helps the scientific community achieve unambiguous attribution of all scholarly contributions. For more information please visit www.springernature.com/orcid.

Before resubmitting the final version of the manuscript, if you are listed as a corresponding author on the manuscript, please follow the steps below to link your account on our MTS with your ORCID. If you don't have an ORCID yet, you will be able to create one in minutes. If you are not listed as a corresponding author, please ensure that the corresponding author(s) comply.

1. From the home page of the [MTS](https://mts-ni.nature.com/cgi-bin/main.plex) click on '**Modify my Springer Nature account**' under '**General tasks**'.
2. In the '**Personal profile**' tab, click on '**ORCID Create/link an Open Researcher Contributor ID(ORCID)**'. This will re-direct you to the ORCID website.
- 3a. If you already have an ORCID account, enter your ORCID email and password and click on '**Authorize**' to link your ORCID with your account on the MTS.
- 3b. If you don't yet have an ORCID, you can easily create one by providing the required information and then click on '**Authorize**'. This will link your newly created ORCID with your account on the MTS.

IMPORTANT: All authors identified as 'corresponding authors' on the manuscript must

follow these instructions. Non-corresponding authors do not have to link their ORCID, but please note that it will not be possible to add/modify ORCID at proof. Thus, if they wish to have their ORCID added to the paper, they must also follow the above procedure prior to acceptance.

To support ORCID's aims, we only allow a single ORCID identifier to be attached to one account. If you have any issues attaching an ORCID identifier to your Manuscript Tracking System account, please contact the [Platform Support Helpdesk](http://platformsupport.nature.com/).

We hope that you will support this initiative and supply the required information. Should you have any query or comments, please do not hesitate to contact our editorial assistant at immunology@us.nature.com.

Nature Immunology has now transitioned to a unified Rights Collection system which will allow our Author Services team to quickly and easily collect the rights and permissions required to publish your work. Once your paper is accepted, you will receive an email in approximately 10 business days providing you with a link to complete the grant of rights. If you choose to publish Open Access, our Author Services team will also be in touch at that time regarding any additional information that may be required to arrange payment for your article.

In recognition of the time and expertise our reviewers provide to Nature Immunology's editorial process, we would like to formally acknowledge their contribution to the external peer review of your manuscript entitled "Dectin-1 directs non-pathogenic TH17 polarization by regulating release of active TGFβ via tightly controlled type I IFN responses". For those reviewers who give their assent, we will be publishing their names alongside the published article.

When you are ready to submit your revised manuscript, please use the URL below to submit the revised version: [REDACTED]

We hope to receive your revised manuscript in 10 days, by 5th Aug 2022. Please let us know if circumstances will delay submission beyond this time. If you have any questions please do not hesitate to contact me.

Sincerely,

Ioana Visan, Ph.D.
Senior Editor
Nature Immunology

Tel: 212-726-9207
Fax: 212-696-9752
www.nature.com/ni

Reviewer #3

(Remarks to the Author)

Thanks to the authors for providing all the additional data and controls and for answering all my questions. This was very helpful and improved the manuscript.

However, there are still two last points related to figure 2. The data in figure 2 provide such a black and white picture of pathogenic versus non pathogenic Th17 cells (which is a concept very much discussed whether these phenotypes exist in vivo) that I have difficulties to fully comprehend this. But this is more based on experience rather than on specific points which look critical, therefore I would like to state that as my personal opinion.

Fig 2 i, j

If I understand correctly: Here the authors analyse IL-17+ cells sorted ex vivo from PMA/iono stimulated T cells from healthy donors and CD patients, sorted for IL-17 and subsequently analyzed for the various genes by RT-PCR?

The authors find an almost black and white difference between IL-17+ cells from healthy donors and CD patients (n=4) with respect to genes expressed by so called pathogenic and non-pathogenic Th17 cells. The analysis judged from the gating strategies looks fine, but such a marked difference is surprising and I am not aware that something similarly drastic has previously been described.

My impression from the overall literature and personal experience is that only mild differences between patients and healthy can be observed. So the authors should at least comment on this striking observation in their assay system (which is sophisticated) and discrepancy to published data (or prove me wrong with examples from the literature). They should also discuss that additional functional data (e.g. cytokine secretion data, i.c. stainings, more and defined patients ...) may be necessary to support this finding in the future.

Fig 2m: These are stainings for IL-10 and IFN γ within T cells sorted for IL-17 + after culture (according to the raw data files provided). From the dot plots and gating strategies, I did not identify any obvious issues.

However, from all my experience of 25 years cytokine staining of human and murine T cells focusing on IL-10 from Th1/Th17 subsets, I have never observed such well separated IFN γ versus IL-10 producing populations of Th17 cells and such extremely high frequencies of 70% IL-10 (uniform staining). All other cytokine stainings in the manuscript look different, although there are no other IL-10 stainings provided in the manuscript.

Author Rebuttal, second revision:

Response to Reviewer #3

Thanks to the authors for providing all the additional data and controls and for answering all my questions. This was very helpful and improved the manuscript.

However, there are still two last points related to figure 2. The data in figure 2 provide such a black and white picture of pathogenic versus non pathogenic Th17 cells (which is a concept very much discussed whether these phenotypes exist in vivo) that I have difficulties to fully comprehend this. But this is more based on experience rather than on specific points which look critical, therefore I would like to state that as my personal opinion.

We thank the reviewer for the positive comments on the new data in our revised manuscript.

Fig 2 i, j

If I understand correctly: Here the authors analyse IL-17+ cells sorted ex vivo from PMA/iono stimulated T cells from healthy donors and CD patients, sorted for IL-17 and subsequently analyzed for the various genes by RT-PCR?

The authors find an almost black and white difference between IL-17+ cells from healthy donors and CD patients (n=4) with respect to genes expressed by so called pathogenic and non-pathogenic Th17 cells. The analysis judged from the gating strategies looks fine, but such a marked difference is surprising and I am not aware that something similarly drastic has previously been described.

My impression from the overall literature and personal experience is that only mild differences between patients and healthy can be observed. So the authors should at least comment on this striking observation in their assay system (which is sophisticated) and discrepancy to published data (or prove me wrong with examples from the literature). They should also discuss that additional functional data (e.g. cytokine secretion data, i.c. stainings, more and defined patients ...) may be necessary to support this finding in the future.

The reviewer is correct that we have analyzed IL-17+ cells sorted from healthy donors and CD patients. These CD patients had acute and untreated immunopathogenesis. We have also sorted IL-17+ cells from CD patients that are in remission, and the differences with healthy donors are much less pronounced, suggesting that the striking differences between healthy and CD donors as shown in the manuscript are because of the disease status of the donors. We are also not aware of any study that has performed a similar technical challenging analysis, which makes it very interesting and we will further pursue this in another study. We have now added the data from the treated CD patients that show only mild differences with healthy persons in **Supplementary Fig. 6a** and described this observation in the text, page 8, line 184-186.

Fig 2m: These are stainings for IL-10 and IFN γ within T cells sorted for IL-17 + after culture (according to the raw data files provided). From the dot plots and gating strategies, I did not identify any obvious issues.

However, from all my experience of 25 years cytokine staining of human and murine T cells focusing on IL-10 from Th1/Th17 subsets, I have never observed such well separated IFN γ versus IL-10 producing populations of Th17 cells and such extremely high frequencies of 70% IL-10 (uniform staining). All other cytokine stainings in the manuscript look different, although there are no other IL-10 stainings provided in the manuscript.

The high frequencies of IL-10-expressing cells and clear separation might be due to our DC stimulation and T cell outgrowth assay: our data show that curdlan and *C. albicans* induce a very strong T_H17 phenotype, while the setup of the outgrowth assay (especially the prolonged culture) is beneficial for obtaining high frequencies of induced subsets. The

1

uniform staining of IL-10 might be caused by the procedure as the cells are permeabilized/fixed, sorted and permeabilized/fixed again (using a special kit for transcription factor staining) before staining. However, the IL-10 expression data are supported by the mRNA expression profiles (**Fig. 2I**).

Decision Letter, third revision:**Subject:** Your manuscript, NI-A32231C**Message:** Our ref: NI-A32231C

19th Sep 2022

Dear Dr. Geijtenbeek,

Thank you for your patience as we've prepared the guidelines for final submission of your Nature Immunology manuscript, "Dectin-1 directs non-pathogenic TH17 polarization by regulating release of active TGF β via tightly controlled type I IFN responses" (NI-A32231C). Please carefully follow the step-by-step instructions provided in the attached file, and add a response in each row of the table to indicate the changes that you have made. Please also check and comment on any additional marked-up edits we have proposed within the text. Ensuring that each point is addressed will help to ensure that your revised manuscript can be swiftly handed over to our production team.

The handling editor would like to start working on your revised paper, with all of the requested files and forms, as soon as possible and has requested that you return them by September 22nd. Please get in contact with us if you anticipate delays.

When you upload your final materials, please include a point-by-point response to any remaining reviewer comments and please make sure to upload your checklist.

If you have not done so already, please alert us to any related manuscripts from your group that are under consideration or in press at other journals, or are being written up for submission to other journals (see: <https://www.nature.com/nature-portfolio/editorial-policies/plagiarism#policy-on-duplicate-publication> for details).

In recognition of the time and expertise our reviewers provide to Nature Immunology's editorial process, we would like to formally acknowledge their contribution to the external peer review of your manuscript entitled "Dectin-1 directs non-pathogenic TH17 polarization by regulating release of active TGF β via tightly controlled type I IFN responses". For those reviewers who give their assent, we will be publishing their names alongside the published article.

Nature Immunology offers a Transparent Peer Review option for new original research manuscripts submitted after December 1st, 2019. As part of this initiative, we encourage our authors to support increased transparency into the peer review process by agreeing to have the reviewer comments, author rebuttal letters, and editorial decision letters published as a Supplementary item. When you submit your final files please clearly state in your cover letter whether or not you would like to participate in this initiative. Please note that failure to state your preference will result in delays in accepting your manuscript for publication.

Cover suggestions

As you prepare your final files we encourage you to consider whether you have any images or illustrations that may be appropriate for use on the cover of Nature

Immunology.

Nature Immunology has now transitioned to a unified Rights Collection system which will allow our Author Services team to quickly and easily collect the rights and permissions required to publish your work. Approximately 10 days after your paper is formally accepted, you will receive an email in providing you with a link to complete the grant of rights. If your paper is eligible for Open Access, our Author Services team will also be in touch regarding any additional information that may be required to arrange payment for your article.

Please note that *Nature Immunology* is a Transformative Journal (TJ). Authors may publish their research with us through the traditional subscription access route or make their paper immediately open access through payment of an article-processing charge (APC). Authors will not be required to make a final decision about access to their article until it has been accepted. [Find out more about Transformative Journals](https://www.springernature.com/gp/open-research/transformative-journals).

If you have any questions about costs, Open Access requirements, or our legal forms, please contact ASJournals@springernature.com.

Please use the following link for uploading these materials: [REDACTED]

Best regards,

Elle Morris
Senior Editorial Assistant
Nature Immunology
Phone: 212 726 9207
Fax: 212 696 9752
E-mail: immunology@us.nature.com

On behalf of

Ioana Visan, Ph.D.
Senior Editor
Nature Immunology

Tel: 212-726-9207
Fax: 212-696-9752
www.nature.com/ni

Final Decision Letter:

Subject: Decision on Nature Immunology submission NI-A32231D

Message: In reply please quote: NI-A32231D

Dear Dr. Geijtenbeek,

I am delighted to accept your manuscript entitled "Dectin-1 directs non-pathogenic TH17 polarization by regulating release of active TGF β via tightly controlled type I IFN responses" for publication in an upcoming issue of Nature Immunology.

Over the next few weeks, your paper will be copyedited to ensure that it conforms to Nature Immunology style. Once your paper is typeset, you will receive an email with a link to choose the appropriate publishing options for your paper and our Author Services team will be in touch regarding any additional information that may be required.

Please note that *Nature Immunology* is a Transformative Journal (TJ). Authors may publish their research with us through the traditional subscription access route or make their paper immediately open access through payment of an article-processing charge (APC). Authors will not be required to make a final decision about access to their article until it has been accepted. [Find out more about Transformative Journals](https://www.springernature.com/gp/open-research/transformative-journals).

Authors may need to take specific actions to achieve [compliance with funder and institutional open access mandates](https://www.springernature.com/gp/open-research/funding/policy-compliance-faqs). If your research is supported by a funder that requires immediate open access (e.g. according to [Plan S principles](https://www.springernature.com/gp/open-research/plan-s-compliance)) then you should select the gold OA route, and we will direct you to the compliant route where possible. For authors selecting the subscription publication route, the journal's standard licensing terms will need to be accepted, including [self-archiving policies](https://www.springernature.com/gp/open-research/policies/journal-policies). Those licensing terms will supersede any other terms that the author or any third party may assert apply to any version of the manuscript.

Your paper will be published online soon after we receive your corrections and will appear in print in the next available issue. Content is published online weekly on Mondays and Thursdays, and the embargo is set at 16:00 London time (GMT)/11:00 am US Eastern time (EST) on the day of publication. Now is the time to inform your Public Relations or Press Office about your paper, as they might be interested in promoting its publication. This will allow them time to prepare an accurate and satisfactory press release. Include your manuscript tracking number (NI-A32231D) and the name of the journal, which they will need when they contact our office.

About one week before your paper is published online, we shall be distributing a press release to news organizations worldwide, which may very well include details of your work. We are happy for your institution or funding agency to prepare its own press release, but it must mention the embargo date and Nature Immunology. Our Press Office will contact you closer to the time of publication, but if you or your Press Office have any enquiries in the meantime, please contact press@nature.com.

Also, if you have any spectacular or outstanding figures or graphics associated with your manuscript - though not necessarily included with your submission - we'd be delighted to consider them as candidates for our cover. Simply send an electronic version (accompanied by a hard copy) to us with a possible cover caption enclosed.

Please note that we encourage the authors to self-archive their manuscript (the accepted version before copy editing) in their institutional repository, and in their funders' archives, six months after publication. Nature Portfolio recognizes the efforts of funding bodies to increase access of the research they fund, and strongly encourages authors to participate in such efforts. For information about our editorial policy, including license agreement and author copyright, please visit www.nature.com/ni/about/ed_policies/index.html

Sincerely,

Ioana Visan, Ph.D.
Senior Editor
Nature Immunology

Tel: 212-726-9207

Fax: 212-696-9752
www.nature.com/ni